# Equivariance Everywhere All At Once: A Recipe for Graph Foundation Models

**Ben Finkelshtein**
University of Oxford

**İsmail İlkan Ceylan**
TU Wien / AITHYRA/ University of Oxford

**Michael Bronstein**
University of Oxford / AITHYRA

**Ron Levie**
Technion – Israel Institute of Technology

## Abstract

Graph machine learning architectures are typically tailored to specific tasks on specific datasets, which hinders their broader applicability. This has led to a new quest in graph machine learning: *how to build graph foundation models (GFMs)* capable of generalizing across arbitrary graphs and features? In this work, we present a recipe for designing GFMs for node-level tasks from first principles. The key ingredient underpinning our study is a systematic investigation of the symmetries that a graph foundation model must respect. In a nutshell, we argue that *label permutation-equivariance* alongside *feature permutation-invariance* are necessary in addition to the common *node permutation-equivariance* on each local neighborhood of the graph. To this end, we first characterize the space of linear transformations that are equivariant to permutations of nodes and labels, and invariant to permutations of features. We then prove that the resulting network is a *universal approximator* on multisets that respect the aforementioned symmetries. Our recipe uses such layers on the multiset of features induced by the local neighborhood of the graph to obtain a class of graph foundation models for node property prediction. We validate our approach through extensive experiments on 29 real-world node classification datasets, demonstrating both strong zero-shot empirical performance and consistent improvement as the number of training graphs increases.

## 1 Introduction

Foundation models have shown remarkable success in diverse domains such as natural language [33, 28, 41], computer vision [8, 31, 43, 1], and audio processing [6, 7, 32]. This success has sparked interest in building foundation models for graph machine learning (*graph foundation models (GFMs)*)[24, 52, 2], raising a fundamental question: *what are the key requirements for a graph machine learning architecture to generalize across tasks, graph structures, and feature and label sets?*

Significant progress has recently been made on link-level tasks in knowledge graphs [13, 21, 12, 51, 18], which typically do not involve node features. This success is partially due to the nature of knowledge graphs, where learning focuses on structured relational patterns rather than raw feature inputs. Unlike in knowledge graphs, natural language processing or computer vision, node-level tasks in general graphs face a fundamental obstacle: the absence of a shared feature "vocabulary": Features may encode textual embeddings in one dataset, molecular properties in another, or social attributes in yet another. This semantic diversity makes it inherently difficult to define a unified feature space, posing a major challenge to building graph foundation models that generalize across domains.

Classical graph neural networks (GNNs) [5, 20, 42, 47] rely on fixed feature ordering and predefined feature sets, making them ill-equipped to generalize across arbitrary graphs with varying features.

39th Conference on Neural Information Processing Systems (NeurIPS 2025).

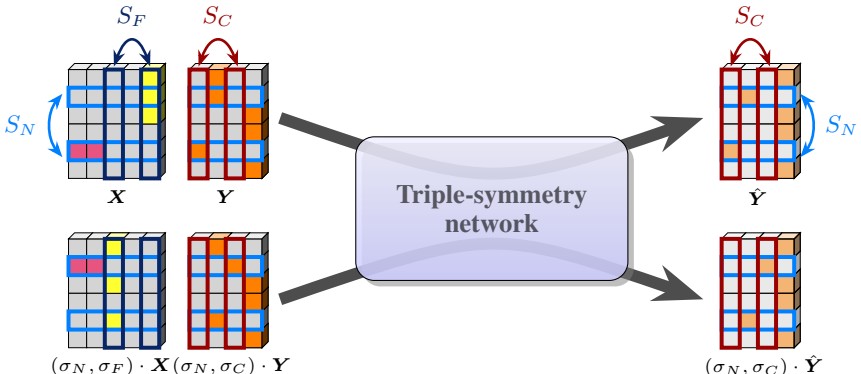

Figure 1: The input to a triple-symmetry network is a feature matrix $X$ and (possibly masked) label matrix $Y$. The encoder must be equivariant to element-wise permutations $\sigma_N \in S_N$ (affecting the rows of both $X$ and $Y$), equivariant to class label permutations $\sigma_C \in S_C$ (affecting the columns of $Y$), and invariant to feature permutations $\sigma_F \in S_F$ (affecting the columns of $X$).

To date, the only architecture shown to generalize zero-shot across graphs with rich node features is that of Zhao et al. [52], which represents a promising step forward. However, their model offers a specific empirical solution and does not address the deeper question: *how can one systematically design generalizable node-level graph foundation models?*

**Contributions.** In this paper, we directly address this challenge by introducing a theoretically-grounded recipe for designing GFMs for node-level classification and regression tasks. On the theoretical front, our starting point is identifying three symmetries that a GFM must respect:

1. *Node permutation-equivariance ($S_N$ in Figure 1).* The standard requirement in graph learning is to ensure that the predictions are invariant under isomorphisms. This is satisfied by design in message-passing GNNs through permutation-invariant local aggregation, which guarantees that a permutation of the input nodes would result in a consistent permutation of the node-level outputs.

2. *Label permutation-equivariance ($S_C$ in Figure 1).* In node-level tasks (classification or regression), outputs should also respect permutations of class labels or multi-regression targets. Permuting the ground-truth should yield a consistent permutation in the predictions, ensuring semantic consistency irrespective of the chosen ordering.

3. *Feature permutation-invariance ($S_F$ in Figure 1).* Graph node features often represent different quantities from different domains and can vary substantially across different graphs, which poses challenges in identifying a shared feature vocabulary. Hence, a GFM should not rely on feature ordering or dimensionality. This invariance ensures robust predictions irrespective of how the features are arranged across different datasets and tasks.

Having established this *triple-symmetry* criterion, we then characterize the space of linear transformations that are equivariant to permutations of nodes and labels, and invariant to permutations of features. We prove that the resulting *triple-symmetry network* (TSNet) (see Figure 1) is a *universal approximator* of functions that take multisets as inputs and respect the aforementioned symmetries. This is our key theoretical contribution, as it allows us to apply TSNets on the neighborhoods of the input graph to define a symmetry preserving message passing scheme.

The resulting TSNet scheme can be interpreted as a message passing network in which messages are not only sent between nodes but also across feature and label channels via sum aggregation. We then extend this, to introduce a more general GFM recipe: use *any* GNN aggregation to pass messages between nodes, feature channels, and labels. Each such GNN yields a GFM which transfer across graphs of arbitrary sizes and with any number of feature and label channels, for both node classification and regression. The resulting GFM is trained on a given set of graphs with their node features and labels, and then transferred to new graphs with different types of features and labels, unseen during training. To summarize, we make the following contributions:

- **Universality results.** We characterize the space of linear maps equivariant to permutations of nodes, features, and labels, and prove that the resulting multilayer architecture is a universal approximator of functions that take multiset as inputs and respect nodes, features and label symmetries (Section 4).

- **A general recipe for GFMs.** We introduce triple-symmetry graph neural networks (TS-GNNs), a modular and practical method to transform any GNN aggregation into a GFM. This allows the GNN to operate on arbitrary graphs, feature, and label sets (Section 5).

- **Empirical validation.** We evaluate our framework on 29 real-world node classification datasets, showing consistent performance improvements over the corresponding standard GNN backbone that TS-GNN uses (e.g., GraphSAGE, GAT [14, 42]). We also establish TS-GNN as the first graph foundation model to show improved zero-shot accuracy with increasing training data (Section 6.2).

## 2   Background: Graphs, Groups, and Equivariance

**Graphs and Vectors.** Vectors, matrices and tensors are denoted in bold, and for any tensor $\boldsymbol{Q}$, we denote its $i$-th row by $\boldsymbol{Q}_{i,:,:}$, its $j$-th column by $\boldsymbol{Q}_{:,j,:}$ and its $z$-th slice by $\boldsymbol{Q}_{:,:,z}$. The vector of $N$ ones and the all-ones $N \times M$ matrix are denoted by $\mathbf{1}_N$ and $\mathbf{1}_{N,M}$, respectively. Throughout the paper, $N$ denotes the number of elements in a set or the number nodes in a graph; $F$ denotes the number of input features per element; and $C$ denotes the number of output classes or regression targets. We focus on node-level tasks and consider an *undirected*, unweighted graph $G = (\mathcal{V}, \mathcal{E}, \boldsymbol{X}, \boldsymbol{Y})$, where $\boldsymbol{X} \in \mathbb{R}^{N \times F}$ is a matrix of node features and $\boldsymbol{Y} \in \mathbb{R}^{N \times C}$ is a matrix encoding labels over $C$ classes (for classification) or $C$ targets (for regression). We represent the topology of the graph $(\mathcal{V}, \mathcal{E})$ using an adjacency matrix $\boldsymbol{A} \in \{0, 1\}^{N \times N}$.

**Groups, permutations and arrays.** Let $S_N$ be the group of permutations (i.e. the *symmetric group*) over the set $[N] = \{1, 2, \ldots, N\}$. We define the action of $S_N \times S_F \times S_C$ on $\mathbb{R}^{K \times N \times (F+C)}$ by

$$((\sigma_N, \sigma_F, \sigma_C) \cdot \boldsymbol{X})_{i,j} = \begin{cases} \boldsymbol{X}_{:,\sigma_N^{-1}(i),\sigma_F^{-1}(j)} & \text{if } j \in [F], \\ \boldsymbol{X}_{:,\sigma_N^{-1}(i),F+\sigma_C^{-1}(j-F)} & \text{otherwise} \end{cases}$$

for all $(\sigma_F, \sigma_C, \sigma_N) \in S_F \times S_C \times S_N$ and $\boldsymbol{X} \in \mathbb{R}^{K \times N \times (F+C)}$. That is, $\sigma_F$ permutes the first $F$ columns (features), $\sigma_C$ permutes the last $C$ columns (labels), and $\sigma_N$ permutes the rows.

**Equivariance and Invariance.** We focus on functions that are equivariant or invariant under joint node-, feature- and label-permutations. Namely, a function $f : \mathbb{R}^{K_1 \times N \times (F+C)} \to \mathbb{R}^{K_2 \times N \times (F+C)}$ is $(S_N \times S_F \times S_C)$-*equivariant* if $f((\sigma_N, \sigma_F, \sigma_C) \cdot \boldsymbol{X}) = (\sigma_N, \sigma_F, \sigma_C) \cdot f(\boldsymbol{X})$ for all $(\sigma_N, \sigma_F, \sigma_C) \in S_N \times S_F \times S_C$ and for all $\boldsymbol{X} \in \mathbb{R}^{K_1 \times N \times (F+C)}$. Similarly, a function $f : \mathbb{R}^{K_1 \times N \times (F+C)} \to \mathbb{R}^{K_2 \times N \times C}$ is $(S_N \times S_C)$-*equivariant* and $S_F$-*invariant* if $f((\sigma_N, \sigma_F, \sigma_C) \cdot \boldsymbol{X}) = (\sigma_N, \sigma_C) \cdot f(\boldsymbol{X})$ for all $(\sigma_N, \sigma_F, \sigma_C) \in S_N \times S_F \times S_C$ and for all $\boldsymbol{X} \in \mathbb{R}^{K_1 \times N \times (F+C)}$.

**Graph neural networks (GNNs).** Our framework provides a recipe for converting some GNNs into GFMs. In particular, we focus on *message-passing neural networks* (MPNNs), a subclass of GNNs that update the initial representations $\boldsymbol{X}_v \in \mathbb{R}^{K^{(\ell)} \times N}$ of each node $v$ for $0 \le \ell \le L - 1$ iterations based on its own state and the state of its neighbors $\mathcal{N}_v$ as

$$\boldsymbol{X}_v^{(\ell+1)} = \sigma\left(\psi\left(\boldsymbol{W}_1^{(\ell)} \boldsymbol{X}_v^{(\ell)}, \{\!\{\boldsymbol{W}_2^{(\ell)} \boldsymbol{X}_u^{(\ell)} \mid u \in \mathcal{N}_v\}\!\}\right)\right),$$

where $\{\!\{\cdot\}\!\}$ denotes a multiset, $\sigma$ is a nonlinearity, $\boldsymbol{W}_1^{(\ell)}, \boldsymbol{W}_2^{(\ell)} \in \mathbb{R}^{K^{(\ell+1)} \times K^{(\ell)}}$ are weight matrices, $K^{(\ell)}$ is the dimensionality of node embeddings at layer $\ell$, $\mathcal{M}(\mathbb{R}^{K \times N})$ denote the space of multiset of elements in $\mathbb{R}^{K \times N}$ and $\psi : \mathbb{R}^{K \times N} \times \mathcal{M}(\mathbb{R}^{K \times N}) \to \mathbb{R}^{K \times N}$ is an *aggregation* function that is invariant to permutations of the elements in its multiset argument.

## 3   Related Work

**Graph foundation models for node classification.** Traditional graph neural networks (GNNs) [5, 20, 47, 42] assume fixed feature ordering and predefined feature sets. This mode of operation severely restricts their ability to generalize across graphs with unknown or dynamically varying feature distributions. Unlike domains such as natural language processing, where pre-defined embeddings allow for a shared vocabulary, this approach is unsuitable when dealing with entirely unknown or dynamic graph attributes. GraphAny [52] addresses this limitation by using a Transformer-based mixer that dynamically combines predictions from multiple closed-form least-squares classifiers. Furthermore, [52] is the first to introduce the necessary of node-equivariance, label-equivariance and feature-invariance in fully inductive node classification. While empirically effective, GraphAny simply performs a learned average across the least-squares classifiers, which is very limiting for the model design space. In contrast, we offer a class of GFMs derived from first principles, offering theoretical guarantees. Existing message-passing neural networks can be seamlessly integrated into our recipe of GFMs.

**Knowledge graph foundation models.** There are recent graph foundation models dedicated for prediction with knowledge graphs (KGs), which can generalize to any KG, including novel entities and novel relation types (unseen during training), by learning transferrable relational invariants. Notable examples include RMPI [13], InGRAM [21], ULTRA [12], TRIX [51], and MOTIF [18]. The expressive power of these knowledge graph foundation models has been recently investigated by Huang et al. [18]. Although these methods demonstrate promising generalization capabilities, they only apply to (knowledge) graphs without node features and lack any theoretical guarantees of universality. This limits their applicability to graphs with rich node features, which is critical for node-level prediction tasks.

**Invariant and equivariant networks.** Expressivity of equivariant and invariant neural networks is well-studied within the graph machine learning community. In their seminal work on DeepSets, Zaheer et al. [50] established universality of $S_N$-invariant networks over sets of size $N$. Subsequently, a universality result for permutation subgroups $G \leq S_N$ is presented by Maron et al. [25]. This universality result is extended by Segol and Lipman [38] to show that DeepSets can approximate any $S_N$-equivariant functions. Further advancements by Maron et al. [26] considered both $S_N \times H$-invariant and equivariant functions, where $H \leq S_D$ and $D$ denotes the number of input features. The proof techniques introduced in Segol and Lipman [38] were partially adopted in Maron et al. [26] and also utilized to prove universality in other contexts. For example, universality for point-cloud-equivariant functions with symmetry group $S_N \times SO(3)$ and for higher-dimensional equivariant representations was established by Dym and Maron [9] and [11], respectively. Our work utilizes the proof techniques in [38, 26] to extend the known universality guarantees to our triple-symmetry GFM framework.

**Scaling Laws in Graph Learning.** Scaling laws have played a central role in understanding and developing foundation models in domains such as natural language and computer vision, where performance is known to improve predictably with increased model and dataset size. Kaplan et al. [19] introduced the original scaling laws for language models, demonstrating that performance improves with larger models and more data. Hoffmann et al. [15] later refined this trend, showing that the models in [19] were under-trained and that slightly better performance could be achieved for smaller models. Similar scaling behaviors have been observed in vision, notably in Vision Transformers [8] and large-scale contrastive models such as CLIP [31].

In graph learning, scaling laws remain relatively underexplored. A first investigation is Liu et al. [23], which studies classical GNNs on graph-level classification tasks under varying model and dataset scales. However, this study is limited to traditional, task-specific architectures and does not address GFMs, which aim to generalize across diverse graphs, label spaces, or feature domains. It remains unclear whether scaling trends observed in narrowly-scoped GNNs also hold in this more challenging and general setting. Our framework is the first to demonstrate scaling-law behavior in GFMs, where performance improves consistently as the number of diverse training graphs increases (see Section 6.2).

## 4 Universal Approximation of Functions With Triple-Symmetries

To derive a GFM that generalizes across arbitrary graphs and across feature and label sets, we first define universal approximators of multiset functions that respect *triple symmetries*: permutations of nodes, features, and labels. In Section 4.1, we characterize all linear maps that are equivariant to these permutations and. By composing the linear layers with nonlinearities, we obtain a symmetry-preserving networks in Section 4.2 with a universal approximation property. We later use these functions in Section 5 on each node's neighborhood to obtain a message-passing scheme for graph data.

### 4.1 Characterizing Equivariant Linear Layers

To define multilayer neural networks that respect triple-symmetries, we first characterize the linear layers that are equivariant to node-, label- and feature- permutations ("Equivariance Everywhere All At Once"). Our result builds on the rich literature on learning on sets [50, 38, 26] and extends these techniques to our symmetry groups. To better locate our result, we first discuss existing characterizations for DeepSets [50] and for Deep Sets for Symmetric elements.

**DeepSets.** A seminal result for learning on sets is presented by Zaheer et al. [50], who characterize all linear maps of the form $T : \mathbb{R}^{N \times F_1} \to \mathbb{R}^{N \times F_2}$ that are $S_N$-equivariant, as

$$T(\boldsymbol{X}) = \mathbf{1}_{N,N} \boldsymbol{X} \boldsymbol{\Lambda}_1 + \boldsymbol{X} \boldsymbol{\Lambda}_2, \text{ where } \boldsymbol{\Lambda}_1, \boldsymbol{\Lambda}_2 \in \mathbb{R}^{F_1 \times F_2}.$$

**Deep sets for symmetric elements (DSS).** The characterization of $S_N$-equivariant functions introduced by Zaheer et al. [50] is extended by Maron et al. [26] to include an additional symmetry across the feature dimension. Specifically, they characterize all $S_N \times H$-equivariant linear maps, where $H \subseteq S_F$ is a subgroup of the feature permutation group $S_F$. Our interests lies in the simpler and more common case in which $H = S_F$, corresponding to the full feature permutation group. In this setting, Maron et al. [26] characterize all linear maps of the form $T : \mathbb{R}^{N \times F \times K_1} \rightarrow \mathbb{R}^{N \times F \times K_2}$ that are $(S_N \times S_F)$-equivariant; that is, for each $k_2 \in [K_2]$, the corresponding output $T(\boldsymbol{X})_{:,:,k_2} \in \mathbb{R}^{N \times F}$ is

$$T(\boldsymbol{X})_{:,:,k_2} = \boldsymbol{1}_{N,N} \boldsymbol{X}^{(1)}_{:,:,k_2} \boldsymbol{1}_{F,F} + \boldsymbol{1}_{N,N} \boldsymbol{X}^{(2)}_{:,:,k_2} + \boldsymbol{X}^{(3)}_{k_2} \boldsymbol{1}_{F,F} + \boldsymbol{X}^{(4)}_{k_2}, \qquad (1)$$

where $\forall i \in [4], \boldsymbol{X}^{(i)}_{:,:,k_2} = \sum_{k_1}^{K_1} \boldsymbol{X}_{:,:,k_1} \Lambda^{(i)}_{k_1,k_2}$ and $\Lambda^{(i)} \in \mathbb{R}^{K_1 \times K_2}$. Note that symmetry across both axes of the feature matrix requires lifting the 2D function inputs used in DeepSets to 3D tensors in DSS and in our framework.

**The limitations of semi-supervised node-classification.** Most existing GNNs and set-based architectures, such as DeepSets or DSS, operate within the classical semi-supervised learning framework: each node in the input graph is associated with a feature vector, but only a subset of nodes have known labels during training. In this setup, the model learns to map node embeddings to the specific label set observed during training. Consequently, the model is inherently tied to this fixed label set and cannot generalize effectively to new or varying label sets encountered at test time.

**The label inpainting task.** To enable generalization across label sets, we move beyond the classical semi-supervised regime and its dependence on the fixed set of training labels and feature, and fixed graph. In *label inpainting*, each datapoint is a graph $G = (\mathcal{V}, \mathcal{E}, \boldsymbol{X}, \boldsymbol{Y})$ with node features $\boldsymbol{X} \in \mathbb{R}^{N \times F}$, and labels in $[C]$. The full graph and node features are given as inputs, along with the labels of a partial set of nodes $L \subset [N]$. The goal is to inpaint (predict) the labels of the remaining masked nodes $R = [N] \setminus L$. We consider a supervised setting in which the dataset consists of many graphs, each with a different set of revealed label nodes. Each training graph has the ground-truth labels of the masked nodes available for supervision during training, but masked labels are hidden in test graphs. Note that difference data points, i.e. graphs at training or inference, may differ in their node features, label sets, and topology.

**Our framework for triple-symmetry.** To build a layered architecture consisting of equivariant linear layers and nonlinearities, we first find all linear alyers that respect triple-symmetries. The following proposition provides a complete characterization of the linear transformations $T = (T_1, T_2) : \mathbb{R}^{N \times (F+C)} \rightarrow \mathbb{R}^{N \times (F+C)}$ that are $(S_N \times S_F \times S_C)$-equivariant. See Section C for more details.

**Proposition 4.1.** *A linear function of the form* $T = (T_1, T_2): \mathbb{R}^{N \times F \times K_1} \times \mathbb{R}^{N \times C \times K_1} \rightarrow \mathbb{R}^{N \times F \times K_2} \times \mathbb{R}^{N \times C \times K_2}$ *is* $(S_N \times S_F \times S_C)$-*equivariant if and only if there exist* $\boldsymbol{\Lambda}^{(1)}, \ldots, \boldsymbol{\Lambda}^{(12)} \in \mathbb{R}^{K_1 \times K_2}$ *such that for every* $\boldsymbol{X} \in \mathbb{R}^{N \times F \times K_1}, \boldsymbol{Y} \in \mathbb{R}^{N \times C \times K_1}$ *and* $k_2 \in [K_2]$, *the corresponding outputs* $T_1(\boldsymbol{X}, \boldsymbol{Y}) \in \mathbb{R}^{N \times F \times K_2}$ *and* $T_2(\boldsymbol{X}, \boldsymbol{Y}) \in \mathbb{R}^{N \times C \times K_2}$ *are given by*

$$
\begin{aligned}
T_1(\boldsymbol{X}, \boldsymbol{Y})_{:,:,k_2} &= \left( \boldsymbol{1}_{N,N} \boldsymbol{X}^{(1)}_{:,:,k_2} + \boldsymbol{X}^{(2)}_{:,:,k_2} \right) \boldsymbol{1}_{F,F} + \boldsymbol{1}_{N,N} \boldsymbol{X}^{(3)}_{:,:,k_2} + \boldsymbol{X}^{(4)}_{:,:,k_2} \\
&\quad + \left( \boldsymbol{1}_{N,N} \boldsymbol{Y}^{(5)}_{:,:,k_2} + \boldsymbol{Y}^{(6)}_{:,:,k_2} \right) \boldsymbol{1}_{C,F}, \\
T_2(\boldsymbol{X}, \boldsymbol{Y})_{:,:,k_2} &= \left( \boldsymbol{1}_{N,N} \boldsymbol{Y}^{(1)}_{:,:,k_2} + \boldsymbol{Y}^{(2)}_{:,:,k_2} \right) \boldsymbol{1}_{C,C} + \boldsymbol{1}_{N,N} \boldsymbol{Y}^{(3)}_{:,:,k_2} + \boldsymbol{Y}^{(4)}_{:,:,k_2} \\
&\quad + \left( \boldsymbol{1}_{N,N} \boldsymbol{X}^{(5)}_{:,:,k_2} + \boldsymbol{X}^{(6)}_{:,:,k_2} \right) \boldsymbol{1}_{F,C},
\end{aligned}
\qquad (2)
$$

*where for every* $i \in [6]$, *we have* $\boldsymbol{X}^{(i)}_{:,:,k_2} = \sum_{k_1=0}^{K_1} \boldsymbol{X}_{:,:,k_1} \Lambda^{(i)}_{k_1,k_2}$ *and* $\boldsymbol{Y}^{(i)}_{:,:,k_2} = \sum_{k_1=0}^{K_1} \boldsymbol{Y}_{:,:,k_1} \Lambda^{(i+6)}_{k_2,k_1}$.

## 4.2 A Universal Approximation Theorem for Functions with Triple-Symmetries

In this section, we introduce *equivariant triple symmetry networks (TSNets)*, based on the $(S_N \times S_F \times S_C)$-equivariant layers in Section 4.1. We prove universality for TSNet$_{\text{Eq}}$, and then derive universality for the *triple symmetry networks (TSNet)* – an $(S_N \times S_C)$-equivariant and $S_F$-invariant architecture that serves as the core building block of our GFM.

**Triple-symmetry networks for multi-sets (TSNets).** We define an equivaraint triple symmetry network (TSNet$_{\text{Eq}}$) $F_{\text{Eq}} : \mathbb{R}^{N \times (F+C) \times 1} \rightarrow \mathbb{R}^{N \times (F+C) \times 1}$ as

$$F_{\text{Eq}} = T^{(L)} \circ \sigma \circ T^{(L-1)} \circ \sigma \circ \cdots \circ T^{(1)},$$

where each $T^{(\ell)} : \mathbb{R}^{N \times (F+C) \times K^{(\ell)}} \to \mathbb{R}^{N \times (F+C) \times K^{(\ell+1)}}$ is an $(S_N \times S_F \times S_C)$-equivariant linear layer, and $\sigma$ is a non-linearity.

We then define the space of *TSNet label inpainters*, TSNet$_{\text{LI}}$ as follows. Define the *label projection map* $\Pi_C$ by $\Pi_C(\boldsymbol{X}, \boldsymbol{Y}) = \boldsymbol{Y}$ for all $(\boldsymbol{X}, \boldsymbol{Y}) \in \mathbb{R}^{N \times F \times K^{(L)}} \times \mathbb{R}^{N \times C \times K^{(L)}}$. A triple symmetry network $F : \mathbb{R}^{N \times (F+C) \times 1} \to \mathbb{R}^{N \times C \times 1}$ in TSNet$_{\text{LI}}$ if it is of the form $F = \Pi_C \circ F_{\text{Eq}}$, where $F_{\text{Eq}} : \mathbb{R}^{N \times (F+C) \times 1} \to \mathbb{R}^{N \times (F+C) \times 1}$ is a TSNet$_{\text{Eq}}$.

Observe that $\Pi_C$ is $S_F$-invariant and each intermediate linear layer $T^{(\ell)}$ is $(S_N \times S_F \times S_C)$-equivariant by construction. Since equivariance and invariance are preserved under compositions, it follows directly that the resulting network $F$ is both $(S_N \times S_C)$-equivariant and $S_F$-invariant, as required. We set $K^{(0)} = K^{(L)} = 1$ so that the network can accept the initial feature and label matrices and outputs predictions with the same shape as the ground-truth labels.

We now state our main result: TSNets are not only symmetry-preserving but are also universal approximators, i.e., capable of approximating any continuous function that respects the triple-symmetries. More accurately, this guarantee holds over any symmetric compact domain which is disjoint from some small set, called the exclusion set $\mathcal{E}$ (Definition D.20).

**Lemma 4.2.** *Let $\mathcal{K} \subset \mathbb{R}^{N \times (F+C)}$ be a compact domain such that $\mathcal{K} = \cup_{g \in S_N \times S_F \times S_C} g\mathcal{K}$ and $\mathcal{K} \cap \mathcal{E} = \emptyset$, where $\mathcal{E} \subset \mathbb{R}^{N \times (F+C)}$ is the exclusion set corresponding to $\mathbb{R}^{N \times (F+C)}$ (Definition D.20). Then, TSNet$_{\text{Eq}}$ is a universal approximator (i.e. in $L_\infty$) of continuous $\mathcal{K} \to \mathbb{R}^{N \times (F+C)}$ functions that are $(S_N \times S_F \times S_C)$-equivariant.*

**Theorem 4.3.** *Let $K \subset \mathbb{R}^{N \times (F+C)}$ be a compact domain such that $\mathcal{K} = \cup_{g \in S_N \times S_F \times S_C} gK$ and $\mathcal{K} \cap \mathcal{E} = \emptyset$, where $\mathcal{E} \subset \mathbb{R}^{N \times (F+C)}$ is the exclusion set corresponding to $\mathbb{R}^{N \times (F+C)}$ (Definition D.20). Then, TSNet$_{\text{LI}}$ is a universal approximator of continuous $\mathcal{K} \to \mathbb{R}^{N \times C}$ functions that are $(S_N \times S_C)$-equivariant and $S_F$-invariant.*

**Relations to the literature.** Our result is closely related to Theorem 3 of [26], which establishes a universality result for $(S_N \times H)$-equivariant networks, where $H \subseteq S_F$. Specifically, the theorem states that for any subgroup $H \subseteq S_F$ and any compact domain $\mathcal{K} \subset \mathbb{R}^{N \times F}$ that is closed under the action of $(S_N \times H)$ and avoids the exception set $\mathcal{E}'$ (defined in Definition E.2). If $H$-equivariant networks are universal for $H$-equivariant functions. Then, $(S_N \times H)$-equivariant networks are universal approximators (in the $L_\infty$ norm) for continuous $(S_N \times H)$-equivariant functions defined on $\mathcal{K}$. However, their proof contains a subtle but important flaw. They introduce an $H$-equivariant polynomial $p_1 : \mathbb{R}^{N \times F} \to \mathbb{R}^F$ and write "If we fix $\boldsymbol{X}_2, ..., \boldsymbol{X}_N$, then $p_1(\boldsymbol{X}_1, \dots, \boldsymbol{X}_N)$ is an $H$-equivariant polynomial in $\boldsymbol{X}_1$", which does not hold generally. By definition, $H$-equivariance of $p_1$ requires the group action to be applied consistently across all arguments: applying the action to only one row while fixing the others violates equivariance. We refer the reader to Section E for a detailed discussion on this issue, why it invalidates the general universality claim for arbitrary subgroups $H \leq S_F$ and how, by leveraging techniques from [38], the claim can still be upheld in the special case $H = S_F$ – a key insight that allows us to show universality for TSNets in Theorem 4.3.

## 5 A Recipe for Graph Foundation Models for Node Property Prediction

In this section, we build on the results from Section 4.1 to derive a practical architecture for node property inpainting (see Figure 2). Specifically, we construct a learnable layer that is equivariant to the symmetry group $(S_N \times S_F \times S_C)$ based on any aggregation scheme. In the graph setting, this elevates the MPNN to a symmetry-aware foundation model that generalizes across diverse feature sets and labels, unlike traditional GNNs.

We start from Equation (2), which can be interpreted as passing messages by applying *sum aggregation* along the feature, label, and node axes. We generalize these sums to arbitrary permutation-invariant operators (e.g., mean, max, or attention style aggregations). Setting the operators to sums recovers the linear structure of Equation (2).

**Triple-symmetry graph neural networks (TS-GNNs).** Consider inputs of the form $(\boldsymbol{X}, \boldsymbol{Y}, \boldsymbol{A}) \in \mathbb{R}^{N \times (F+C) \times 1} \times [0, 1]^{N \times N}$, where $\boldsymbol{X} \in \mathbb{R}^{N \times F}$ is a feature matrix, $\boldsymbol{Y} \in \mathbb{R}^{N \times C}$ is a label matrix, and $\boldsymbol{A} \in [0, 1]^{N \times N}$ is the graph adjacency matrix. Formally, we define a triple-symmetry graph

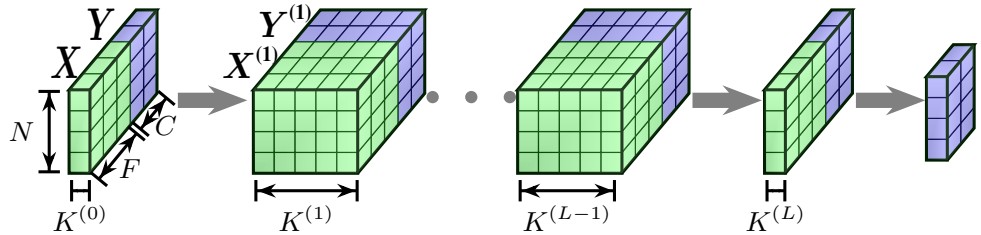

Figure 2: An illustration of the feature and label embeddings across the layers of our triple-symmetric graph neural network architecture. The architecture is composed of feature-, label- and node-equivariant aggregation layers and a final feature-invariant projection layer.

neural network $F : \mathbb{R}^{N \times (F+C) \times 1} \times [0,1]^{N \times N} \to \mathbb{R}^{N \times C \times 1}$, as

$$F = \pi \circ T^{(L)} \circ \sigma \circ T^{(L-1)} \circ \sigma \circ \cdots \circ T^{(1)},$$

$$T^{(\ell)} : \mathbb{R}^{N \times (F+C) \times K^{(\ell)}} \times [0,1]^{N \times N} \to \mathbb{R}^{N \times (F+C) \times K^{(\ell+1)}} \times [0,1]^{N \times N},$$

where $\sigma$ is a non-linearity (e.g., ReLU) and each $T^{(\ell)}$ is an $(S_N \times S_F \times S_C)$-equivariant layer with a general aggregation along the feature, label and node axes, that acts jointly on the node features and node labels, while also using the adjacency matrix. Each $T^{(\ell)}$ updates the feature $\boldsymbol{X}_v^{(\ell)} \in \mathbb{R}^{K^{(\ell)} \times F}$ and label $\boldsymbol{Y}_v^{(\ell)} \in \mathbb{R}^{K^{(\ell)} \times C}$ representations of node $v \in \mathcal{V}$ at layer $\ell$ as

$$\begin{aligned}
\boldsymbol{X}_v^{(\ell+1)} = {}& \psi\big(\boldsymbol{W}_1^{(\ell)} \boldsymbol{X}_v^{(\ell)}, \{\!\{\boldsymbol{W}_2^{(\ell)} \boldsymbol{X}_u^{(\ell)} \mid u \in \mathcal{N}_v\}\!\}\big) \\
& + \psi\big(\boldsymbol{W}_3^{(\ell)} \nu(\boldsymbol{X}_v^{(\ell)}) \mathbf{1}_{1,F}, \{\!\{\boldsymbol{W}_4^{(\ell)} \nu(\boldsymbol{X}_u^{(\ell)}) \mathbf{1}_{1,F} \mid u \in \mathcal{N}_v\}\!\}\big) \\
& + \psi\big(\boldsymbol{W}_5^{(\ell)} \nu(\boldsymbol{Y}_v^{(\ell)}) \mathbf{1}_{1,F}, \{\!\{\boldsymbol{Y}_6^{(\ell)} \nu(\boldsymbol{Y}_u^{(\ell)}) \mathbf{1}_{1,F} \mid u \in \mathcal{N}_v\}\!\}\big) + \Theta_F, \\
\boldsymbol{Y}_v^{(\ell+1)} = {}& \psi\big(\boldsymbol{W}_7^{(\ell)} \boldsymbol{Y}_v^{(\ell)}, \{\!\{\boldsymbol{W}_8^{(\ell)} \boldsymbol{Y}_u^{(\ell)} \mid u \in \mathcal{N}_v\}\!\}\big) \\
& + \psi\big(\boldsymbol{W}_9^{(\ell)} \nu(\boldsymbol{Y}_v^{(\ell)}) \mathbf{1}_{1,C}, \{\!\{\boldsymbol{W}_{10}^{(\ell)} \nu(\boldsymbol{Y}_u^{(\ell)}) \mathbf{1}_{1,C} \mid u \in \mathcal{N}_v\}\!\}\big) \\
& + \psi\big(\boldsymbol{W}_{11}^{(\ell)} \nu(\boldsymbol{X}_v^{(\ell)}) \mathbf{1}_{1,C}, \{\!\{\boldsymbol{W}_{12}^{(\ell)} \nu(\boldsymbol{X}_u^{(\ell)}) \mathbf{1}_{1,C} \mid u \in \mathcal{N}_v\}\!\}\big) + \Theta_C,
\end{aligned}$$

where $\psi$ is the node aggregation function, $\nu$ is the label/feature aggregation function, $\boldsymbol{W}_i^{(\ell)} \in \mathbb{R}^{K^{(\ell+1)} \times K^{(\ell)}}$, $i \in [12]$, are learnable weight matrices (replacing the coefficients $\boldsymbol{\Lambda}_i$), and the terms $\Theta_F$ and $\Theta_C$ are $(S_N \times S_F \times S_C)$-equivariant feature-label mixing terms (detailed below). Finally, the label projection map $\Pi_C$ is applied to obtain node-wise predictions. Similarly to TSNets, we also set $K^{(0)} = K^{(L)} = 1$. It is easy to see that the architecture is $(S_N \times S_C)$-equivariant and $S_F$-invariant, as desired. In this formulation, the aggregations can be based on any MPNN, e.g., GAT, GraphSAGE, or GCNII [42, 14, 5].

**How to mix features and labels?** A key consideration in designing graph foundation models is determining the degree of interaction between feature and label embeddings. In the extreme, if we do not impose any such interactions then the model's predictions would be independent of the input features, leading to a degradation in model performance. Effective information flow between feature and label representations is thus essential for building strong graph foundation models. This raises a central design question: *To what extent should feature embeddings and label embeddings interact?*

In TS-GNNs, we initially perform mixing by applying permutation-invariant pooling operations (e.g., feature-wise or label-wise sum pooling) and appending the pooled representations to the opposing modality (feature-to-label and label-to-feature). While such operations are theoretically sufficient for our universal approximation result (Section 4.2), they often suffer from practical limitations, because all feature or label information is compressed into a single vector representation, creating a representational bottleneck .

To overcome this issue and enable richer feature-label mixing, we introduce an additional mixing operation based on least-squares solutions, which showed strong empirical results [52]. Specifically, we solve the following optimization problems:

$$\boldsymbol{T}_F = \underset{\boldsymbol{T}_F \in \mathbb{R}^{F \times C}}{\arg\min} \|\boldsymbol{Y} - \boldsymbol{A}\boldsymbol{X}\boldsymbol{T}_F\|_2 \quad \text{and} \quad \boldsymbol{T}_C = \underset{\boldsymbol{T}_C \in \mathbb{R}^{C \times F}}{\arg\min} \|\boldsymbol{X} - \boldsymbol{A}\boldsymbol{Y}\boldsymbol{T}_C\|_2$$

where $\boldsymbol{A}$ is a random-walk normalized adjacency matrix. These projections transform features into the label space and labels into the feature space, respectively. We incorporate these least-squares transformations into each layer as follows:

$$\Theta_F = \psi\left(\boldsymbol{W}_{13}^{(\ell)}\boldsymbol{Y}_v^{(\ell)}\boldsymbol{T}_C, \{\!\!\{\boldsymbol{W}_{14}^{(\ell)}\boldsymbol{Y}_u^{(\ell)}\boldsymbol{T}_C \mid u \in \mathcal{N}_v\}\!\!\}\right),$$

$$\Theta_C = \psi\left(\boldsymbol{W}_{15}^{(\ell)}\boldsymbol{X}_v^{(\ell)}\boldsymbol{T}_F, \{\!\!\{\boldsymbol{W}_{16}^{(\ell)}\boldsymbol{X}_u^{(\ell)}\boldsymbol{T}_F \mid u \in \mathcal{N}_v\}\!\!\}\right),$$

where $\boldsymbol{W}_{13}^{(\ell)}, \boldsymbol{W}_{14}^{(\ell)}, \boldsymbol{W}_{15}^{(\ell)}, \boldsymbol{W}_{16}^{(\ell)} \in \mathbb{R}^{K^{(\ell+1)} \times K^{(\ell)}}$ are learnable weight matrices.

These terms share the same objective as our original mixing terms, but crucially avoid collapsing over feature or label dimensions. Instead, they introduce structured, linear mappings across modalities via $\boldsymbol{T}_F$ and $\boldsymbol{T}_C$, enabling localized mixing. Importantly, these least-squares transformations are linear and are equivariant to node-, feature-, and label- permutations, allowing us to retain the triple-symmetry property at every layer. This design also lays a foundation for future extensions involving more expressive mixing mechanisms.

**How to incorporate bias terms?** Another important design choice involves the integration of bias terms into a model that needs to generalize to varying feature scales. Even after row-wise or global $L_1/L_2$ normalization, node features can differ both within a graph and across graphs. A bias vector learned on the set of training graphs may be much larger or much smaller than the features which are going to be observed at test time. This is clearly problematic for generalization, as the bias term may for instance dominate the actual input features in magnitude, leading to them being largely ignored. To avoid this, one option is to completely avoid using bias terms in the layers, but this is too restrictive. In our architecture, we solve the least-squares problem including an added bias term, and the bias terms obtained, are injected only through the feature-to-label and label-to-feature mixers ($\Theta_F$ and $\Theta_C$). Such biases are learned relative to the current feature scale, and thus remain scale-aware and adapt naturally to new graphs, offering better generalization.

## 6 Empirical Evaluation

We empirically validate the capabilities of TS-GNNs, addressing the core research questions:

**Q1** Can the proposed graph foundation models effectively generalize to unseen graphs with varying structures, feature configurations, and label semantics without retraining?

**Q2** Does the zero-shot generalization ability of the proposed graph foundation models improve as the number of training graphs increases?

In Section G, we additionally investigate supplementary questions:

**Q3** How sensitive are TS-GNNs to the choice of pretraining dataset in low-data regimes?

**Q4** What is the impact of least-squares mixing and the bias term on TS-GNNs' performance?

**Q5** Is triple-symmetry necessary for graph foundation models?

To address these questions, we perform experiments across 29 real-world node classification datasets, covering diverse domains, graph sizes, and feature representations. Note that we focus on the node classification setting due to the lack of node-regression datasets, and refer the reader to the appropriate discussion in Section H. We train TS-GNNs by randomly masking the labels of 50% of the labeled nodes at each epoch and optimizing the model to predict the masked labels from the node features and the remaining visible labels. We optimize all models using the Adam optimizer, with detailed hyperparameter settings provided in Section J. Experiments are executed on a single NVIDIA L40 GPU, and publicly accessible at: `https://github.com/benfinkelshtein/EquivarianceEverywhere`.

**Datasets.** We use an ensemble of 29 node classification datasets and their respective official splits. Specifically, we use roman-empire, amazon-ratings, minesweeper, questions and tolokers from [30], cora, citeseer and pubmed from [49], chameleon and squirrel from [36], cornell, wisconsin, texas and actor from [29], full-dblp and full-cora from [3], wiki-attr and blogcatalog from [48], wiki-cs from [27], co-cs, co-physics, computers and photo from [39], brazil, usa and europe from [34], last-fm-asia and deezer from [35], and arxiv from [17]. As full-Cora, co-cs, and co-physics, have an exceptionally large feature dimension, we first reduce their feature dimensionality with PCA to $2048$ components before training. Lastly, we apply $L_2$ normalization to each node's feature vector.

Table 1: Test accuracy of the baselines MeanGNN and GAT and models GraphAny and our GFM variants, all trained on cora. Top three models are colored by First, Second, Third.

| | End-to-end | | Zero-shot | | |
|---|---|---|---|---|---|
| | MeanGNN | GAT | GraphAny | TS-Mean | TS-GAT |
| actor | $32.03 \pm 0.29$ | $32.59 \pm 0.83$ | $27.54 \pm 0.20$ | $28.09 \pm 0.93$ | $28.80 \pm 2.12$ |
| amazon-ratings | $40.74 \pm 0.13$ | $40.63 \pm 0.66$ | $42.80 \pm 0.09$ | $42.27 \pm 1.40$ | $41.92 \pm 0.47$ |
| arxiv | $50.94 \pm 0.38$ | $57.93 \pm 3.44$ | $58.85 \pm 0.03$ | $56.33 \pm 2.58$ | $53.13 \pm 2.04$ |
| blogcatalog | $84.48 \pm 0.74$ | $78.20 \pm 7.23$ | $71.54 \pm 3.04$ | $76.30 \pm 2.92$ | $78.44 \pm 5.18$ |
| brazil | $32.31 \pm 7.50$ | $35.38 \pm 4.21$ | $33.84 \pm 15.65$ | $39.23 \pm 5.70$ | $36.92 \pm 10.03$ |
| chameleon | $61.75 \pm 0.94$ | $58.46 \pm 6.23$ | $63.64 \pm 1.48$ | $60.83 \pm 5.41$ | $54.65 \pm 4.58$ |
| citeseer | $65.06 \pm 1.30$ | $63.92 \pm 0.84$ | $68.88 \pm 0.10$ | $68.66 \pm 0.19$ | $68.70 \pm 0.48$ |
| co-cs | $80.87 \pm 0.69$ | $82.28 \pm 0.86$ | $90.06 \pm 0.80$ | $90.92 \pm 0.47$ | $91.13 \pm 0.43$ |
| co-physics | $79.05 \pm 1.13$ | $85.92 \pm 1.10$ | $91.85 \pm 0.34$ | $92.61 \pm 0.61$ | $92.23 \pm 0.61$ |
| computers | $73.88 \pm 0.88$ | $70.94 \pm 3.40$ | $82.79 \pm 1.13$ | $81.37 \pm 1.25$ | $80.23 \pm 2.44$ |
| cornell | $63.78 \pm 1.48$ | $69.73 \pm 2.26$ | $63.24 \pm 1.32$ | $68.65 \pm 2.42$ | $71.35 \pm 4.52$ |
| deezer | $54.91 \pm 1.81$ | $55.22 \pm 2.33$ | $51.82 \pm 2.49$ | $52.31 \pm 2.52$ | $52.88 \pm 2.86$ |
| europe | $39.12 \pm 6.44$ | $39.00 \pm 4.30$ | $41.25 \pm 7.25$ | $35.88 \pm 6.91$ | $38.38 \pm 6.32$ |
| full-DBLP | $65.01 \pm 2.40$ | $67.34 \pm 2.75$ | $71.48 \pm 1.44$ | $66.42 \pm 3.65$ | $64.30 \pm 3.75$ |
| full-cora | $55.85 \pm 1.04$ | $59.95 \pm 0.88$ | $51.18 \pm 0.78$ | $53.58 \pm 0.73$ | $52.82 \pm 0.90$ |
| last-fm-asia | $77.23 \pm 1.01$ | $72.65 \pm 0.48$ | $81.14 \pm 0.42$ | $78.03 \pm 1.02$ | $76.60 \pm 1.44$ |
| minesweeper | $84.06 \pm 0.18$ | $84.15 \pm 0.24$ | $80.46 \pm 0.11$ | $80.68 \pm 0.38$ | $80.72 \pm 0.78$ |
| photo | $88.95 \pm 1.08$ | $80.78 \pm 3.59$ | $89.91 \pm 0.88$ | $90.18 \pm 1.30$ | $89.66 \pm 1.23$ |
| pubmed | $74.56 \pm 0.13$ | $75.12 \pm 0.89$ | $76.46 \pm 0.08$ | $74.98 \pm 0.56$ | $75.46 \pm 0.86$ |
| questions | $97.16 \pm 0.06$ | $97.13 \pm 0.05$ | $97.07 \pm 0.03$ | $97.02 \pm 0.01$ | $97.02 \pm 0.01$ |
| roman-empire | $69.37 \pm 0.66$ | $69.80 \pm 4.18$ | $63.34 \pm 0.58$ | $66.36 \pm 1.02$ | $67.76 \pm 0.60$ |
| squirrel | $43.32 \pm 0.66$ | $38.16 \pm 1.04$ | $49.74 \pm 0.47$ | $41.81 \pm 0.80$ | $38.29 \pm 4.29$ |
| texas | $76.76 \pm 1.48$ | $81.62 \pm 6.45$ | $71.35 \pm 2.16$ | $73.51 \pm 4.01$ | $74.05 \pm 4.10$ |
| tolokers | $78.59 \pm 0.66$ | $78.22 \pm 0.37$ | $78.20 \pm 0.03$ | $78.12 \pm 0.09$ | $78.01 \pm 0.23$ |
| usa | $43.93 \pm 1.16$ | $43.03 \pm 2.08$ | $43.35 \pm 1.62$ | $42.34 \pm 2.12$ | $41.69 \pm 2.59$ |
| wiki-attr | $74.23 \pm 0.89$ | $68.91 \pm 9.50$ | $60.27 \pm 3.06$ | $69.89 \pm 1.31$ | $72.53 \pm 1.59$ |
| wiki-cs | $71.97 \pm 1.70$ | $74.99 \pm 0.59$ | $74.11 \pm 0.60$ | $74.16 \pm 2.07$ | $73.24 \pm 2.49$ |
| wisconsin | $74.12 \pm 12.20$ | $73.33 \pm 8.27$ | $59.61 \pm 5.77$ | $61.18 \pm 11.38$ | $65.88 \pm 9.86$ |
| Average (28 graphs) | $65.50 \pm 1.75$ | $65.55 \pm 2.82$ | $65.56 \pm 1.86$ | $65.78 \pm 2.28$ | $65.60 \pm 2.74$ |

## 6.1 Do TS-GNNs Generalize to Unseen Graphs?

**Setup.** To evaluate generalization (**Q1**), we design an experiment to assess how well each model performs after being trained on a single graph (cora) – measuring how much transferable knowledge can be extracted from that graph. Our setup closely follows the setup introduced in GraphAny [52]. We experiment with two widely-used baseline GNN architectures: MeanGNN and GAT [42]. Since these models are inherently non-transferable across datasets, we train and evaluate each baseline in an end-to-end fashion on every dataset. Further to this, we provide zero-shot experiments with TS-Mean and TS-GAT, which are TS-GNNs using MeanGNN and GAT, respectively (as their aggregation function). Additional GNN variants, such as TS-GCNII, are evaluated in Section G.4. We also experiment with the only other node-level graph foundation model GraphAny [52], which provides a strong zero-shot prediction baseline. In the zero-shot setup, we train each model TS-Mean, TS-GAT, and GraphAny on the cora dataset and evaluate their performance on the remaining 28 datasets. All reported results reflect the mean accuracy and standard deviation over five random seeds.

**Results.** Table 1 shows that both TS-Mean and TS-GAT consistently outperform their respective baselines, despite being trained in an end-to-end manner. This highlights the generalization capabilities of our proposed framework and its compatibility with varying GNN architectures. Furthermore, both TS-Mean and TS-GAT outperform GraphAny on average across 28 datasets under its own evaluation protocol, further underscoring the strong generalization capabilities of our framework. These results directly address our initial research question (**Q1**) – our proposed framework, that is also theoretically grounded, successfully generalizes across graphs, features, and label distributions without retraining.

## 6.2 Does More Training Data Improve Generalization?

**Setup.** To assess the impact of the amount of training data on zero-shot generalization (**Q2**), we train TS-Mean and GraphAny on increasingly larger subsets of a held-out training pool. We reserve 9 representative graphs for training and construct training subsets of sizes 1, 3, 5, 7, and 9, where each larger subset includes all datasets from the smaller trainsets. The remaining 20 datasets are used for zero-shot evaluation. For each subset size, we report mean zero-shot accuracy, averaged over five random seeds. Detailed per-dataset results are provided in Section G.5.

**Results.** Figure 3 shows that the zero-shot accuracy of TS-Mean improves steadily as more training graphs are introduced. This behavior aligns with the expected characteristics of a graph foundation model – the ability to benefit from increased data diversity, directly addressing **Q2**. In contrast, GraphAny's performance remains unchanged as more training graphs are added. This

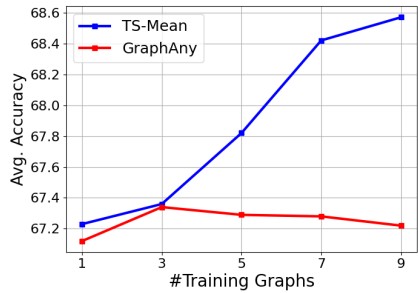

Figure 3: Average zero-shot accuracy of TS-Mean and GraphAny across 20 datasets as a function of the number of training graphs.

counterintuitive result suggests that GraphAny is not well-suited for the foundation model setting, and may instead be better tailored to scenarios with limited training data. Interestingly, both models perform comparably when trained on three graphs or less, indicating that TS-Mean is competitive in the low-data regime. However, as the training set grows, TS-Mean increasingly outperforms GraphAny, highlighting the benefits of performance that scales with training data.

While we do not claim to establish a formal "scaling law," this offers, to our knowledge, the first empirical indication that a GFM improves with more training data – a key property of foundation models in language and vision. These findings position TS-GNN as a strong and, at present, the only viable candidate for node-level graph foundation modeling.

## 7 Conclusion

This work introduces a principled, theoretically-grounded framework for designing GFMs for node-level classification and regression tasks. At the heart of our approach lies the triple-symmetry criterion: a GFM must be equivariant to node and label permutations and invariant to permutations of input features. These symmetries capture the key inductive bias required for generalization across graphs with varying structures, feature spaces, and labels.

Building on this, we propose a triple-symmetry neural network which is proven to bea universal approximator of triple-symmetry respecting functions. By applying TSNets to the multisets of features induced by the local graph neighborhoods, and by extending TSNet to general aggregation schemes, we derived triple-symmetry graph neural networks: a flexible framework that extends any standard GNN into a triple-symmetry-aware GFM capable of zero-shot generalization.

Our empirical study validates the effectiveness of this recipe. We train the GFM variations of the well-used models MeanGNN and GAT on a single dataset, achieving robust zero-shot generalization across 29 diverse node classification benchmarks, outperforming not only their original counterparts, but also the prior state-of-the-art GraphAny. Furthermore, we present the first empirical evidence that a graph foundation model can systematically benefit from increasing training data: TS-Mean exhibits steadily improving zero-shot performance as more training graphs are introduced, while GraphAny fails to do so. These findings underscore the strong generalization capabilities of our principled recipe.

Altogether, our work provides a unified theoretical and practical foundation for constructing generalizable GNNs. Future work could explore extending this recipe to edge-level or graph-level tasks as our recipe is currently limited to the node-level case. As the landscape of graph learning moves toward more flexible and general architectures, our work offers a compelling blueprint for building robust, transferable and principled models.

## Acknowledgments

We thank Yonatan Sverdlov for discussions regarding related literature and its theoretical relevance to this work. This research was supported by a grant from the United States-Israel Binational Science Foundation (BSF), Jerusalem, Israel, and the United States National Science Foundation (NSF), (NSF-BSF, grant No. 2024660) , and by the Israel Science Foundation (ISF grant No. 1937/23). BF is funded by the Clarendon scholarship. MB is supported by EPSRC Turing AI World-Leading Research Fellowship No. EP/X040062/1 and EPSRC AI Hub on Mathematical Foundations of Intelligence: An "Erlangen Programme" for AI No. EP/Y028872/1.

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

# A  Groups, Permutations, and Symmetries

## A.1  A Primer on Representation Theory

We denote the identity element of any group by $e$. The notation $H \leq G$ indicates that $H$ is a subgroup of the group $G$. Denote by $\mathrm{GL}(V)$ the general linear group of invertible linear transformations on the vector space $V$. Denote by $\mathrm{id}_V$ the identity map on a vector space $V$.

Representation theory studies how abstract groups can be understood by expressing their elements as linear transformations on vector spaces. This is formalized by a group homomorphism from the group into the general linear group, called a representation. We begin by recalling the notion of a group homomorphism, which is a structure-preserving map between groups.

**Definition A.1** (Group Homomorphism). *Let $G$ and $H$ be groups. A map $\phi : G \to H$ is called a* group homomorphism *if it satisfies*

$$\forall g_1, g_2 \in G : \quad \phi(g_1 \cdot g_2) = \phi(g_1) \cdot \phi(g_2).$$

**Definition A.2** (Representations). *Let $G$ be a group. A* representation $\rho : G \to \mathrm{GL}(V)$ *of a group $G$ on a vector space $V$ over a field $F$ is a group homomorphism $\rho : G \to \mathrm{GL}(V)$.*

Next we define *group action*, which generalizes group representation. We often want to describe how the elements of a group can effect a set – transforming its elements in a way that reflects the group's algebraic structure. This idea is formalized through the concept of a group action.

**Definition A.3** (Group Action). *Let $G$ be a group and $X$ a set. A* group action *of $G$ on $X$ is a mapping $G \times X \to X$, denoted by $\cdot$, such that*

- $g \cdot (h \cdot x) = (gh) \cdot x$,

- $e \cdot x = x$,

*for all $g, h \in G$ and all $s \in S$.*

We say that the group element $g \in G$ *acts on* the element $x \in X$ via $g \cdot x$. A representation $\rho : G \to \mathrm{GL}(V)$ is a special case of a group action where the set $X$ is a vector space $V$ and the action of $G$ on $V$ is given by $g \cdot v := \rho(g)v$ for $v \in V$.

Given a group action, it is natural to ask about the set of all elements that can be reached from a given one via the group action. This leads to the concept of an orbit.

**Definition A.4** (Orbit). *Let $G$ be a group acting on a set $X$. The* orbit *of $x \in X$ is defined to be*

$$\mathcal{O}_G(x) := \{g \cdot x \mid g \in G\}.$$

Given a group representation, we often look for subspaces that are preserved by the action of all group elements. These subspaces are known as $G$-invariant subspaces and the restricted representation to this subspace is called a subrepresentation.

**Definition A.5** ($G$-invariant Subspace). *Let $\rho : G \to \mathrm{GL}(V)$ be a representation. A linear subspace $W \subseteq V$ is called $G$-invariant if*

$$\forall\, g \in G,\ w \in W : \quad \rho(g)w \in W.$$

**Definition A.6** (Subrepresentation). *Let $\rho : G \to \mathrm{GL}(V)$ be a representation. Let $W \subseteq V$ be a $G$-invariant subspace. The map $\rho|_W : G \to \mathrm{GL}(W)$ defined to be $\rho(g)$ restricted to inputs from $W$, is called the* subrepresentation *of $\rho$ on $W$.*

Some representations cannot be broken down into smaller, nontrivial subrepresentations. These are known as irreducible representations.

**Definition A.7** (Irreducible Representation). *A representation $\rho : G \to \mathrm{GL}(V)$ is called* irreducible *if the only $G$-invariant subspaces of $V$ are the trivial subspaces $\{0\}$ and $V$.*

We next define symmetry preserving mappings. Such mappings are typically called *intertwining operators* in representation theory, but a more common term for them in deep learning is *equivariant mappings*.

**Definition A.8** (Equivariant Mapping between Representations). *Let $\rho_V : G \to \mathrm{GL}(V)$ and $\rho_W : G \to \mathrm{GL}(W)$ be representations of a group $G$. A linear mapping $f : V \to W$ is called $G$-equivariant (with respect to $\rho_V$ and $\rho_W$) if it satisfies*

$$\forall\, g \in G,\ v \in V : \quad f(\rho_V(g)v) = \rho_W(g)f(v).$$

Next, we define which representations are considered as equivalent.

**Definition A.9** (Isomorphic Representations). *Let $\rho_V : G \to \mathrm{GL}(V)$ and $\rho_W : G \to \mathrm{GL}(W)$ be two representations. The representations are called* isomorphic *(denoted by $\rho_V \cong \rho_W$) if there exists a bijective linear map $\varphi : V \to W$ (i.e., an isomorphism of linear spaces $V \cong W$) such that*

$$\forall\, v \in V,\ g \in G : \quad \varphi(\rho_V(g)v) = \rho_W(g)\varphi(v).$$

*Such a $\varphi$ is called a* representation isomorphism.

A classical result characterizing linear equivariant mappings between irreducible representations is Schur's lemma. Given two irreducible representations, one may anticipate that there could potentially be many $G$-equivariant mappings between them. However, Schur's lemma states that is essentially either one or zero such mappings. The following is one variant of the lemma.

**Lemma A.10** (Theorem 2 in [45], Schur's Lemma). *Let $G$ be a finite group. Let $\rho_V : G \to \mathrm{GL}(V)$ and $\rho_W : G \to \mathrm{GL}(W)$ be irreducible representations. Let $\mathcal{L}$ be the space of $G$-equivariant linear mappings from $V$ to $W$.*

1. *If $\rho_V$ and $\rho_W$ are not isomorphic, then $\mathcal{L}$ is $0$-dimensional (only $0$ intertwines $\rho_V$ and $\rho_W$).*

2. *If $\rho_V$ and $\rho_W$ are isomorphic, then $\mathcal{L}$ is $1$-dimensional (up to multiplication by scalar, there is only one equivariant operator between $\rho_V$ and $\rho_W$, which is a representation isomorphism).*

Part 2 of Schur's lemma means that if there exists an isomorphism $\varphi$ between the irreducible representations $\rho_V : G \to \mathrm{GL}(V)$ and $\rho_W : G \to \mathrm{GL}(W)$ then any other isomorphism of these representations is a scalar times $\varphi$. Moreover, any linear $G$-equivariant mapping must be an isomorphism or the zero mapping.

For two representations $\rho_V : G \to \mathrm{GL}(V)$ and $\rho_W : G \to \mathrm{GL}(W)$, we define the representation $\rho_V \oplus \rho_W : G \to \mathrm{GL}(V \oplus W)$, for $v \in V$ and $w \in W$, by

$$(\rho_V \oplus \rho_W)(g)(v,w) = (\rho_V(g)v, \rho_W(g)w).$$

The tensor product of $\rho_V \otimes \rho_W : G \to \mathrm{GL}(V \otimes W)$ is defined on simple tensors $v \otimes w$ (where $v \in V$ and $w \in W$) by

$$(\rho_V \otimes \rho_W)(g)(v,w) = (\rho_V(g)v) \otimes (\rho_W(g)w).$$

Since the linear closure of the simple tensors is the whole tensor product space $V \otimes W$, the above definition uniquely extends from simple tensors $w \otimes z$ to general elements in the tensor product space $Z \in V \otimes W$. For example, when $V = \mathbb{R}^N$ and $W = \mathbb{R}^M$, we identify $V \otimes W$ with $\mathbb{R}^{N \times M}$, and for matrices $\rho_V(g) \in \mathbb{R}^{N \times N}$ and $\rho_W(g) \in \mathbb{R}^{M \times M}$ we have for any $\boldsymbol{Z} \in \mathbb{R}^{N \times M}$

$$(\rho_V \otimes \rho_W)(g)\boldsymbol{Z} = \rho_V(g)\boldsymbol{Z}\rho_W(g)^\top.$$

In this example, the simple tensors are rank-1 matrices, and the space of all matrices is spanned by the rank-1 matrices.

From this point onward, when the representation of the group $G$ is clear from context, we use the shorthand notation $g \cdot \boldsymbol{X}$ instead of $\rho_V(g)\boldsymbol{X}$ to denote the representation $\rho_V(g)$ acting on $\boldsymbol{X} \in V$.

## A.2    Representation Theory of the Symmetric Group

Here, we present standard results on the standard permutation representation of the symmetric group. Recall that $S_N$ denotes the group of permutations, i.e. the *symmetric group*.

**Definition A.11** (The Symmetric Group). *The* symmetric group $S_N$ *is the group of all permutations of the set $[N] = \{1, \ldots, N\}$.*

We denote by $\sigma = (ij) \in S_N$ the permutation that swaps the $i$-th and $j$-th elements, leaving the remaining elements fixed.

**Definition A.12** (The Standard Permutation Representation). *The* standard permutation representation $\rho_N : S_N \to \mathrm{GL}\left(\mathbb{R}^N\right)$ *is given by*

$$\forall \sigma_N \in S_N, \boldsymbol{x} \in \mathbb{R}^N : \quad \rho_N(\sigma_N)\boldsymbol{x} = \left(x_{\sigma_N^{-1}(1)}, \ldots, x_{\sigma_N^{-1}(N)}\right).$$

The following two subspaces of $\mathbb{R}^N$, denoted by $V_0^N$ and $V_1^N$, can be shown to be the only two irreducible subrepresentations of the standard permutation representation of $S_N$ on $\mathbb{R}^N$.

**Definition A.13** (The standard subspace and representation). *The space*

$$V_0^N = \{\boldsymbol{x} \in \mathbb{R}^N \mid \mathbf{1}_N^\top \boldsymbol{x} = 0\}$$

*is called the* $(N-1)$*-dimensional* $S_N$*-invariant* standard subspace. *The* standard representation *of* $S_N$, *denoted by* $\rho_{V_0^N} : S_N \to \mathrm{GL}\left(V_0^N\right)$, *is defined as*

$$\forall \sigma_N \in S_N, \ \boldsymbol{x} \in V_0^N : \quad \rho_{V_0^N}(\sigma_N)\boldsymbol{x} = \left(x_{\sigma_N^{-1}(1)}, \ldots, x_{\sigma_N^{-1}(N)}\right).$$

**Definition A.14** (The trivial subspace and representation). *The space*

$$V_1^N = \{c\mathbf{1}_N \mid c \in \mathbb{R}\}$$

*is called the* $1$*-dimensional* $S_N$*-invariant* trivial subspace. *The* trivial representation *of* $S_N$, *denoted by* $\rho_{V_1^N} : S_N \to \mathrm{GL}\left(V_1^N\right)$, *is defined as*

$$\forall \sigma_N \in S_N : \ \rho_{V_1^N}(\sigma_N) = I, \quad \textit{(where I is the identity operator)}.$$

We adopt the names of the above two spaces from Chapter 3 in [40].

It can now be seen that $\mathbb{R}^N$ decomposes to a direct sum of the two irreducible subspaces of $S_N$ defined above. Moreover, the standard permutation representation also decomposes to the above two representations.

**Lemma A.15.** *The subspaces $V_1^N$ and $V_0^N$ of $\mathbb{R}^N$ are orthogonal, and $\mathbb{R}^N \cong V_1^N \oplus V_0^N$. Moreover, $\rho_N \cong \rho_{V_0^N} \oplus \rho_{V_1^N}$, and the representations $\rho_{V_0^N}$ and $\rho_{V_1^N}$ are the only two irreducible subrepresentations of $\rho_N$.*

## A.3 The Triple-Symmetric Group

Let $K, N, F, C \in \mathbb{N}$. In the setting of the triple-symmetric group, we consider three copies of symmetric groups $S_N$, $S_F$, and $S_C$, each acting on one dimension of the input tensor. Namely, each of the three groups acts on $\boldsymbol{X} \in \mathbb{R}^{K \times N \times (F+C)}$ as follows.

- $S_N$ acts by permuting the *second axis*, i.e.

$$(\sigma_N \cdot \boldsymbol{X})_{k,n,j} = \boldsymbol{X}_{k,\sigma_N^{-1}(n),j}, \quad \forall \sigma_N \in S_N, k \in [K], n \in [N], j \in [F+C].$$

- $S_F$ acts by permuting the first $F$ entries of the *third axis*, i.e.

$$(\sigma_F \cdot \boldsymbol{X})_{k,n,f} = \boldsymbol{X}_{k,n,\sigma_F^{-1}(f)}, \quad \forall \sigma_F \in S_F, k \in [K], n \in [N], f \in [F],$$

- $S_C$ acts by permuting the last $C$ entries of the *third axis*, i.e.

$$(\sigma_F \cdot \boldsymbol{X})_{k,n,F+c} = \boldsymbol{X}_{k,n,F+\sigma_C^{-1}(c)}, \quad \forall \sigma_C \in S_F, k \in [K], n \in [N], c \in [C].$$

When $K = 1$, we simply consider the space $\mathbb{R}^{N \times (F+C)}$, where, here, $S_N$ acts of the first axis, and $S_F$ and $S_C$ act on the second axis as above.

# B  Proof of Proposition 4.1

In this section we provide the proof for Proposition 4.1.

Let $\rho: G \to \mathrm{GL}\left(\mathbb{R}^{N \times (F+C)}\right)$ be the representation defined by

$$((\sigma_N, \sigma_F, \sigma_C) \cdot \boldsymbol{X})_{n,j} = \begin{cases} \boldsymbol{X}_{\sigma_N^{-1}(n), \sigma_F^{-1}(j)} & \text{if } j \in [F], \\ \boldsymbol{X}_{\sigma_N^{-1}(n), F + \sigma_C^{-1}(j-F)} & \text{otherwise .} \end{cases} \tag{3}$$

Denote

$$R^F = \{\boldsymbol{x} \in \mathbb{R}^{F+C} \mid \boldsymbol{x}_{F+1:F+C} = 0\}, \qquad R^C = \{\boldsymbol{x} \in \mathbb{R}^{F+C} \mid \boldsymbol{x}_{1:F} = 0\},$$
$$R_0^F = \{\boldsymbol{x} \in R^F \mid \boldsymbol{1}_{F+C}^\top \boldsymbol{x} = 0\}, \qquad R_0^C = \{\boldsymbol{x} \in R^C \mid \boldsymbol{1}_{F+C}^\top \boldsymbol{x} = 0\},$$
$$R_1^F = \{(c\,\boldsymbol{1}_F, 0) \mid c \in \mathbb{R}\}, \qquad R_1^C = \{(0, c\,\boldsymbol{1}_C) \mid c \in \mathbb{R}\}.$$

**Lemma B.1.** *The only irreducible subrepresentaions of $\rho$ are on the following eight invariant subspaces*

$$W_{a,b,L} = V_a^N \otimes R_b^L \subset \mathbb{R}^{N \times (F+C)}$$

*for $a, b \in \{0,1\}$ and $L \in \{F, C\}$, where $V_a^N$ are defined in Definitions A.13 and A.14.*

*Proof.*  By construction

$$\mathbb{R}^{N \times (F+C)} \cong \left(\mathbb{R}^N \otimes \mathbb{R}^F\right) \oplus \left(\mathbb{R}^N \otimes \mathbb{R}^C\right) \quad \text{and} \quad \rho \cong (\rho_{\mathbb{R}^N} \otimes \rho_{\mathbb{R}^F}) \oplus (\rho_{\mathbb{R}^N} \otimes \rho_{\mathbb{R}^C}), \tag{4}$$

where $\rho_{\mathbb{R}^D}: G \to \mathrm{GL}\left(\mathbb{R}^D\right)$ are the representations of $G$ on $\mathbb{R}^D$ for $D \in \{N, F, C\}$.

By Lemma A.15, the subspaces $V_1^D$ and $V_0^D$ of $\mathbb{R}^D \cong V_1^D \oplus V_0^D$ are orthogonal, and the representations $\rho_{V_0^D}$ and $\rho_{V_1^D}$ are the only two irreducible subrepresentations of $\rho_D \cong \rho_{V_0^D} \oplus \rho_{V_1^D}$, where $D \in \{N, F, C\}$. Thus, by substituting this into Equation (4), we get the eight invariant subspaces

$$\mathbb{R}^{N \times (F+C)} \cong \left(V_0^N \otimes \mathbb{R}_0^F\right) \oplus \left(V_1^N \otimes \mathbb{R}_0^F\right) \oplus \left(V_0^N \otimes \mathbb{R}_1^F\right) \oplus \left(V_1^N \otimes \mathbb{R}_1^F\right)$$
$$\oplus \left(V_0^N \otimes \mathbb{R}_0^C\right) \oplus \left(V_1^N \otimes \mathbb{R}_0^C\right) \oplus \left(V_0^N \otimes \mathbb{R}_1^C\right) \oplus \left(V_1^N \otimes \mathbb{R}_1^C\right)$$

and the corrisponding eight irreducible representation

$$\rho \cong \left(\rho_{V_0^N} \otimes \rho_{\mathbb{R}_0^F}\right) \oplus \left(\rho_{V_1^N} \otimes \rho_{\mathbb{R}_0^F}\right) \oplus \left(\rho_{V_0^N} \otimes \rho_{\mathbb{R}_1^F}\right) \oplus \left(\rho_{V_1^N} \otimes \rho_{\mathbb{R}_1^F}\right)$$
$$\oplus \left(\rho_{V_0^N} \otimes \rho_{\mathbb{R}_0^C}\right) \oplus \left(\rho_{V_1^N} \otimes \rho_{\mathbb{R}_0^C}\right) \oplus \left(\rho_{V_0^N} \otimes \rho_{\mathbb{R}_1^C}\right) \oplus \left(\rho_{V_1^N} \otimes \rho_{\mathbb{R}_1^C}\right).$$

$\square$

**Proposition 4.1.** *Let $F \neq C \in \mathbb{N}$. A linear function*

$$T = (T_1, T_2) \colon \mathbb{R}^{K_1 \times N \times F} \times \mathbb{R}^{K_1 \times N \times C} \to \mathbb{R}^{K_2 \times N \times F} \times \mathbb{R}^{K_2 \times N \times C}$$

*is $(S_N \times S_F \times S_C)$-equivariant if and only if there exist $\boldsymbol{\Lambda}^{(1)}, \ldots, \boldsymbol{\Lambda}^{(12)} \in \mathbb{R}^{K_1 \times K_2}$ such that for every $\boldsymbol{X} \in \mathbb{R}^{N \times F \times K_1}, \boldsymbol{Y} \in \mathbb{R}^{N \times C \times K_1}$ and $k_2 \in [K_2]$, the corresponding outputs $T_1(\boldsymbol{X}, \boldsymbol{Y}) \in \mathbb{R}^{N \times F \times K_2}$ and $T_2(\boldsymbol{X}, \boldsymbol{Y}) \in \mathbb{R}^{N \times C \times K_2}$ are given by*

$$T_1(\boldsymbol{X}, \boldsymbol{Y})_{:,:,k_2} = \left(\boldsymbol{1}_{N,N} \boldsymbol{X}^{(1)}_{:,:,k_2} + \boldsymbol{X}^{(2)}_{:,:,k_2}\right) \boldsymbol{1}_{F,F} + \boldsymbol{1}_{N,N} \boldsymbol{X}^{(3)}_{:,:,k_2} + \boldsymbol{X}^{(4)}_{:,:,k_2}$$
$$+ \left(\boldsymbol{1}_{N,N} \boldsymbol{Y}^{(5)}_{:,:,k_2} + \boldsymbol{Y}^{(6)}_{:,:,k_2}\right) \boldsymbol{1}_{C,F},$$

$$T_2(\boldsymbol{X}, \boldsymbol{Y})_{:,:,k_2} = \left(\boldsymbol{1}_{N,N} \boldsymbol{Y}^{(1)}_{:,:,k_2} + \boldsymbol{Y}^{(2)}_{:,:,k_2}\right) \boldsymbol{1}_{C,C} + \boldsymbol{1}_{N,N} \boldsymbol{Y}^{(3)}_{:,:,k_2} + \boldsymbol{Y}^{(4)}_{:,:,k_2}$$
$$+ \left(\boldsymbol{1}_{N,N} \boldsymbol{X}^{(5)}_{:,:,k_2} + \boldsymbol{X}^{(6)}_{:,:,k_2}\right) \boldsymbol{1}_{F,C},$$

*for every $k_2 \in [K_2]$, where for every $i \in [6]$, we define $\boldsymbol{X}^{(i)}_{:,:,k_2} = \sum_{k_1=0}^{K_1} \boldsymbol{X}_{:,:,k_1} \Lambda^{(i)}_{k_1,k_2}$ and $\boldsymbol{Y}^{(i)}_{:,:,k_2} = \sum_{k_1=0}^{K_1} \boldsymbol{Y}_{:,:,k_1} \Lambda^{(i+6)}_{k_1,k_2}$ .*

*Proof.* For $i \in \{1, 2\}$, consider the representation $\rho_i : G \to \mathrm{GL}\left(\mathbb{R}^{N \times (F+C) \times K_i}\right)$, where $G := S_N \times S_F \times S_C$ is defined in Section A.3 by

$$((\sigma_N, \sigma_F, \sigma_C) \cdot \boldsymbol{X})_{:,n,j} = \begin{cases} \boldsymbol{X}_{:,\sigma_N^{-1}(n),\sigma_F^{-1}(j)} & \text{if } j \in [F], \\ \boldsymbol{X}_{:,\sigma_N^{-1}(n),F+\sigma_C^{-1}(j-F)} & \text{otherwise} \end{cases} \tag{5}$$

for all $(\sigma_N, \sigma_F, \sigma_C) \in S_N \times S_F \times S_C$ and $\boldsymbol{X} \in \mathbb{R}^{N \times (F+C) \times K_i}$. Similarly, denote by $\rho : G \to \mathrm{GL}\left(\mathbb{R}^{N \times (F+C)}\right)$ the representation as in (5) with $K_i = 1$.

Let $i \in \{1, 2\}$. For each $k \in [K_i]$, let $V_k^i \subset \mathbb{R}^{N \times (F+C) \times K_i}$ be the subspace of vectors $\boldsymbol{X}$ such that $\boldsymbol{X}_{n,j,k'} = 0$ for every $k' \neq k$. Note that $V_K^i$ is an invariant subspace. The subrepresentation $\rho_i|_{V_k^i}$ is isomorphic to $\mathbb{R}^{N \times (F+C)}$ via the isomorphism $\pi_k^i : \mathbb{R}^{N \times (F+C)} \to V_k^i$ defined by $\pi_k^i(\boldsymbol{X}) = \boldsymbol{X}_{:,:,k}$.

By Lemma B.1, the only irreducible subspaces of $\rho_i$ are

$$\{V_{k,a,b,L}^i := (\pi_k^i)^{-1}(V_a^N \otimes R_b^L)\}_{k \in [K_i], a,b \in \{0,1\}, L \in \{F,C\}}.$$

By dimensionality consideration, the only pairs of subrepresentations of $\rho_1$ and $\rho_2$ that can be isomorphic are $\rho_1|_{V_{k_1,a,b,L}^1}$ and $\rho_2|_{V_{k_2,a,b,L}^2}$ for any $k_1 \in [K_1]$, $k_2 \in [K_2]$ and $a, b \in \{0,1\}$, $L \in \{F,C\}$. Moreover, it is easy to verify that $\rho_1|_{V_{k_1,a,b,L}^1} \sim \rho_2|_{V_{k_2,a,b,L}^2}$ for any such $k_1, k_2, a, b, L$. Indeed, a representation isomorphism is given by $\pi_{k_2}^2 \circ (\pi_{k_1}^1)^{-1}$.

Note that we have the direct sum decomposition into orthogonal subspaces

$$\mathbb{R}^{N \times (F+C) \times K_i} = \bigoplus_{k \in [K_i], a,b \in \{0,1\}, L \in \{F,C\}} V_{k,a,b,L}^i.$$

Let $P_{k,a,b,L}^i$ be the orthogonal projection upon $V_{k,a,b,L}^i$. Let $T$ be an intertwining operator between $\rho_1$ and $\rho_2$. We can decompose $T$ into the direct sum

$$T = \sum_{k,k',a,a',b,b',L,L'} P_{k',a',b',L'}^i T P_{k,a,b,L}^i. \tag{6}$$

By Schur's lemma (Lemma A.10), the only nonzero components of (6) are those with $a = a', b = b', L = L'$. Moreover, by Schur's lemma, the operator $T_{k,k',a,b,L} := P_{k,a,b,L}^i T P_{k,a,b,L}^i = T P_{k,a,b,L}^i$ can only come from a one dimensional space of operators.

Next, we find these one dimensional spaces explicitly:

$$T_1 : V_{k,k',0,0,F} \to V_{k,k',0,0,F}, \qquad T_1(\boldsymbol{X}) = \boldsymbol{X} - \mathbf{1}_{N,N}\,\boldsymbol{X}\,\mathbf{1}_{F,F}$$

$$T_2 : V_{k,k',0,1,F} \to V_{k,k',0,1,F}, \qquad T_2(\boldsymbol{X}) = \left(\boldsymbol{X} - \mathbf{1}_{N,N}\,\boldsymbol{X}\right)\mathbf{1}_{F,F}$$

$$T_3 : V_{k,k',1,0,F} \to V_{k,k',1,0,F}, \qquad T_3(\boldsymbol{X}) = \mathbf{1}_{N,N}\left(\boldsymbol{X} - \boldsymbol{X}\,\mathbf{1}_{F,F}\right)$$

$$T_4 : V_{k,k',1,1,F} \to V_{k,k',1,1,F}, \qquad T_4(\boldsymbol{X}) = \mathbf{1}_{N,N}\,\boldsymbol{X}\,\mathbf{1}_{F,F}$$

$$T_5 : V_{k,k',0,0,C} \to V_{k,k',0,0,C}, \qquad T_5(\boldsymbol{Y}) = \boldsymbol{Y} - \mathbf{1}_{N,N}\,\boldsymbol{Y}\,\mathbf{1}_{C,C}$$

$$T_6 : V_{k,k',0,1,C} \to V_{k,k',0,1,C}, \qquad T_6(\boldsymbol{Y}) = \left(\boldsymbol{Y} - \mathbf{1}_{N,N}\,\boldsymbol{Y}\right)\mathbf{1}_{C,C}$$

$$T_7 : V_{k,k',1,0,C} \to V_{k,k',1,0,C}, \qquad T_7(\boldsymbol{Y}) = \mathbf{1}_{N,N}\left(\boldsymbol{Y} - \boldsymbol{Y}\,\mathbf{1}_{C,C}\right)$$

$$T_8 : V_{k,k',1,1,C} \to V_{k,k',1,1,C}, \qquad T_8(\boldsymbol{Y}) = \mathbf{1}_{N,N}\,\boldsymbol{Y}\,\mathbf{1}_{C,C}$$

$$T_9 : V_{k,k',0,1,F} \to V_{k,k',0,1,C}, \qquad T_9(\boldsymbol{X}) = \boldsymbol{X}\,\mathbf{1}_{F,C}$$

$$T_{10} : V_{k,k',0,1,C} \to V_{k,k',0,1,F}, \qquad T_{10}(\boldsymbol{Y}) = \boldsymbol{Y}\,\mathbf{1}_{C,F}$$

$$T_{11} : V_{k,k',1,1,F} \to V_{k,k',1,1,C}, \qquad T_{11}(\boldsymbol{X}) = \mathbf{1}_{N,N}\,\boldsymbol{X}\,\mathbf{1}_{F,C}$$

$$T_{12} : V_{k,k',1,1,C} \to V_{k,k',1,1,F}, \qquad T_{12}(\boldsymbol{Y}) = \mathbf{1}_{N,N}\,\boldsymbol{Y}\,\mathbf{1}_{C,F}$$

It is easy to see that each of the above transformation is a representation isomorphism. Lastly, the set of transformations from Proposition 4.1 is a reparameterization of the above isomorphisms.

$\square$

## C Extended notations of TSNets

**Definition C.1.** *Let* $\text{state}_1, \text{state}_2 \in \{\text{self}, \text{pool}\}$ *and* $D, L \in \{F, C\}$. *For* $k_1 \le K_1 \in \mathbb{N}$ *and* $k_2 \le K_2 \in \mathbb{N}$, *denote by* $[D_{\text{state}_1} \xrightarrow{N_{\text{state}_2}} L]^{(k_1, k_2)} : \mathbb{R}^{N \times (F+C) \times K_1} \to \mathbb{R}^{N \times (F+C) \times K_2}$ *the operator, defined for* $(\boldsymbol{X}, \boldsymbol{Y}) \in \mathbb{R}^{N \times (F+C) \times K_1}$ *by*

$$T = \sum_{i=1}^{12} \sum_{k_2=1}^{K_2} \sum_{k_1=1}^{K_1} \Lambda_{k_1, k_2}^{(i)} \left[ D_{\text{state}1(i)} \xrightarrow{N_{\text{state}2(i)}} L(i) \right]^{(k_1, k_2)},$$

*and for any* $(k_1', k_2') \neq (k_1, k_2)$

$$[D_{\text{state}_1} \xrightarrow{N_{\text{state}_2}} L]^{(k_1, k_2)}(\boldsymbol{X}, \boldsymbol{Y})_{k_2', :, :} = (0, 0).$$

This is exactly the form stated in Proposition 4.1: in particular,

$$T_1(\boldsymbol{X}, \boldsymbol{Y})_{:,:,k_2} = \left( \mathbf{1}_{N,N} \boldsymbol{X}_{:,:,k_2}^{(1)} + \boldsymbol{X}_{:,:,k_2}^{(2)} \right) \mathbf{1}_{F,F} + \mathbf{1}_{N,N} \boldsymbol{X}_{:,:,k_2}^{(3)} + \boldsymbol{X}_{:,:,k_2}^{(4)}$$
$$+ \left( \mathbf{1}_{N,N} \boldsymbol{Y}_{:,:,k_2}^{(5)} + \boldsymbol{Y}_{:,:,k_2}^{(6)} \right) \mathbf{1}_{C,F},$$
$$T_2(\boldsymbol{X}, \boldsymbol{Y})_{:,:,k_2} = \left( \mathbf{1}_{N,N} \boldsymbol{Y}_{:,:,k_2}^{(1)} + \boldsymbol{Y}_{:,:,k_2}^{(2)} \right) \mathbf{1}_{C,C} + \mathbf{1}_{N,N} \boldsymbol{Y}_{:,:,k_2}^{(3)} + \boldsymbol{Y}_{:,:,k_2}^{(4)}$$
$$+ \left( \mathbf{1}_{N,N} \boldsymbol{X}_{:,:,k_2}^{(5)} + \boldsymbol{X}_{:,:,k_2}^{(6)} \right) \mathbf{1}_{F,C},$$

where for each $i \in [6]$, $\boldsymbol{X}_{:,:,k_2}^{(i)} = \sum_{k_1=1}^{K_1} \boldsymbol{X}_{:,:,k_1} \Lambda_{k_1, k_2}^{(i)}$ and $\boldsymbol{Y}_{:,:,k_2}^{(i)} = \sum_{k_1=1}^{K_1} \boldsymbol{Y}_{:,:,k_1} \Lambda_{k_1, k_2}^{(i+6)}$.

For $(\boldsymbol{X}, \boldsymbol{Y}) \in \mathbb{R}^{N \times (F+C)}$ (i.e., $K_1 = K_2 = 1$), the twelve building blocks read

$$[F_{\text{self}} \xrightarrow{N_{\text{self}}} F](\boldsymbol{X}, \boldsymbol{Y}) = (\boldsymbol{X}, 0), \qquad\qquad [F_{\text{pool}} \xrightarrow{N_{\text{self}}} F](\boldsymbol{X}, \boldsymbol{Y}) = (\boldsymbol{X} \, \mathbf{1}_{F,F}, 0),$$

$$[F_{\text{self}} \xrightarrow{N_{\text{pool}}} F](\boldsymbol{X}, \boldsymbol{Y}) = (\mathbf{1}_{N,N} \boldsymbol{X}, 0), \qquad [F_{\text{pool}} \xrightarrow{N_{\text{pool}}} F](\boldsymbol{X}, \boldsymbol{Y}) = (\mathbf{1}_{N,N} \boldsymbol{X} \, \mathbf{1}_{F,F}, 0),$$

$$[C_{\text{pool}} \xrightarrow{N_{\text{self}}} F](\boldsymbol{X}, \boldsymbol{Y}) = (\boldsymbol{Y} \, \mathbf{1}_{C,F}, 0), \qquad [C_{\text{pool}} \xrightarrow{N_{\text{pool}}} F](\boldsymbol{X}, \boldsymbol{Y}) = (\mathbf{1}_{N,N} \boldsymbol{Y} \, \mathbf{1}_{C,F}, 0),$$

$$[C_{\text{self}} \xrightarrow{N_{\text{self}}} C](\boldsymbol{X}, \boldsymbol{Y}) = (0, \boldsymbol{Y}), \qquad\qquad [C_{\text{pool}} \xrightarrow{N_{\text{self}}} C](\boldsymbol{X}, \boldsymbol{Y}) = (0, \boldsymbol{Y} \, \mathbf{1}_{C,C}),$$

$$[C_{\text{self}} \xrightarrow{N_{\text{pool}}} C](\boldsymbol{X}, \boldsymbol{Y}) = (0, \mathbf{1}_{N,N} \boldsymbol{Y}), \qquad [C_{\text{pool}} \xrightarrow{N_{\text{pool}}} C](\boldsymbol{X}, \boldsymbol{Y}) = (0, \mathbf{1}_{N,N} \boldsymbol{Y} \, \mathbf{1}_{C,C}),$$

$$[F_{\text{pool}} \xrightarrow{N_{\text{self}}} C](\boldsymbol{X}, \boldsymbol{Y}) = (0, \boldsymbol{X} \, \mathbf{1}_{F,C}), \qquad [F_{\text{pool}} \xrightarrow{N_{\text{pool}}} C](\boldsymbol{X}, \boldsymbol{Y}) = (0, \mathbf{1}_{N,N} \boldsymbol{X} \, \mathbf{1}_{F,C}).$$

**Interpretation of *self* and *pool*.**

- *Self* along an axis means "no aggregation" on that axis (identity action). E.g., $F_{\text{self}}$ keeps per-feature structure; $N_{\text{self}}$ keeps per-node structure.

- *Pool* along an axis means "sum-and-broadcast" on that axis via $\mathbf{1}$: left multiplication by $\mathbf{1}_{N,N}$ pools over nodes and copies the node-sum to every node; right multiplication by $\mathbf{1}_{F,F}$ (resp. $\mathbf{1}_{C,C}$) pools over features (resp. labels) and broadcasts to all features (resp. labels).

- Cross–family maps ($F \to C$ or $C \to F$) necessarily use the corresponding all-ones bridge $\mathbf{1}_{F,C}$ or $\mathbf{1}_{C,F}$, hence require the source to be in the "pool" state.

## D Proof of Theorem 4.3

This section presents the proof of Theorem 4.3. We describe relevant background and definitions in Section D.1. We then review results on symmetric polynomials in Section D.3 and extend them to our setting of triple symmetry in Section D.4. The subsequent Section D.6–Section D.10 provides supporting lemmas and approximation properties that play a role in Theorem 4.3. In Sections D.11 and D.12 we discuss properties of topological spaces required for the proof of Theorem 4.3. Finally, we complete the proof of Theorem 4.3 in Section D.13.

**Proof strategy.** We prove Theorem 4.3 while avoiding the aforementioned caveat by closely utilizing the proof techniques introduced in [38] and [26]. Specifically, our proof idea can be decomposed into four high-level steps: **(i) Characterization of $(S_F \times S_C)$-invariant polynomials.** It is well established that any $S_N$-invariant polynomial can be represented as a polynomials in the multisymmetric power-sum polynomials [4, 38, 37]. We provide a non-trivial extension to a richer symmetry setting by defining the doubly-symmetric polynomials (DMPs) (see Definition D.7) and showing that any $(S_F \times S_C)$-invariant polynomial, can be represented as a polynomials in the DMPS (see Lemma D.8). **(ii) Expressing $(S_{N-1} \times S_{F-1} \times S_C)$-invariant polynomials using DMPs.** We adapt the techniques from Theorem 3 of [26] and Lemma 2 of [38] to use the DMPs introduced in (i) and the $(S_N \times S_F \times S_C)$-equivariant linear maps we derived in Equation (2) to express $(S_{N-1} \times S_{F-1} \times S_C)$- or $(S_{N-1} \times S_F \times S_{C-1})$-invariant polynomials. **(iii) Composition of $(S_N \times S_F \times S_C)$-equivariant polynomials.** We extend Lemmas 1 from [38] to show that the outputs of an $(S_N \times S_F \times S_C)$-equivariant polynomial are $(S_{N-1} \times S_{F-1} \times S_C)$- or $(S_{N-1} \times S_F \times S_{C-1})$-invariant. By leveraging the stronger symmetry structure provided by the permutation groups rather than their subgroups, we avoid the pitfall in Theorem 3 by [26] – enabling the composition of equivariant functions via invariant ones, which was also the goal of the flawed transition in [26], as discussed above. **(iv) Approximating $(S_N \times S_C)$-equivariant and $S_F$-invariant functions using TSNets.** We replicate each intermediate step from (ii) and (iii), constructing approximators for the corresponding polynomial spaces using the $(S_N \times S_F \times S_C)$-equivariant linear maps derived in Equation (2), and multi-layer perceptrons instantiated from the Universal Approximation Theorem (D.12). Finally, by incorporating the label projection map $\pi$ as the final layer of our architecture, we enforce $S_F$-invariance and yield an approximator for the desired class of $(S_N \times S_C)$-equivariant and $S_F$-invariant functions, completing our construction for the triple-symmetry setting. A more detailed illustration, incorporating additional steps, is provided in Figure 4 with the complete proof detailed in Section D.

Note that when labels are removed (i.e., $C = 0$), steps (ii) and (iii) recover a corrected but less general version of Theorem 3 in [26] for $H = S_F$, using techniques from [38, 26].

## D.1 Background and Definitions

**Norms.** The $L_\infty$ norm of a vector $\boldsymbol{x} \in \mathbb{R}^L$ is defined to be $\|\boldsymbol{x}\|_\infty = \max_{i \in [L]} |\boldsymbol{x}_i|$. The $L_\infty(\mathcal{K})$ norm of a continuous function $f : \mathcal{K} \to \mathbb{R}^L$ over a compact topological space $\mathcal{K}$ is defined to be

$$\|f\|_\infty = \max_{\boldsymbol{x} \in \mathcal{K}} \|f(\boldsymbol{x})\|_\infty .$$

We often denote in short $L_\infty$ instead of $L_\infty(\mathcal{K})$. Note that any continuous function over a compact domain $\mathcal{K}$ has a finite $L_\infty(\mathcal{K})$ norm. We denote the space of all continuous functions over $\mathcal{K}$ by $L_\infty(\mathcal{K})$.[1]

**Multivariate functions.** Let $\boldsymbol{X} \in \mathbb{R}^{N \times (F+C)}$. We denote the first $F$ columns by $\boldsymbol{X}^{(F)} \in \mathbb{R}^{N \times F}$ and the last $C$ columns by $\boldsymbol{X}^{(C)} \in \mathbb{R}^{N \times C}$, and write $\boldsymbol{X} = (\boldsymbol{X}^{(F)}, \boldsymbol{X}^{(C)})$.

Let $\boldsymbol{X} \in \mathbb{R}^{N \times D}$ and let $p : \mathbb{R}^{N \times D} \to \mathbb{R}^{L \times K}$. For any $l \in [L], k \in [K]$, we denote by $p_{l,k} : \mathbb{R}^{N \times D} \to \mathbb{R}$ the $(l, k)$-th element of $p(\boldsymbol{X})$. For any $l \in [L]$, we denote by $p_l : \mathbb{R}^{N \times D} \to \mathbb{R}^K$ the $l$-th row of $p(\boldsymbol{X})$. We call $p_l$ the $K$-dimensional feature vector corresponding to the $l$-th row of $p(\boldsymbol{X})$, and write

$$p_l(\boldsymbol{X}) = (p(\boldsymbol{X})_{l,1}, p(\boldsymbol{X})_{l,2}, \dots, p(\boldsymbol{X})_{l,K}) \in \mathbb{R}^K.$$

## D.2 Restrictions of the triple-symmetric group representation

Consider the subgroup $S_{N-1} \times S_{F-1} \times S_C$ of the triple-symmetric group. We define the action of $S_{N-1} \times S_{F-1} \times S_C$ on $\mathbb{R}^{N \times (F+C)}$ by restricting the permutation $\sigma_{F-1} \in S_{F-1}$ to act only on the second to the $F$-th columns, and restricting the permutation $\sigma_{N-1} \in S_{N-1}$ to act only on last

---

[1] In the literature the space of continuous functions over $\mathcal{K}$ with the infinity norm is typically denoted by $C(\mathcal{K})$.

$(N-1)$-th rows, keeping element $(1,1)$ fixed. Namely,

$$((\sigma_{N-1}, \sigma_{F-1}, \sigma_C) \cdot \mathbf{X})_{n,j} = \begin{cases} \mathbf{X}_{1,1} & \text{if } n = 1 \text{ and } j = 1, \\ \mathbf{X}_{1,1+\sigma_{F-1}^{-1}(j-1)} & \text{if } n = 1 \text{ and } j \in [F] \setminus [1], \\ \mathbf{X}_{1,F+\sigma_C^{-1}(j-F)} & \text{if } n = 1 \text{ and } j \in [F+C] \setminus [F], \\ \mathbf{X}_{1+\sigma_{N-1}^{-1}(n-1),1} & \text{if } n \in [N] \setminus [1] \text{ and } j = 1, \\ \mathbf{X}_{1+\sigma_{N-1}^{-1}(n-1),1+\sigma_{F-1}^{-1}(j-1)} & \text{if } n \in [N] \setminus [1] \text{ and } j \in [F] \setminus [1], \\ \mathbf{X}_{1+\sigma_{N-1}^{-1}(n-1),F+\sigma_C^{-1}(j-F)} & \text{otherwise}, \end{cases}$$

for all $(\sigma_{N-1}, \sigma_{F-1}, \sigma_C) \in S_{N-1} \times S_{F-1} \times S_C$ and $\mathbf{X} \in \mathbb{R}^{N \times (F+C)}$.

Similarly, the action of $S_{N-1} \times S_F \times S_{C-1}$ on $\mathbb{R}^{N \times (F+C)}$ is defined as follows. The permutation $\sigma_{C-1} \in S_{C-1}$ acts only on the last $C-1$ label columns and the permutation $\sigma_{N-1} \in S_{N-1}$ acts act only on last $N-1$-th rows, keeping element $(1, F+1)$ fixed. Namely,

$$((\sigma_{N-1}, \sigma_F, \sigma_{C-1}) \cdot \mathbf{X})_{n,j} =$$

$$= \begin{cases} \mathbf{X}_{1,F+1} & \text{if } n = 1 \text{ and } j = F+1, \\ \mathbf{X}_{1,F+1+\sigma_{C-1}^{-1}(j-F-1)} & \text{if } n = 1 \text{ and } j \in [F+C] \setminus [F+1], \\ \mathbf{X}_{1,\sigma_F^{-1}(j)} & \text{if } n = 1 \text{ and } j \in [F], \\ \mathbf{X}_{1+\sigma_{N-1}^{-1}(n-1),F+1} & \text{if } n \in [N] \setminus [1] \text{ and } j = F+1, \\ \mathbf{X}_{1+\sigma_{N-1}^{-1}(n-1),F+1+\sigma_{C-1}^{-1}(j-F-1)} & \text{if } n \in [N] \setminus [1] \text{ and } j \in [F+C] \setminus [F+1], \\ \mathbf{X}_{1+\sigma_{N-1}^{-1}(n-1),\sigma_F^{-1}(j)} & \text{otherwise}, \end{cases}$$

for all $(\sigma_{N-1}, \sigma_F, \sigma_{C-1}) \in S_{N-1} \times S_F \times S_{C-1}$ and $\mathbf{X} \in \mathbb{R}^{N \times (F+C)}$.

By abuse of notation, for $(\sigma_N, e, e) \in S_N \times S_F \times S_C$, we often denote in short $\sigma_N = (\sigma_N, e, e)$, and similarly denote $\sigma_F = (e, \sigma_F, e)$, and $\sigma_C = (e, e, \sigma_C)$.

### D.3 Symmetric Polynomials

This subsection introduces key definitions and lemmas from symmetric polynomials and invariant theory that will be extended to our setting of triple-symmetry in Section D.4.

**Polynomials.** A *multi-index* $\boldsymbol{\alpha}$ with $D \in \mathbb{N}$ components is an element of $\mathbb{N}_0^D$, where $\mathbb{N}_0$ is the set of non-negative integers. Given a multi-index $\boldsymbol{\alpha} \in \mathbb{N}_0^D$ and $\boldsymbol{x} = (\boldsymbol{x}_1, \cdots, \boldsymbol{x}_D) \in \mathbb{R}^D$, we define the *monomial* $\boldsymbol{x}^{\boldsymbol{\alpha}} := \boldsymbol{x}_1^{\alpha_1} \cdots \boldsymbol{x}_D^{\alpha_D}$ and call $\|\boldsymbol{\alpha}\|_1 := \sum_d \alpha_d$ the *degree* of the monomial.

**Definition D.1** (Polynomial Ring). *A polynomial ring is a set $R$ of polynomials from $\mathbb{R}^L$ to $\mathbb{R}$ invariant to the following operations: for all $p, q \in R$ and $c \in \mathbb{R}$*

- *Addition. $p + q \in R$,*

- *Multiplication. $pq \in R$,*

- *Scalar multiplication. $cp \in R$.*

Equivalently, a set $R$ of polynomials $\mathbb{R}^L \to \mathbb{R}$ is a ring if for every $D \in \mathbb{N}$, every $p_1, \ldots, p_D \in R$, and every polynomial $q : \mathbb{R}^D \to \mathbb{R}$, we have $q(p_1, \ldots, p_D) \in R$.

Let $R$ be a ring of polynomials. A finite set of polynomials $p_1, \ldots, p_L \subset R$ is called a *generating set* for $R$ if every polynomial in $R$ can be expressed as a polynomial in $p_1, \ldots, p_L$ with real coefficients.

**Symmetry preserving polynomials.**

**Definition D.2** (Invariant functions). *Let $G$ be a group that acts on $\mathbb{R}^L$. A function $p : \mathbb{R}^L \to \mathbb{R}^D$ is called $G$-invariant if*

$$p(g \cdot \boldsymbol{x}) = p(\boldsymbol{x}), \quad \forall g \in G, \boldsymbol{x} \in \mathbb{R}^L.$$

**Definition D.3** (Equivariant functions). *Let $G$ be a group that acts on $\mathbb{R}^L$. A function $p : \mathbb{R}^L \to \mathbb{R}^L$ is called $G$-equivariant if*

$$p(g \cdot \boldsymbol{x}) = g \cdot p(\boldsymbol{x}), \quad \forall g \in G, \boldsymbol{x} \in \mathbb{R}^L.$$

We now define the power-sum multi-symmetric polynomials, which form a generating set for the ring of invariant polynomials to permutations.

**Definition D.4** (Power-sum Multi-symmetric Polynomials). *Let $M, D \in \mathbb{N}$. Let $T = \binom{M+D}{D}$, and let $\{\boldsymbol{\alpha}^{(t)}\}_{t=1}^T$ be an enumeration of all multi-indices $\boldsymbol{\alpha}^{(t)} \in \mathbb{N}_0^D$ such that $\left\|\boldsymbol{\alpha}^{(t)}\right\|_1 \leq M$.[2] The power-sum multi-symmetric polynomials (PMP) $\{s_t\}_{t=1}^T : \mathbb{R}^{N \times D} \to \mathbb{R}$ are defined by*

$$s_t(\boldsymbol{X}) = \sum_{n=1}^N \boldsymbol{X}_{n,:}^{\boldsymbol{\alpha}^{(t)}}.$$

The following well-known result from invariant theory shows that the PMPs are a generating set for the ring of $S_N$ invariant polynomials.

**Lemma D.5** (Corollary 8.4 in [37]). *Let $p : \mathbb{R}^{N \times D} \to \mathbb{R}$ be an $S_N$-invariant polynomial with degree $M$. Then, it can be written as a polynomial in the PMPs*

$$p(\boldsymbol{X}) = q(s_1(\boldsymbol{X}), \ldots, s_T(\boldsymbol{X})),$$

*for some polynomial $q : \mathbb{R}^T \to \mathbb{R}$, where $T = \binom{M+D}{D}$.*

The following lemma enables the decomposition of polynomials that are $S_{N-1}$-invariant into a combination of polynomials that are $S_N$-invariant. The lemma, originally stated as Lemma 2 in [38], had a minor error in the summation index bounds, which we correct.

**Lemma D.6** (Lemma 2 in [38]). *Let $p : \mathbb{R}^{N \times D} \to \mathbb{R}$ be an $S_{N-1}$-invariant polynomial of degree $M$. Then, there exist $S_N$-invariant polynomials $\{q_{\boldsymbol{\alpha}} : \mathbb{R}^{N \times D} \to \mathbb{R}\}_{\|\boldsymbol{\alpha}\|_1 \leq M}$ such that*

$$p(\boldsymbol{X}) = \sum_{\|\boldsymbol{\alpha}\|_1 \leq M} \boldsymbol{X}_{1,:}^{\boldsymbol{\alpha}} q_{\boldsymbol{\alpha}}(\boldsymbol{X}).$$

*Here, $\boldsymbol{\alpha}$ runs over all multi-indices in $\mathbb{N}_0^D$ of degree less or equal to $M$.*

### D.4 Triple-symmetric Polynomials

In this subsection, we extend the results on symmetric polynomials to our triple-symmetry setting. This extension provides us with the mathematical tools needed to show that: (1) any symmetry preserving continuous function over symmetric compact domains can be approximated by triple-symmetry preserving polynomials. (2) any symmetry preserving polynomial over symmetric compact domains can be approximated by TSNets. Together, these results establish the universality of TSNets under triple symmetry.

We now define the doubly power-sum multi-symmetric polynomials, which form a generating set for the ring of $(S_F \times S_C)$-invariant polynomials $\mathbb{R}^{N \times (F+C)} \to \mathbb{R}$.

**Definition D.7** (Doubly power-sum Multi-symmetric Polynomials). *Let $M, N \in \mathbb{N}$. Let $T = \binom{M+N}{N}$, and let $\{\boldsymbol{\alpha}^{(t)}\}_{t=1}^T$ be an enumeration of all multi-indices $\boldsymbol{\alpha}^{(t)} \in \mathbb{N}_0^N$ such that $\left\|\boldsymbol{\alpha}^{(t)}\right\|_1 \leq M$. The doubly power-sum multi-symmetric polynomials (DMPs) $\{s_t^{(F)} : \mathbb{R}^{N \times (F+C)} \to \mathbb{R}\}_{t=1}^T \cup \{s_t^{(C)} : \mathbb{R}^{N \times (F+C)} \to \mathbb{R}\}_{t=1}^T$ are the polynomials*

$$s_t^{(F)}(\boldsymbol{X}) = \sum_{f=1}^F \boldsymbol{X}_{:,f}^{\boldsymbol{\alpha}^{(t)}} \quad \text{and} \quad s_t^{(C)}(\boldsymbol{X}) = \sum_{c=1}^C \boldsymbol{X}_{:,F+c}^{\boldsymbol{\alpha}^{(t)}}.$$

We next show that the DMPs form a generating set for the ring of $(S_F \times S_C)$-invariant polynomials $\mathbb{R}^{N \times (F+C)} \to \mathbb{R}$. This is later employed in the construction of the approximating neural networks in Theorem 4.3.

---

[2]We count the number of possible values of $\boldsymbol{\alpha}$ by introducing a slack variable $\alpha_{D+1} := M - \sum_{i=1}^D \alpha_i$, so that $\sum_{i=1}^{D+1} \alpha_i = M$. The number of such non-negative integer solutions is equivalent to distributing $M$ indistinguishable balls into $D+1$ distinguishable bins, which yields $\binom{M+D}{D}$ possibilities.

**Lemma D.8.** *Let $p : \mathbb{R}^{N \times (F+C)} \to \mathbb{R}$ be an $S_F \times S_C$-invariant polynomial of degree $M$. Then, $p$ can be written as a polynomial in the DMPs*

$$p(\boldsymbol{X}) = q(s_1^{(F)}(\boldsymbol{X}), \ldots, s_T^{(F)}(\boldsymbol{X}), s_1^{(C)}(\boldsymbol{X}), \ldots, s_T^{(C)}(\boldsymbol{X})),$$

*for some polynomial $q : \mathbb{R}^{2T} \to \mathbb{R}$, where $T = \binom{M+N}{N}$.*

*Proof.* Let $p$ be an invariant polynomial of degree $M$ as stated in the lemma. Using the notation $\boldsymbol{X} = (\boldsymbol{X}^{(F)}, \boldsymbol{X}^{(C)}) \in \mathbb{R}^{N \times (F+C)}$ for a point in the domain of $p$, we freeze $\boldsymbol{X}^{(C)}$ as a fixed parameter, and treat $p$ as a function only of $\boldsymbol{X}^{(F)}$, denoted by $p^{\boldsymbol{X}^{(C)}} : \boldsymbol{X}^{N \times F} \to \mathbb{R}$, i.e.,

$$p(\boldsymbol{X}) = p^{\boldsymbol{X}^{(C)}}(\boldsymbol{X}^{(F)}). \tag{7}$$

Note that $p^{\boldsymbol{X}^{(C)}}$ is an $S_F$-invariant polynomial $\mathbb{R}^{N \times F} \to \mathbb{R}$ for every $\boldsymbol{X}^{(C)} \in \mathbb{R}^{N \times C}$.

The space of $S_F$-invariant polynomials over $\mathbb{R}^{N \times F}$ of bounded degree is a finite dimensional linear space, and therefore has a basis $\{b_l\}_{l=1}^{L}$ with degree for each $b_l$ at most $M$. Hence, we can write

$$p^{\boldsymbol{X}^{(C)}}(\boldsymbol{X}^{(F)}) = \sum_{l=1}^{L} c_l(\boldsymbol{X}^{(C)}) b_l(\boldsymbol{X}^{(F)}), \tag{8}$$

where the coefficients $\{c_l(\boldsymbol{X}^{(C)})\}_{l=1}^{L} \in \mathbb{R}$ depend on $\boldsymbol{X}^{(C)}$. By Equations (7) and (8), we get

$$p(\boldsymbol{X}) = \sum_{l=1}^{L} c_l(\boldsymbol{X}^{(C)}) b_l(\boldsymbol{X}^{(F)}), \tag{9}$$

where the coefficients $\{c_l\}_{l=1}^{L} : \mathbb{R}^{N \times C} \to \mathbb{R}$ must be polynomials of degree at most $M$ in $\boldsymbol{X}^{(C)}$.

Let $\sigma_C \in S_C$. Since $p$ is $S_C$-invariant, we get

$$p(\boldsymbol{X}) = p(\sigma_C \cdot \boldsymbol{X}).$$

Namely,

$$\sum_{l=1}^{L} c_l(\boldsymbol{X}^{(C)}) b_l(\boldsymbol{X}^{(F)}) = \sum_{l=1}^{L} c_l(\sigma_C \cdot \boldsymbol{X}^{(C)}) b_l(\boldsymbol{X}^{(F)}).$$

Now, due to the fact that basis expansions are unique, we must have

$$\forall l \in [L], \quad c_l(\boldsymbol{X}^{(C)}) = c_l(\sigma_C \cdot \boldsymbol{X}^{(C)}).$$

Hence, each $\{c_l\}_{l=1}^{L}$ is an $S_C$-invariant polynomial.

As a result, by Lemma D.5, each $\{c_l\}_{l=1}^{L}$ can be written as a polynomial $v_l : \mathbb{R}^T \to \mathbb{R}$ in the multi-symmetric power-sum polynomials (PMPs) $\{s_i\}_{i=1}^{T} : \mathbb{R}^{N \times C} \to \mathbb{R}$, i.e.,

$$c_l(\boldsymbol{X}^{(C)}) = v_l(s_1(\boldsymbol{X}^{(C)}), \ldots, s_T(\boldsymbol{X}^{(C)})) \quad \forall l \in [L], \tag{10}$$

where $T = \binom{M+N}{N}$. Moreover, since $b_l$ are $S_F$-invariant, by Lemma D.5 each $\{b_l\}_{l=1}^{L}$ can also be written as a polynomial $u_l : \mathbb{R}^T \to \mathbb{R}$ in the same PMPs $\{s_i\}_{i=1}^{T} : \mathbb{R}^{N \times F} \to \mathbb{R}$, i.e.,

$$b_l(\boldsymbol{X}^{(F)}) = u_l(s_1(\boldsymbol{X}^{(F)}), \ldots, s_T(\boldsymbol{X}^{(F)})) \quad \forall l \in [L]. \tag{11}$$

By substituting Equations (10) and (11) into Equation (9), we deduce that $p$ is a doubly power-sum multi-symmetric polynomial. $\qquad\square$

Similarly to Lemma D.6, which enables the decomposition of $S_{N-1}$-invariant polynomials into a combination of polynomials that are $(S_F \times S_C)$-invariant, the following lemma enables the decomposition of $(S_{F-1} \times S_C)$-invariant polynomials into a combination of polynomials that are $(S_F \times S_C)$-invariant.

**Lemma D.9.** *Let* $p : \mathbb{R}^{N \times (F+C)} \to \mathbb{R}$ *be an* $(S_{F-1} \times S_C)$*-invariant polynomial of degree* $M$. *Then, there exist* $S_F \times S_C$*-invariant polynomials* $\{q_{\boldsymbol{\kappa}} : \mathbb{R}^{N \times (F+C)} \to \mathbb{R}\}_{\|\boldsymbol{\kappa}\|_1 \leq M}$ *such that*

$$p(\boldsymbol{X}) = \sum_{\|\boldsymbol{\kappa}\|_1 \leq M} \boldsymbol{X}_{:,1}^{\boldsymbol{\kappa}} q_{\boldsymbol{\kappa}}(\boldsymbol{X}).$$

*Here,* $\boldsymbol{\kappa}$ *are multi-indeces in* $\mathbb{N}_0^N$.

*Proof.* Following Definition D.7 with respect to the the domain $\mathbb{R}^{N \times ((F-1)+C)}$, we denote the doubly multi-symmetric power-sum polynomials (DMPs) $\mathbb{R}^{N \times ((F-1)+C)} \to \mathbb{R}$, taking the the variable $\boldsymbol{Z} \in \mathbb{R}^{N \times ((F-1)+C)}$, by

$$\hat{s}_t^{(F)}(\boldsymbol{Z}) = \sum_{f=1}^{F-1} \boldsymbol{Z}_{:,f}^{\boldsymbol{\alpha}^{(t)}} \quad \text{and} \quad \hat{s}_t^{(C)}(\boldsymbol{Z}) = \sum_{c=1}^{C} \boldsymbol{Z}_{:,F-1+c}^{\boldsymbol{\alpha}^{(t)}},$$

where these polynomials are parameterized by $t \in [T]$ for $T = \binom{M+N}{N}$. Similarly, following Definition D.7 with respect to the the domain $\mathbb{R}^{N \times (F+C)}$, we denote the DMPs which take the variable $\boldsymbol{Y} \in \mathbb{R}^{N \times (F+C)}$ by

$$s_t^{(F)}(\boldsymbol{Y}) = \sum_{f=1}^{F} \boldsymbol{Y}_{:,f}^{\boldsymbol{\alpha}^{(t)}} \quad \text{and} \quad s_t^{(C)}(\boldsymbol{Y}) = \sum_{c=1}^{C} \boldsymbol{Y}_{:,F+c}^{\boldsymbol{\alpha}^{(t)}},$$

where these polynomials are parameterized by the same enumeration $\boldsymbol{\alpha}^{(t)} \in \mathbb{N}_0^N$, $t \in [T]$, as the $\mathbb{R}^{N \times ((F-1)+C)} \to \mathbb{R}$ DMPs. Now, note that for each $t \in [T]$ we have

$$\hat{s}_t^{(F-1)}(\boldsymbol{X}_{:,2}, \ldots, \boldsymbol{X}_{:,F+C}) = s_t^{(F)}(\boldsymbol{X}) - \boldsymbol{X}_{:,1}^{\boldsymbol{\beta}^{(t)}} \text{ and } \hat{s}_t^{(C)}(\boldsymbol{X}_{:,2}, \ldots, \boldsymbol{X}_{:,F+C}) = s_t^{(C)}(\boldsymbol{X}). \quad (12)$$

Observe that we can expand the polynomial $p(\boldsymbol{X})$ in $\boldsymbol{X}_{:,1}$ as

$$p(\boldsymbol{X}) = \sum_{\|\boldsymbol{\gamma}\|_1 \leq M} \boldsymbol{X}_{:,1}^{\boldsymbol{\gamma}} p_{\boldsymbol{\gamma}}(\boldsymbol{X}_{:,2}, \ldots, \boldsymbol{X}_{:,F+C}), \quad (13)$$

where each $p_{\boldsymbol{\gamma}} : \mathbb{R}^{N \times (F-1+C)} \to \mathbb{R}$ is a polynomial of degree at most $M - \|\boldsymbol{\gamma}\|_1$. Since derivatives commute with permutations, and specifically with the representation of $S_{F-1} \times S_C$, any partial derivative of an $(S_{F-1} \times S_C)$-invariant polynomial is also an $(S_{F-1} \times S_C)$-invariant polynomial. Moreover, plugging $\boldsymbol{X}_{:,1} = 0$ into an $(S_{F-1} \times S_C)$-invariant polynomial $u : \mathbb{R}^{F \times C} \to \mathbb{R}$ gives an $(S_{F-1} \times S_C)$-invariant polynomial $u|_{\boldsymbol{X}_{:,1}=0} : \mathbb{R}^{N \times (F-1+C)} \to \mathbb{R}$. Hence, the polynomials

$$p_{\boldsymbol{\gamma}}(\boldsymbol{X}_{:,2}, \ldots, \boldsymbol{X}_{:,F+C}) = \frac{1}{\prod_{i=1}^N \gamma_i!} \left( \frac{\partial^{\|\boldsymbol{\gamma}\|_1}}{\partial \boldsymbol{X}_{:,1}^{\boldsymbol{\gamma}}} p \right)(0, \boldsymbol{X}_{:,2}, \ldots, \boldsymbol{X}_{:,F+C}), \quad \forall \|\boldsymbol{\gamma}\|_1 \leq M$$

are $(S_{F-1} \times S_C)$-invariant. Therefore, by Lemma D.8, each $p_{\boldsymbol{\gamma}}$ can be expressed as a polynomial $\hat{q}_{\boldsymbol{\gamma}} : \mathbb{R}^{2T} \to \mathbb{R}$ in the DMPs $\mathbb{R}^{N \times ((F-1)+C)} \to \mathbb{R}$, i.e.,

$$p_{\boldsymbol{\gamma}}(\boldsymbol{X}_{:,2}, \ldots, \boldsymbol{X}_{:,F+C}) = \hat{q}_{\boldsymbol{\gamma}} \Big( \hat{s}_1^{(F-1)}(\boldsymbol{X}_{:,2}, \ldots, \boldsymbol{X}_{:,F+C}), \ldots, \hat{s}_T^{(F-1)}(\boldsymbol{X}_{:,2}, \ldots, \boldsymbol{X}_{:,F+C}),$$
$$\hat{s}_1^{(C)}(\boldsymbol{X}_{:,2}, \ldots, \boldsymbol{X}_{:,F+C}), \ldots, \hat{s}_T^{(C)}(\boldsymbol{X}_{:,2}, \ldots, \boldsymbol{X}_{:,F+C}) \Big).$$

By Equation (12), $p_{\boldsymbol{\gamma}}$ can be written as

$$p_{\boldsymbol{\gamma}}(\boldsymbol{X}_{:,2}, \ldots, \boldsymbol{X}_{:,F+C}) = \hat{q}_{\boldsymbol{\gamma}} \Big( s_1^{(F)}(\boldsymbol{X}) - \boldsymbol{X}_{:,1}^{\boldsymbol{\beta}^{(1)}}, \ldots, s_T^{(F)}(\boldsymbol{X}) - \boldsymbol{X}_{:,1}^{\boldsymbol{\beta}^{(T)}}, \quad (14)$$
$$s_1^{(C)}(\boldsymbol{X}^{(C)}), \ldots, s_T^{(C)}(\boldsymbol{X}^{(C)}) \Big).$$

We can now expand $\hat{q}_{\boldsymbol{\gamma}}$ in the right-hand-side of Equation (14) in monomials of $\boldsymbol{X}_{:,1}$, and write

$$p_{\boldsymbol{\gamma}}(\boldsymbol{X}_{:,2}, \ldots, \boldsymbol{X}_{:,F+C}) = \sum_{\|\boldsymbol{\delta}\|_1 \leq M - \|\boldsymbol{\gamma}\|_1} \boldsymbol{X}_{:,1}^{\boldsymbol{\delta}} \widetilde{q}_{\boldsymbol{\gamma}, \boldsymbol{\delta}} \Big( s_1^{(F)}(\boldsymbol{X}), \ldots, s_T^{(F)}(\boldsymbol{X}), \quad (15)$$
$$s_1^{(C)}(\boldsymbol{X}), \ldots, s_T^{(C)}(\boldsymbol{X}) \Big)$$

where $\widetilde{q}_{\gamma,\delta}$ is an $S_F \times S_C$-invariant polynomial as it is a polynomial in the $S_F \times S_C$-invariant DMPs. When substituting Equation (15) into Equation (13), we obtain

$$p(\boldsymbol{X}) = \sum_{\|\boldsymbol{\gamma}\|_1 \leq M} \sum_{\|\boldsymbol{\delta}\|_1 \leq M - \|\boldsymbol{\gamma}\|_1} \boldsymbol{X}_{:,1}^{\gamma+\delta} \widetilde{q}_{\gamma,\delta} \left( s_1^{(F)}(\boldsymbol{X}), \ldots, s_T^{(F)}(\boldsymbol{X}), s_1^{(C)}(\boldsymbol{X}), \ldots, s_T^{(C)}(\boldsymbol{X}) \right).$$

Now, by performing the change of variables $\boldsymbol{\delta} = \boldsymbol{\kappa} - \boldsymbol{\gamma}$, we have that

$$p(\boldsymbol{X}) = \sum_{\|\boldsymbol{\kappa}\|_1 \leq M} \boldsymbol{X}_{:,1}^{\kappa} \sum_{\substack{\|\boldsymbol{\gamma}\|_1 \leq M \\ \|\boldsymbol{\kappa}-\boldsymbol{\gamma}\|_1 \leq M - \|\boldsymbol{\gamma}\|_1}} \widetilde{q}_{\gamma,\kappa-\gamma} \left( s_1^{(F)}(\boldsymbol{X}), \ldots, s_T^{(F)}(\boldsymbol{X}), s_1^{(C)}(\boldsymbol{X}), \ldots, s_T^{(C)}(\boldsymbol{X}) \right).$$

$$(16)$$

Lastly, by denoting

$$q_{\kappa}(\boldsymbol{X}) = \sum_{\substack{\|\boldsymbol{\gamma}\|_1 \leq M \\ \|\boldsymbol{\kappa}-\boldsymbol{\gamma}\|_1 \leq M - \|\boldsymbol{\gamma}\|_1}} \widetilde{q}_{\gamma,\kappa-\gamma} \left( s_1^{(F)}(\boldsymbol{X}), \ldots, s_T^{(F)}(\boldsymbol{X}), s_1^{(C)}(\boldsymbol{X}), \ldots, s_T^{(C)}(\boldsymbol{X}) \right),$$

Equation (16) can be written as

$$p(\boldsymbol{X}) = \sum_{\|\boldsymbol{\kappa}\|_1 \leq M} \boldsymbol{X}_{:,1}^{\kappa} q_{\kappa}(\boldsymbol{X}).$$

Here, note that as $q_{\kappa}(\boldsymbol{X})$ are defined as linear combinations of the $(S_F \times S_C)$-invariant polynomials $\widetilde{q}_{\gamma,\kappa-\gamma}$, so $q_{\kappa}(\boldsymbol{X})$ are also $(S_F \times S_C)$-invariant, concluding the proof. $\qquad\square$

Analogously, a similar lemma holds for $(S_F \times S_{C-1})$-invariant polynomials.

**Lemma D.10.** *Let $p : \mathbb{R}^{N \times (F+C)} \to \mathbb{R}$ be an $(S_F \times S_{C-1})$-invariant polynomial with degree $M$. Then, there exist $S_F \times S_C$-invariant polynomials $\{q_{\kappa} : \mathbb{R}^{N \times (F+C)} \to \mathbb{R}\}_{\|\boldsymbol{\kappa}\|_1 \leq M}$ such that*

$$p(\boldsymbol{X}) = \sum_{\|\boldsymbol{\kappa}\|_1 \leq M} \boldsymbol{X}_{:,F+1}^{\kappa} q_{\kappa}(\boldsymbol{X}).$$

### D.5 Supporting Universal Approximation Properties

In this subsection, we present universal approximation properties that support the proof of the universal approximation result Theorem 4.3.

First, we extend Lemma 4 from [38], which shows that continuous equivariant functions can be uniformly approximated by equivariant polynomials. Specifically, our extension generalizes the result from functions defined on the cube to functions defined on arbitrary compact $G$-invariant subsets.

**Lemma D.11.** *Let $G \leq S_L$. Let $\mathcal{K} \subset \mathbb{R}^L$ be a compact domain such that $\mathcal{K} = \cup_{g \in G} g\mathcal{K}$. Then the space of $G$-equivariant polynomials $p : \mathcal{K} \to \mathbb{R}^L$ is dense in the space of continuous $G$-equivariant functions $f : \mathcal{K} \to \mathbb{R}^L$ in $L_{\infty}(\mathcal{K})$.*

*Proof.* Let $\epsilon > 0$, and let $f : \mathcal{K} \to \mathbb{R}^L$ be a continuous $G$-equivariant function. For each output coordinate $f_l : \mathcal{K} \to \mathbb{R}$, the classical Stone–Weierstrass theorem guarantees the existence of a polynomial $p_l : \mathbb{R}^L \to \mathbb{R}$ such that

$$|f_l(\boldsymbol{X}) - p_l(\boldsymbol{X})| \leq \epsilon, \quad \text{for all } \boldsymbol{X} \in \mathcal{K}.$$

Define the polynomial $p : \mathbb{R}^L \to \mathbb{R}^L$ by

$$p(\boldsymbol{X}) = (p_1(\boldsymbol{X}), \ldots, p_L(\boldsymbol{X})) \in \mathbb{R}^L, \quad \forall \boldsymbol{X} \in \mathbb{R}^L.$$

To construct a $G$-equivariant approximating polynomial, we now introduce a symmetrization of $p$. Define $\widetilde{p} : \mathbb{R}^L \to \mathbb{R}^L$ by

$$\widetilde{p}(\boldsymbol{X}) = \frac{1}{|G|} \sum_{g \in G} g \cdot p(g^{-1} \cdot \boldsymbol{X}).$$

Let $\boldsymbol{X} \in \mathbb{R}^L$. Now, since $f$ is $G$-equivariant, we have

$$\|f(\boldsymbol{X}) - \widetilde{p}(\boldsymbol{X})\|_\infty = \left\| \frac{1}{|G|} \sum_{g \in G} g \cdot \left( f(g^{-1} \cdot \boldsymbol{X}) - p(g^{-1} \cdot \boldsymbol{X}) \right) \right\|_\infty$$

$$\leq \frac{1}{|G|} \sum_{g \in G} \left\| f(g^{-1} \cdot \boldsymbol{X}) - p(g^{-1} \cdot \boldsymbol{X}) \right\|_\infty \leq \epsilon.$$

Thus, $\widetilde{p}$ is a $G$-equivariant polynomial that uniformly approximates $f$, concluding the proof. $\qquad\square$

We mention the classical Universal Approximation Theorem (UAT) as it is later employed to show universality in Lemma 4.2, which has a central role in the proof of Theorem D.14.

**Theorem D.12** (Universal Approximation Theorem (UAT) [22])**.** *Let $\sigma : \mathbb{R} \to \mathbb{R}$ be a continuous, non-polynomial function. Then, for every $M, L \in \mathbb{N}$, compact $\mathcal{K} \subseteq \mathbb{R}^M$, continuous function $f : \mathcal{K} \to \mathbb{R}^L$ , and $\varepsilon > 0$, there exist $D \in \mathbb{N}, \boldsymbol{W}_1 \in \mathbb{R}^{D \times M}, \boldsymbol{b}_1 \in \mathbb{R}^D$ and $\boldsymbol{W}_2 \in \mathbb{R}^{L \times D}$ such that*

$$\sup_{\boldsymbol{x} \in \mathcal{K}} \|f(\boldsymbol{x}) - \boldsymbol{W}_2 \sigma \left( \boldsymbol{W}_1 \boldsymbol{x} + \boldsymbol{b}_1 \right)\| \leq \varepsilon. \tag{17}$$

Note that since all finite dimensional norms are equivalent, one can take any norm in (17).

We next study universality with respect to composition of functions. For that, we first define universal approximation rigorously.

**Definition D.13** (Universal Approximator)**.** *Let $D_1, D_2 \in \mathbb{N}$. A space of continuous functions $\mathcal{N}$ from a domain $Q_1 \subset \mathbb{R}^{D_1}$ to a domain $Q_2 \subset \mathbb{R}^{D_2}$ is said to be a* universal approximator *of another space of continuous functions $\mathcal{B}$ from $Q_1$ to $Q_2$, if for every $f \in \mathcal{B}$, every compact domain $C \subset \mathbb{R}^{D_1}$ such that $C \subset Q_1$, and every $\epsilon > 0$, there is a function $q \in \mathcal{N}$ such that for every $x \in C$*

$$\|f(x) - q(x)\|_\infty < \epsilon.$$

The following theorem is a simple approximation theoretic result.

**Theorem D.14.** *Let $L \in \mathbb{N}$, and $D_1, \ldots D_{L+1} \in \mathbb{N}$. Let $\mathcal{N}_l$ be a* universal approximator *of $\mathcal{B}_l$ for $l \in [L]$. Let the domain of the functions $\mathcal{B}_l$ be $Q_l \subset \mathbb{R}^{D_l}$ and the range be $Q_{l+1} \subset \mathbb{R}^{D_{l+1}}$, for $l \in [L]$. Consider the function spaces*

$$\mathcal{N} := \{\theta_L \circ \ldots \circ \theta_1 \mid \theta_l \in \mathcal{N}_l, \ l \in [L]\},$$

*and*

$$\mathcal{B} := \{f_L \circ \ldots \circ f_1 \mid f_l \in \mathcal{B}_l, \ l \in [L]\}.$$

*Then $\mathcal{N}$ is a universal approximator of $\mathcal{B}$.*

*Proof.* We prove this by induction on $L$.

**Base case** $L = 1$**.** In this case $\mathcal{N} = \mathcal{N}_1$ and $\mathcal{B} = \mathcal{B}_1$, so the claim follows directly from the assumption that $\mathcal{N}_1$ is a universal approximator of $\mathcal{B}_1$.

**Inductive Step.** Assume the result holds for $L - 1$ and consider a function

$$f = f_L \circ f_{L-1} \circ \cdots \circ f_1 \in \mathcal{B},$$

where $f_l \in \mathcal{B}_l$. Let $C_1 \subset Q_1$ be compact and let $\varepsilon > 0$. Define the compact set

$$C_L := (f_{L-1} \circ \cdots \circ f_1)(C_1) \subset Q_L.$$

By continuity of $f_L$ on $Q_L$ and compactness of $C_L$, $f_L$ is uniformly continuous on $C_L$. Thus, there exists $\delta > 0$ such that for all $y, y' \in C_L$ with $\|y - y'\| \leq \delta$, we have

$$\|f_L(y) - f_L(y')\| \leq \frac{\varepsilon}{2}. \tag{18}$$

By the induction hypothesis, there exist $\theta_1 \in \mathcal{N}_1, \ldots, \theta_{L-1} \in \mathcal{N}_{L-1}$ such that

$$\sup_{x \in C_1} \|(f_{L-1} \circ \cdots \circ f_1)(x) - (\theta_{L-1} \circ \cdots \circ \theta_1)(x)\| \leq \delta,$$

and by Equation (18), we get

$$\sup_{x \in C_1} \|f(x) - (f_L \circ \theta_{L-1} \circ \cdots \circ \theta_1)(x)\| \leq \frac{\varepsilon}{2}. \tag{19}$$

Define the compact set $\tilde{C}_L := (\theta_{L-1} \circ \cdots \circ \theta_1)(C_1)$. Define the compact set $C := C_L \cup \tilde{C}_L \subset Q_L$. Since $f_L \in \mathcal{B}_L$ and $\mathcal{N}_L$ is a universal approximator of $\mathcal{B}_L$, there exists $\theta_L \in \mathcal{N}_L$ such that

$$\sup_{y \in C} \|f_L(y) - \theta_L(y)\| \leq \frac{\varepsilon}{2},$$

which implies

$$\sup_{x \in C_1} \|(f_L \circ \theta_{L-1} \circ \cdots \circ \theta_1)(x) - (\theta_L \circ \theta_{L-1} \circ \cdots \circ \theta_1)(x)\| \leq \frac{\varepsilon}{2}. \tag{20}$$

Define $\theta = \theta_L \circ \theta_{L-1} \circ \cdots \circ \theta_1$. By the triangle inequality

$$\sup_{x \in C_1} \|f(x) - \theta(x)\| =$$

$$= \sup_{x \in C_1} \|f(x) - (f_L \circ \theta_{L-1} \circ \cdots \circ \theta_1)(x) + (f_L \circ \theta_{L-1} \circ \cdots \circ \theta_1)(x) - \theta(x)\|$$

$$\leq \sup_{x \in C_1} \|f(x) - (f_L \circ \theta_{L-1} \circ \cdots \circ \theta_1)(x)\| + \sup_{x \in C_1} \|(f_L \circ \theta_{L-1} \circ \cdots \circ \theta_1)(x) - \theta(x)\|$$

and by Equations (19) and (20), we get

$$\sup_{x \in C_1} \|f(x) - \theta(x)\| \leq \frac{\varepsilon}{2} + \frac{\varepsilon}{2} = \varepsilon.$$

This completes the induction step and the proof. $\qquad\square$

### D.6 Supporting Lemmas for $G$-descriptors

In this subsection, we introduce a special class of $G$-invariant polynomials, which are called $G$-*descriptors*. These polynomials are designed to distinguish between different orbits under the action of a subgroup $G \leq S_N$ of the symmetric group. We establish the existence of $(S_{F-1} \times S_C)$-descriptors and $S_{N-1}$-descriptors which we later use to express $(S_{N-1} \times S_{F-1} \times S_C)$-invariant polynomials defined over a symmetric compact domain.

Given a subgroup $G$ of the symmetric group, we next define the notion of a $G$-descriptor. Roughly speaking, a $G$-descriptor is a polynomial that maps different orbits under the representation of $G$ to different values.

**Definition D.15** ($G$-descriptor). *Let $G \leq S_M$ act on $\mathbb{R}^{M \times D}$ be changing the order of the rows, and let $\mathcal{K} \subseteq \mathbb{R}^{M \times D}$ such that $\mathcal{K} = \cup_{g \in G} g\mathcal{K}$. We call a $G$-invariant function $q : \mathcal{K} \to \mathbb{R}^L$ a $G$-descriptor, if it satisfies*

$$\forall \boldsymbol{X}, \boldsymbol{Y} \in \mathcal{K}, \quad q(\boldsymbol{X}) = q(\boldsymbol{Y}) \iff \exists g \in G : \boldsymbol{Y} = g \cdot \boldsymbol{X}.$$

Next, we cite Lemma 3 from [26], which shows the existence of a descriptor given a subgroup of the symmetric group.

**Lemma D.16** (Lemma 3 in [26]). *Let $G \leq S_M$ act on $\mathbb{R}^{M \times D}$ by changing the order of the rows. Then, there exists $L \in \mathbb{N}$ and a polynomial $G$-descriptor $u : \mathbb{R}^{M \times D} \to \mathbb{R}^L$.*

### D.7 Expressing $G$-Polynomial Descriptors Using Symmetric Polynomials

In the proof of Lemma D.24, we define polynomial descriptors. We then show that these descriptors can be written using symmetric polynomials, which then allows approximating them by TSNets. In this subsection, we show how certain polynomials descriptors can be written using symmetric polynomials.

We first define basic operations related to the constriction of symmetric polynomials.

**Definition D.17.** *Let $K, M, D \in \mathbb{N}$, let $T = T_{M,D} = \binom{M+D}{D}$ (i.e., the number of monomials in $\mathbb{R}^D$ up to degree $M$), and let $\{\boldsymbol{\alpha}^{(t)}\}_{t=1}^T$ be an enumeration of all multi-indices $\boldsymbol{\alpha}^{(t)} \in \mathbb{N}_0^D$ such that $\|\boldsymbol{\alpha}^{(t)}\|_1 \leq M$.*

- *The mapping $c^{(M,D)} : \mathbb{R}^D \to \mathbb{R}^T$ is the function that calculates all monomials up to degree $T$, namely, for $\boldsymbol{D} \in \mathbb{R}^D$*

$$c^{(M,D)}(\boldsymbol{D}) := \left( \boldsymbol{D}^{\boldsymbol{\alpha}^{(1)}}, \ldots, \boldsymbol{D}^{\boldsymbol{\alpha}^{(T)}} \right). \tag{21}$$

- *The mapping $b^{(K,M,D)} : \mathbb{R}^{K \times D} \to \mathbb{R}^{K \times T}$ is defined by*

$$\forall j \in [K],\ \boldsymbol{V} \in \mathbb{R}^{K \times D} : \quad b^{(K,M,D)}(\boldsymbol{V})_{j,:} = c^{(M,D)}(\boldsymbol{V}_{j,:}). \tag{22}$$

- *The mapping $t_{\mathrm{PMP}}^{(K,M,D)} : \mathbb{R}^{K \times T} \to \mathbb{R}^T$ is defined, for $\boldsymbol{E} \in \mathbb{R}^{K \times T}$, by*

$$t_{\mathrm{PMP}}^{(K,M,D)}(\boldsymbol{E}) = \sum_{k=1}^{K} \boldsymbol{E}_{k,:}. \tag{23}$$

- *For $F, C \in \mathbb{N}$ such that $F + C = K$, the mapping $t_{\mathrm{DMP}}^{(F+C,M,D)} : \mathbb{R}^{(F+C) \times T} \to \mathbb{R}^{2T}$ is defined, for $\boldsymbol{H} \in \mathbb{R}^{(F+C) \times T}$, by*

$$t_{\mathrm{DMP}}^{(F+C,M,D)}(\boldsymbol{H}) = (\sum_{f=1}^{F} \boldsymbol{H}_{f,:}, \sum_{c=1}^{C} \boldsymbol{H}_{F+c,:}). \tag{24}$$

**Remark D.18.** *Note that $t_{\mathrm{PMP}}^{(K,M,D)} \circ b^{(K,M,D)}$ is the mapping that computes the sequence of all PMPs up to degree $M$. Similarly, $t_{\mathrm{DMP}}^{(F+C,M,D)} \circ b^{(K,M,D)}$ computes the sequence of all DMPs up to degree $M$.*

**Lemma D.19.** *Let $K, M, D \in \mathbb{N}$ and $T = T_{M,D} = \binom{M+D}{D}$.*

1. *Let $q : \mathbb{R}^{K \times D} \to \mathbb{R}^L$ be an $S_{K-1}$-polynomial descriptor of degree $M$, namely,*

$$\forall \boldsymbol{Z}, \boldsymbol{Y} \in \mathbb{R}^{K \times D} : \quad \left( q(\boldsymbol{Z}) = q(\boldsymbol{Y}) \right) \Longleftrightarrow \left( \exists \sigma_{K-1} \in S_{K-1} : \boldsymbol{Z} = \sigma_{K-1} \cdot \boldsymbol{Y} \right).$$

*Then, there exist polynomials $U_{\boldsymbol{\beta}} : \mathbb{R}^T \to \mathbb{R}^L$, parameterized by $\boldsymbol{\beta} \in \mathbb{Z}_0^D$, such that for $\boldsymbol{V} \in \mathbb{R}^{K \times D}$*

$$q(\boldsymbol{V}) = \sum_{\|\boldsymbol{\beta}\|_1 \leq M} \boldsymbol{V}_{1,:}^{\boldsymbol{\beta}} W_{\boldsymbol{\beta}}(\boldsymbol{V}),$$

*where*

$$W_{\boldsymbol{\beta}} = U_{\boldsymbol{\beta}} \circ t_{\mathrm{PMP}}^{(K,M,D)} \circ b^{(K,M,D)}.$$

2. *Let $F, C \in \mathbb{N}$ such that $F + C = K$. Let $q : \mathbb{R}^{(F+C) \times D} \to \mathbb{R}^L$ be a $(S_{F-1} \times S_C)$-polynomial descriptor of degree $M$, namely, $\forall \boldsymbol{Z}, \boldsymbol{Y} \in \mathbb{R}^{(F+C) \times D}$ :*

$$\left( q(\boldsymbol{Z}) = q(\boldsymbol{Y}) \right)$$

$$\Updownarrow$$

$$\left( \exists (\sigma_{F-1}, \sigma_C) \in S_{F-1} \times S_C : \boldsymbol{Z} = (\sigma_{F-1}, \sigma_C) \cdot \boldsymbol{Y} \right).$$

*Then, there exist polynomials $U_{\boldsymbol{\beta}} : \mathbb{R}^{2T} \to \mathbb{R}^L$, parameterized by $\boldsymbol{\beta} \in \mathbb{Z}_0^D$, such that for $\boldsymbol{V} \in \mathbb{R}^{(F+C) \times D}$*

$$q(\boldsymbol{V}) = \sum_{\|\boldsymbol{\beta}\|_1 \leq M} \boldsymbol{V}_{1,:}^{\boldsymbol{\beta}} W_{\boldsymbol{\beta}}(\boldsymbol{V}),$$

*where*

$$W_{\boldsymbol{\beta}} = U_{\boldsymbol{\beta}} \circ t_{\mathrm{DMP}}^{(F+C,M,D)} \circ b^{(F+C,M,D)}.$$

The result is also immediately true when transposing the order of $F$ and $C$.

*Proof.* We prove 1, and note that 2 is shown analogously.

Recall the property defining the polynomial $q : \mathbb{R}^{K \times D} \to \mathbb{R}^L$

$$\forall \boldsymbol{Z}, \boldsymbol{Y} \in \mathbb{R}^{K \times D} : \quad \Big( q(\boldsymbol{Z}) = q(\boldsymbol{Y}) \Big) \Longleftrightarrow \Big( \exists \sigma_{K-1} \in S_{K-1} : \boldsymbol{Z} = \sigma_{K-1} \cdot \boldsymbol{Y} \Big).$$

Fix $l \in [L]$, and consider the $l$-th output dimension of $q$, namely $q_l : \mathbb{R}^{K \times D} \to \mathbb{R}$. By Lemma D.6, the $S_{K-1}$-invariant polynomial $q_l$ can be expressed as

$$q_l(\boldsymbol{V}) = \sum_{\|\boldsymbol{\beta}\|_1 \leq M_N} \boldsymbol{V}_{1,:}^{\boldsymbol{\beta}} w_{l,\boldsymbol{\beta}}(\boldsymbol{V}), \tag{25}$$

where $w_{l,\boldsymbol{\beta}} : \mathbb{R}^{K \times D} \to \mathbb{R}$ are $S_K$-invariant polynomials with degree at most $M$, and $\boldsymbol{\beta} \in \mathbb{N}_0^D$ are multi-indices with $\|\boldsymbol{\beta}\|_1 \leq M$. Next, we express (25) using PMPs (Definition D.4). The power-sum multi-symmetric polynomials (PMPs) $s_t : \mathbb{R}^{K \times D} \to \mathbb{R}$ map the variable $\boldsymbol{V} \in \mathbb{R}^{K \times D}$ to

$$s_t(\boldsymbol{V}) = \sum_{k=1}^K \boldsymbol{V}_{k,:}^{\boldsymbol{\alpha}^{(t)}},$$

where these polynomials are indexed by $t \in [T]$, in order to enumerate all multi-indices $\boldsymbol{\alpha}^{(t)} \in \mathbb{N}_0^D$ such that $\left\| \boldsymbol{\alpha}^{(t)} \right\|_1 \leq M$. Fix $\boldsymbol{\beta} \in \mathbb{N}_0^D$ with $\|\boldsymbol{\beta}\|_1 \leq M$. By Lemma D.5, the $S_K$-invariant polynomial $w_{l,\boldsymbol{\beta}}$ of (25) can be expressed as a polynomial $u_{l,\boldsymbol{\beta}} : \mathbb{R}^T \to \mathbb{R}$ in the PMPs $\{s_t\}_{t=1}^T$ by

$$w_{l,\boldsymbol{\beta}}(\boldsymbol{V}) = u_{l,\boldsymbol{\beta}} \left( s_1(\boldsymbol{V}), \ldots, s_T(\boldsymbol{V}) \right). \tag{26}$$

By Remark D.18, we can write $w_{l,\boldsymbol{\beta}}$ as a composition of the three mappings

$$w_{l,\boldsymbol{\beta}} = u_{l,\boldsymbol{\beta}} \circ t_{\mathrm{PMP}}^{(K,M,D)} \circ b^{(K,M,D)}, \tag{27}$$

where $t_{\mathrm{PMP}}^{(K,M,D)} \circ b^{(K,M,D)}$ is the mapping that computes the sequence of all PMPs (see Definition D.17).

We next express $q$ via the above construction. Define $U_{\boldsymbol{\beta}} : \mathbb{R}^T \to \mathbb{R}^L$, by

$$\forall l \in [L] : \quad (U_{\boldsymbol{\beta}})_l = u_{l,\boldsymbol{\beta}} \tag{28}$$

and similarly define $W_{\boldsymbol{\beta}} : \mathbb{R}^{K \times D} \to \mathbb{R}^L$ by

$$\forall l \in [L] : \quad (W_{\boldsymbol{\beta}})_l = w_{l,\boldsymbol{\beta}}. \tag{29}$$

Now, using Equations (27) to (29), we see that $W_{\boldsymbol{\beta}}$ can be written as the composition

$$W_{\boldsymbol{\beta}} = U_{\boldsymbol{\beta}} \circ t_{\mathrm{PMP}}^{(K,M,D)} \circ b^{(K,M,D)},$$

and by Equation (25), we have

$$q(\boldsymbol{V}) = \sum_{\|\boldsymbol{\beta}\|_1 \leq M} \boldsymbol{V}_{1,:}^{\boldsymbol{\beta}} W_{\boldsymbol{\beta}}(\boldsymbol{V}),$$

which shows 1.

To prove 2, we use Lemma D.9 instead of Lemma D.6 and follow an analogous construction. $\qquad \square$

## D.8 The Exclusion Set

As part of the proof that TSNets are universal approximators of continuous functions over symmetric compact domains (see Theorem 4.3), we need to define a descriptor that is injective on group orbits – the descriptor should assign the same value to inputs in the same orbit and different values otherwise. However, this injectivity can fail on a small subset of the domain. To still guarantee injectivity, we exclude this small subset from the domain of definition and refer to it as the *exclusion set*. This exclusion set has a specific form, as defined next.

**Definition D.20** (The exclusion set). *The set*

$$\mathcal{E} = \bigcup_{1 \leq f_1 < f_2 \leq F} \left\{ \boldsymbol{X} \in \mathbb{R}^{N \times (F+C)} \;\middle|\; \sum_{n=1}^{N} \boldsymbol{X}_{n,f_1} = \sum_{n=1}^{N} \boldsymbol{X}_{n,f_2} \right\} \bigcup$$

$$\bigcup_{1 \leq c_1 < c_2 \leq C} \left\{ \boldsymbol{X} \in \mathbb{R}^{N \times (F+C)} \;\middle|\; \sum_{n=1}^{N} \boldsymbol{X}_{n,F+c_1} = \sum_{n=1}^{N} \boldsymbol{X}_{n,F+c_2} \right\} \bigcup$$

$$\bigcup_{1 \leq n_1 < n_2 \leq N} \left\{ \boldsymbol{X} \in \mathbb{R}^{N \times (F+C)} \;\middle|\; \sum_{j=1}^{F+C} \boldsymbol{X}_{n_1,j} = \sum_{j=1}^{F+C} \boldsymbol{X}_{n_2,j} \right\}$$

*is called the* exclusion set corresponding to $\mathbb{R}^{N \times (F+C)}$.

The exclusion set $\mathcal{E}$ is a finite union of linear subspaces of co-dimension one. We consider input matrices $\boldsymbol{X} \in \mathbb{R}^{N \times (F+C)}$, where $\boldsymbol{X}^{(F)}$ correspond to continuous real-valued features and $\boldsymbol{X}^{(C)}$ encode one-hot vectors with entries in $\{0, 1\}$, assigning probability one to exactly one class and zero to the others in each row. The real-valued features $\boldsymbol{X}^{(F)}$ vary continuously in $\mathbb{R}^F$. The set of features that lie inside the exclusion set $\mathcal{E}$ is meagre with respect to the standard topology of the feature space (more specifically, it is nowhere dense). Hence, being outside $\mathcal{E}$ is the generic case in the topological sense. From a probabilistic perspective, if the features are sampled from a joint distribution that is continuous with respect to the Lebesgue measure (in the Radon–Nikodym sense), then the features belong to $\{\boldsymbol{X}^{(F)} \mid \boldsymbol{X} \in \mathcal{E}\}$ in probability zero. Hence, for the features to belong to $\mathcal{E}$ is an event that almost surely does not occur. With regard to the labels, the exclusion set $\mathcal{E}$ rules out the special scenario in which multiple classes have exactly the same number of examples. If labels are samples from a reasonable joint probability space, where labels are not constrained to be exactly balanced, such an even typically occurs in low probability. To summarize, from both the feature and label perspectives, the data typically lies outside the exclusion set $\mathcal{E}$, and hence considering a domain $\mathcal{K}$ disjoint from $\mathcal{E}$ is reasonable. A similar philosophy regarding an exclusion set is adopted by [26].

## D.9 TSNets With Output Projected to $(1, 1)$ and $(1, F + 1)$

To prove that our invariant neural networks $\text{TSNet}_{\text{LI}}$ are universal approximators of symmetry preserving continuous functions, we first prove that $\text{TSNet}_{\text{Eq}}$ is a universal approximator of equivariant continuous function. For that, we first define spaces of neural network which are $\text{TSNet}_{\text{Eq}}$ with outputs restricted to the entries $(1, 1)$ and $(1, F + 1)$.

**Definition D.21** (The $\text{TSNet}^{(F)}$, $\text{TSNet}^{(C)}$, $\text{TSNet}^{(NF)}$ and $\text{TSNet}^{(NC)}$ Architectures).

- *A $\text{TSNet}^{(F)}$ is any MLP* $\theta : \mathbb{R}^{K_1 \times N \times (F+C)} \to \mathbb{R}^{N \times K_2}$ *of the following form. There exists a $\text{TSNet}_{\text{Eq}}$* $\phi : \mathbb{R}^{K_1 \times N \times (F+C)} \to \mathbb{R}^{K_2 \times N \times (F+C)}$ *such that for every* $(\boldsymbol{X}, \boldsymbol{Y}) \in \mathbb{R}^{K_1 \times N \times (F+C)}$

$$\theta(\boldsymbol{X}, \boldsymbol{Y}) = \big(\phi(\boldsymbol{X}, \boldsymbol{Y})\big)_{:,:,1}.$$

- *A $\text{TSNet}^{(C)}$ is any MLP* $\theta : \mathbb{R}^{K_1 \times N \times (F+C)} \to \mathbb{R}^{N \times K_2}$ *such that there exists a $\text{TSNet}_{\text{Eq}}$* $\phi : \mathbb{R}^{K_1 \times N \times (F+C)} \to \mathbb{R}^{K_2 \times N \times (F+C)}$ *such that*

$$\theta(\boldsymbol{X}, \boldsymbol{Y}) = \big(\phi(\boldsymbol{X}, \boldsymbol{Y})\big)_{:,:,F+1}.$$

- *Similarly, a $\text{TSNet}^{(NF)}$ is any MLP* $\theta : \mathbb{R}^{K_1 \times N \times (F+C)} \to \mathbb{R}^{K_2}$ *such that*

$$\theta(\boldsymbol{X}, \boldsymbol{Y}) = \big(\phi(\boldsymbol{X}, \boldsymbol{Y})\big)_{:,1,1},$$

*and a $\text{TSNet}^{(NC)}$ is any MLP* $\theta : \mathbb{R}^{K_1 \times N \times (F+C)} \to \mathbb{R}^{K_2}$ *such that*

$$\theta(\boldsymbol{X}, \boldsymbol{Y}) = \big(\phi(\boldsymbol{X}, \boldsymbol{Y})\big)_{:,1,F+1}.$$

Note that $\text{TSNet}^{(F)}$ are invariant to $(\sigma_N \times \sigma_{F-1} \times \sigma_C)$, $\text{TSNet}^{(F)}$ to $(\sigma_N \times \sigma_F \times \sigma_{C-1})$, $\text{TSNet}^{(NF)}$ to $(\sigma_{N-1} \times \sigma_{F-1} \times \sigma_C)$, and $\text{TSNet}^{(NC)}$ to $(\sigma_{N-1} \times \sigma_F \times \sigma_{C-1})$.

**Remark D.22.** *The entries $[:, 1, 1]$ are special for $TSNet^{(NF)}$, and break the $S_N$ and $S_F$ symmetries. Indeed, all "self" transformations in the last linear layer of a $TSNet^{(NF)}$ can be written in terms of $[:, 1, 1]$, e.g.,*

$$[F_{\text{self}} \xrightarrow{N_{\text{self}}} F](\boldsymbol{X}, \boldsymbol{Y})_{1,1} = \boldsymbol{X}_{1,1}.$$

*Similarly, the entires $[:, 1, F+1]$ are special for $TSNet^{(NC)}$, $[:, :, 1]$ for $TSNet^{(F)}$, and $[:, :, F+1]$ for $TSNet^{(C)}$.*

Given two networks, $\theta^{(NF)} \in TSNet^{(NF)}$ and $\theta^{(NC)} \in TSNet^{(NC)}$, one can recover a corresponding $TSNet_{\text{Eq}}$ as follows.

**Lemma D.23.**

- *The MLP $\theta$ is a $TSNet_{\text{Eq}}$ if and only if there is a $TSNet^{(F)}$ $\theta^{(F)}$ and a $TSNet^{(C)}$ $\theta^{(C)}$ such that*

$$\forall f \in [F]: \quad \theta(\boldsymbol{X}, \boldsymbol{Y})_{:,:,f} = \theta^{(NF)}\big(e, (1f), e) \cdot (\boldsymbol{X}, \boldsymbol{Y})\big)$$

  *and*

$$\forall c \in [C]: \quad \theta(\boldsymbol{X}, \boldsymbol{Y})_{:,:,F+c} = \theta^{(NC)}\big(e, e, (1c)) \cdot (\boldsymbol{X}, \boldsymbol{Y})\big),$$

  *for $(1f) \in S_F$ (the permutation that swaps only 1 with $f$) and $(1c) \in S_C$.*

- *The MLP $\theta$ is a $TSNet_{\text{Eq}}$ if and only if there is a $TSNet^{(NF)}$ $\theta^{(NF)}$ and a $TSNet^{(NC)}$ $\theta^{(NC)}$ such that*

$$\forall n \in [N], f \in [F]: \quad \theta(\boldsymbol{X}, \boldsymbol{Y})_{:,n,f} = \theta^{(NF)}\big(((1n), (1f), e) \cdot (\boldsymbol{X}, \boldsymbol{Y})\big)$$

  *and*

$$\forall n \in [N], c \in [C]: \quad \theta(\boldsymbol{X}, \boldsymbol{Y})_{:,n,F+c} = \theta^{(NC)}\big(((1n), e, (1c)) \cdot (\boldsymbol{X}, \boldsymbol{Y})\big),$$

  *for $(1n) \in S_N$, $(1f) \in S_F$, and $(1c) \in S_C$.*

*Proof.* We prove 2, and note that 1 is analogous. Suppose that $\theta$ is defined via the above formulas given $\theta^{(NF)} = \big(\phi(\cdot)\big)_{:1,1}$ in $TSNet^{(NF)}$ and $\theta^{(NC)} = \big(\psi(\cdot)\big)_{:1,F+1}$ in $TSNet^{(NC)}$. For any $n \in [N]$ and $f \in [F]$ we have

$$\theta(\boldsymbol{X}, \boldsymbol{Y})_{:,n,f} = \theta^{(NF)}\big(((1n), (1f), e) \cdot (\boldsymbol{X}, \boldsymbol{Y})\big) = \Big(\phi\big(((1n), (1f), e) \cdot (\boldsymbol{X}, \boldsymbol{Y})\big)\Big)_{:,1,1}$$

$$= \Big(((1n), (1f), e) \cdot \phi\big((\boldsymbol{X}, \boldsymbol{Y})\big)\Big)_{:,1,1} = \phi(\boldsymbol{X}, \boldsymbol{Y})_{:,n,f}.$$

Hence, $\theta(\cdot)_{:,:,1:F} = \phi(\cdot)_{:,:,1:F} : \mathbb{R}^{N \times (F+C)} \to \mathbb{R}^{N \times F}$, so $\theta(\cdot)_{:,:,1:F}$ is equivariant to $\sigma_N \times \sigma_F \times \{e\}$ and invariant to $\{e\} \times \{e\} \times \sigma_C$. Similarly, $\theta(\cdot)_{:,:,F+1:F+C} = \psi(\cdot)_{:,:,F+1:F+C} : \mathbb{R}^{N \times (F+C)} \to \mathbb{R}^{N \times F}$ is equivariant to $\sigma_N \times \{e\} \times \sigma_F$ and invariant to $\{e\} \times S_F \times \{e\}$. Together, $\theta$ is equivariant to $S_N \times S_F \times S_C$.

The other direction is trivial by taking $\theta^{(NF)} = \big(\theta(\cdot)\big)_{:1,1} \in TSNet^{(NF)}$ and $\theta^{(NC)} = \big(\theta(\cdot)\big)_{:1,F+1} \in TSNet^{(NC)}$. $\qquad\square$

## D.10 Existence of $\mathbb{R}^{N \times (F+C)} \setminus \mathcal{E}$ descriptors that can be approximated by $TSNets^{(NF)}$ and $TSNets^{(NC)}$

The key step in the proof of the universal approximation theorem Lemma 4.2 is constructing a polynomial descriptor that can be approximated by TSNets. The following lemma established this.

**Lemma D.24.** *Let $N, F, C \in \mathbb{N}$ and consider the triple-symmetric group $S_N \times S_F \times S_C$ and its standard representation in $\mathbb{R}^{N \times (F+C)}$ and restrictions to $S_{N-1} \times S_{F-1} \times S_C$ and $S_{N-1} \times S_F \times S_{C-1}$ (see Subsecction D.2). Let $\mathcal{E}$ be the exclusion set corresponding to $\mathbb{R}^{N \times (F+C)}$ (Definition D.20). Then, there exist $L \in \mathbb{N}$ and two spaces $\mathcal{B}^{(NF)}$ and $\mathcal{B}^{(NC)}$ of polynomials from $\mathbb{R}^{N \times (F+C)} \setminus \mathcal{E}$ to $\mathbb{R}^L$ such that $TSNet^{(NF)}$ is a universal approximator of $\mathcal{B}^{(NF)}$ and $TSNet^{(NC)}$ is a universal approximator of $\mathcal{B}^{(NC)}$, with the following two properties.*

1. *There exists an $S_{N-1} \times S_{F-1} \times S_C$-polynomial descriptor $Q^{(NF)} : \left(\mathbb{R}^{N \times (F+C)} \setminus \mathcal{E}\right) \to \mathbb{R}^L$ in $\mathcal{B}^{(NF)}$.*

2. *There exists an $S_{N-1} \times S_F \times S_{C-1}$-polynomial descriptor $Q^{(NC)} : \left(\mathbb{R}^{N \times (F+C)} \setminus \mathcal{E}\right) \to \mathbb{R}^L$ in $\mathcal{B}^{(NC)}$.*

Note that, for example, 1. of Lemma D.24 means that $\forall \boldsymbol{Z}, \boldsymbol{Y} \in \mathbb{R}^{N \times (F+C)} \setminus \mathcal{E}$ :

$$Q^{(NF)}(\boldsymbol{Z}) = Q^{(NF)}(\boldsymbol{Y})$$

$$\Updownarrow \tag{30}$$

$$\exists (\sigma_{N-1}, \sigma_{F-1}, \sigma_C) \in S_{N-1} \times S_{F-1} \times S_C : \quad \boldsymbol{Y} = (\sigma_{N-1}, \sigma_{F-1}, \sigma_C) \cdot \boldsymbol{Z}.$$

*Proof of of Lemma D.24.* We prove 1. of Lemma D.24, and note that the proof of 2. is analogous.

**(1) A $(S_{F-1} \times S_C)$-descriptor $Q^{(F)}$.** By Lemma D.16, there exists an $S_{F-1} \times S_C$-polynomial descriptor (see Definition D.15) $q^{(F)} : \mathbb{R}^{(F+C) \times 2} \to \mathbb{R}^{L_F}$ with degree $M_F$, namely, a polynomial such that

$$\forall \boldsymbol{Z}, \boldsymbol{Y} \in \mathbb{R}^{(F+C) \times 2} : \tag{31}$$

$$\left(q^{(F)}(\boldsymbol{Z}) = q^{(F)}(\boldsymbol{Y})\right) \iff \left(\exists (\sigma_{F-1}, \sigma_C) \in S_{F-1} \times S_C : \boldsymbol{Z} = (\sigma_{F-1}, \sigma_C) \cdot \boldsymbol{Y}\right).$$

We define a polynomial $Q^{(F)} : \mathbb{R}^{N \times (F+C)} \setminus \mathcal{E} \to \mathbb{R}^{N \times L_F}$ as follows. Given $\boldsymbol{X} \in \mathbb{R}^{N \times (F+C)} \setminus \mathcal{E}$, the polynomial is defined by taking the row vector $\sum_{i=1}^N \boldsymbol{X}_{i,:} \in \mathbb{R}^{F+C}$ and appending it to each row of $\boldsymbol{X}$ forming a new dimension and creating the intermediate tensor

$$\hat{\boldsymbol{X}} = [\boldsymbol{X}, \boldsymbol{1}_N \sum_{i=1}^N \boldsymbol{X}_{i,:}] \in \mathbb{R}^{N \times (F+C) \times 2}. \tag{32}$$

To define $Q^{(F)}$, the function $q^{(F)}$ is then applied on $\hat{\boldsymbol{X}}$ via

$$\forall n \in [N], \quad Q^{(F)}(\boldsymbol{X})_{n,:} := q^{(F)}(\hat{\boldsymbol{X}}_{n,:,:}) = q^{(F)}([\boldsymbol{X}_{n,:}, \sum_{i=1}^N \boldsymbol{X}_{i,:}]). \tag{33}$$

By construction, this makes $Q^{(F)}$ an $S_N$-equivariant polynomial. Note that given a permutation $(\sigma_{F-1}, \sigma_C) \in S_{F-1} \times S_C$, for every $\boldsymbol{Z}, \boldsymbol{Y} \in \mathbb{R}^{N \times (F+C)}$, we have

$$\left(\forall n : \boldsymbol{Z}_{n,:} = (\sigma_{F-1}, \sigma_C) \cdot \boldsymbol{Y}_{n,:}\right)$$

$$\iff \left(\forall n : \left[\boldsymbol{Z}, \boldsymbol{1}_N \sum_{i=1}^N \boldsymbol{Z}_{i,:}\right]_{n,:,:} = \left[(\sigma_{F-1}, \sigma_C) \cdot \boldsymbol{Y}, (\sigma_{F-1}, \sigma_C) \cdot \boldsymbol{1}_N \sum_{i=1}^N \boldsymbol{Y}_{i,:}\right]_{n,:,:}\right)$$

$$\iff \left(\forall n : \hat{\boldsymbol{Z}}_{n,:,:} = (\sigma_{F-1}, \sigma_C) \cdot \hat{\boldsymbol{Y}}_{n,:,:}\right).$$

Thus, we obtained that

$$\left(\forall n : \boldsymbol{Z}_{n,:} = (\sigma_{F-1}, \sigma_C) \cdot \boldsymbol{Y}_{n,:}\right) \iff \left(\forall n : \hat{\boldsymbol{Z}}_{n,:,:} = (\sigma_{F-1}, \sigma_C) \cdot \hat{\boldsymbol{Y}}_{n,:,:}\right). \tag{34}$$

We now prove that $Q^{(F)}$ is an $S_N$-equivariant polynomial that satisfies

$$\forall \boldsymbol{Z}, \boldsymbol{Y} \in \mathbb{R}^{N \times (F+C)} \setminus \mathcal{E} : \tag{35}$$

$$\left(Q^{(F)}(\boldsymbol{Z}) = Q^{(F)}(\boldsymbol{Y})\right) \iff \left(\exists (\sigma_{F-1}, \sigma_C) \in S_{F-1} \times S_C : \boldsymbol{Y} = (\sigma_{F-1}, \sigma_C) \cdot \boldsymbol{Z}\right).$$

**Direction $\Leftarrow$ of Equation (35).** Assume there exists $(\sigma_{F-1}, \sigma_C) \in S_{F-1} \times S_C$, such that $\boldsymbol{Y} = (\sigma_{F-1}, \sigma_C) \cdot \boldsymbol{Z} \in \mathbb{R}^{N \times (F+C)}$. By definition, the action of $(\sigma_{F-1}, \sigma_C)$ permutes the columns of $\boldsymbol{Z}$ identically across all rows, namely,

$$\forall n \in [N] : \quad \boldsymbol{Y}_{n,:} = (\sigma_{F-1}, \sigma_C) \cdot \boldsymbol{Z}_{n,:}$$

and by Equation (34)
$$\forall n \in [N]: \quad \hat{Y}_{n,:,:} = (\sigma_{F-1}, \sigma_C) \cdot \hat{Z}_{n,:,:}. \tag{36}$$
By Equations (31) and (36), we obtain
$$\forall n \in [N]: \quad q^{(F)}(\hat{Z}_{n,:,:}) = q^{(F)}(\hat{Y}_{n,:,:}),$$
and by the definition of $Q^{(F)}$ in Equation (33), we have that $Q^{(F)}(\boldsymbol{Y}) = Q^{(F)}(\boldsymbol{Z})$.

**Direction $\Rightarrow$ of Equation (35).** We now show the opposite direction. Assume that $Q^{(F)}(\boldsymbol{Y}) = Q^{(F)}(\boldsymbol{Z})$ for some $\boldsymbol{Y}, \boldsymbol{Z} \in \mathbb{R}^{N \times (F+C)} \setminus \mathcal{E}$. By the definition of $Q^{(F)}$ in Equation (33), we have that
$$\forall n \in [N]: \quad q^{(F)}(\hat{Z}_{n,:,:}) = q^{(F)}(\hat{Y}_{n,:,:}),$$
and by Equation (31), the equality implies the existence of a permutation $(\sigma_{F-1}^{(n)}, \sigma_C^{(n)}) \in S_{F-1} \times S_C$ for each $n \in [N]$, such that
$$\hat{Y}_{n,:,:} = (\sigma_{F-1}^{(n)}, \sigma_C^{(n)}) \cdot \hat{Z}_{n,:,:}. \tag{37}$$
Next, we would like to show that
$$\forall n_1, n_2 \in [N]: \quad (\sigma_{F-1}^{(n_1)}, \sigma_C^{(n_1)}) = (\sigma_{F-1}^{(n_2)}, \sigma_C^{(n_2)}).$$
For that, consider two indices $n_1 \neq n_2 \in [N]$. By Equation (37) and the definition of $\hat{Z}$ and $\hat{Y}$ (Equation (32)), we get
$$\mathbf{1}_N \sum_{i=1}^{N} \boldsymbol{Y}_{i,:} = (\sigma_{F-1}^{(n_1)}, \sigma_C^{(n_1)}) \cdot \mathbf{1}_N \sum_{i=1}^{N} \boldsymbol{Z}_{i,:} \text{ and } \mathbf{1}_N \sum_{i=1}^{N} \boldsymbol{Y}_{i,:} = (\sigma_{F-1}^{(n_2)}, \sigma_C^{(n_2)}) \cdot \mathbf{1}_N \sum_{i=1}^{N} \boldsymbol{Z}_{i,:}. \tag{38}$$
Since $\mathbf{1}_N \sum_{i=1}^{N} \boldsymbol{Z}_{i,:}$ has equal rows (and so does $\mathbf{1}_N \sum_{i=1}^{N} \boldsymbol{Y}_{i,:}$) and $S_{F-1} \times S_C$ permutes all columns simultaneously, Equation (38) can be written as
$$(\sigma_{F-1}^{(n_1)}, \sigma_C^{(n_1)}) \cdot \sum_{i=1}^{N} \boldsymbol{Z}_{i,:} = (\sigma_{F-1}^{(n_2)}, \sigma_C^{(n_2)}) \cdot \sum_{i=1}^{N} \boldsymbol{Z}_{i,:}. \tag{39}$$
Recall that $\boldsymbol{Y}, \boldsymbol{Z} \in \mathbb{R}^{N \times (F+C)} \setminus \mathcal{E}$. Hence the first $F$ entries of $\sum_{i=1}^{N} \boldsymbol{Y}_{i,:}$ and $\sum_{i=1}^{N} \boldsymbol{Z}_{i,:}$, each have $F$ distinct elements. Moreover, the last $C$ entries of $\sum_{i=1}^{N} \boldsymbol{Y}_{i,:}$ and $\sum_{i=1}^{N} \boldsymbol{Z}_{i,:}$, each have $C$ distinct elements. Therefore, the only way that Equation (39) holds is when $\sigma_{F-1}^{(n_1)} = \sigma_{F-1}^{(n_2)}$ and $\sigma_C^{(n_1)} = \sigma_C^{(n_2)}$. Thus, all row–wise permutations coincide, i.e.
$$\exists (\sigma_{F-1}, \sigma_C) \in S_{F-1} \times S_C, \, \forall i \in [N]: \quad (\sigma_{F-1}^{(i)}, \sigma_C^{(i)}) = (\sigma_{F-1}, \sigma_C),$$
and consequently by Equations (34) and (37),
$$\forall i \in [N]: \quad \boldsymbol{Y}_{i,:} = (\sigma_{F-1}, \sigma_C) \cdot \boldsymbol{Z}_{i,:}.$$
Thus, $Q^{(F)} : \mathbb{R}^{N \times (F+C)} \rightarrow \mathbb{R}^{N \times L_F}$ is an $S_N$-equivariant polynomial that satisfies Equation (35).

Similarly, by Lemma D.16, there exists an $S_F \times S_{C-1}$-polynomial descriptor $q^{(C)} : \mathbb{R}^{(F+C) \times 2} \rightarrow \mathbb{R}^{L_C}$ with degree $M_C$, namely, a polynomial such that

$\forall \boldsymbol{Z}, \boldsymbol{Y} \in \mathbb{R}^{(F+C) \times 2}$:
$$\left( q^{(C)}(\boldsymbol{Z}) = q^{(C)}(\boldsymbol{Y}) \right) \iff \left( \exists (\sigma_F, \sigma_{C-1}) \in S_F \times S_{C-1}: \boldsymbol{Z} = (\sigma_F, \sigma_{C-1}) \cdot \boldsymbol{Y} \right).$$

We define a polynomial $Q^{(C)} : \mathbb{R}^{N \times (F+C)} \setminus \mathcal{E} \rightarrow \mathbb{R}^{N \times L_C}$ similarly to $Q^{(C)}$. Given $\boldsymbol{X} \in \mathbb{R}^{N \times (F+C)} \setminus \mathcal{E}$, we form the intermediate tensor, as in Equation (32)
$$\hat{\boldsymbol{X}} = [\boldsymbol{X}, \mathbf{1}_N \sum_{i=1}^{N} \boldsymbol{X}_{i,:}] \in \mathbb{R}^{N \times (F+C) \times 2}.$$

To define $Q^{(C)}$, the function $q^{(C)}$ is then applied on $\hat{\boldsymbol{X}}$ via
$$\forall n \in [N], \quad Q^{(C)}(\boldsymbol{X})_{n,:} := q^{(C)}(\hat{\boldsymbol{X}}_{n,:,:}) = q^{(C)}([\boldsymbol{X}_{n,:}, \sum_{i=1}^{N} \boldsymbol{X}_{i,:}]). \tag{40}$$

Then $Q^{(C)}$ is an $S_N$-equivariant polynomial that satisfies the analogue of Equation (35)

$$\forall \mathbf{Z}, \mathbf{Y} \in \mathbb{R}^{N \times (F+C)} \setminus \mathcal{E}:$$
$$\left(Q^{(C)}(\mathbf{Z}) = Q^{(C)}(\mathbf{Y})\right) \iff \left(\exists (\sigma_F, \sigma_{C-1}) \in S_F \times S_{C-1}: \mathbf{Y} = (\sigma_F, \sigma_{C-1}) \cdot \mathbf{Z}\right).$$

**(2) A $(S_{N-1} \times S_{F-1} \times S_C)$-descriptor $Q^{(NF)}$.** By Lemma D.16, there exists an $S_{N-1}$-invariant polynomial $q^{(N)}: \mathbb{R}^{N \times 3} \to \mathbb{R}^{L_N}$ of degree $M_N$, such that

$$\forall \mathbf{Z}, \mathbf{Y} \in \mathbb{R}^{N \times 3}: \quad \left(q^{(N)}(\mathbf{Z}) = q^{(N)}(\mathbf{Y})\right) \iff \left(\exists \sigma_{N-1} \in S_{N-1}: \mathbf{Z} = \sigma_{N-1} \cdot \mathbf{Y}\right). \quad (41)$$

We define a polynomial $Q^{(NF)}: \mathbb{R}^{N \times (F+C)} \setminus \mathcal{E} \to \mathbb{R}^{L_N \times L_F}$, where $L_F$ is the output dimension of the polynomial descriptor $q^{(F)}$ given in Equation (31), as follows. Given an input $\mathbf{X} \in \mathbb{R}^{N \times (F+C)} \setminus \mathcal{E}$, the output $Q^{(NF)}(\mathbf{X})$ is defined via the following composition of computations.

  (i) Compute $Q^{(F)}(\mathbf{X}) \in \mathbb{R}^{N \times L_F}$, as defined in Equation (33).

  (ii) Consider the column vectors $\sum_{f=1}^{F} \mathbf{X}_{:,f} \in \mathbb{R}^N$ and $\sum_{c=1}^{C} \mathbf{X}_{:,F+c} \in \mathbb{R}^N$. Append these vectors to $Q_F(\mathbf{X})$ forming a new dimension. Namely, construct the intermediate tensor

$$\widetilde{\mathbf{X}} = [Q^{(F)}(\mathbf{X}), \sum_{f=1}^{F} \mathbf{X}_{:,f} \mathbf{1}_{L_F}^\top, \sum_{c=1}^{C} \mathbf{X}_{:,F+c} \mathbf{1}_{L_F}^\top] \in \mathbb{R}^{N \times L_F \times 3}. \quad (42)$$

  (iii) Apply the function $q^{(N)}$ on $\widetilde{\mathbf{X}}$ to define $Q^{(NF)}(\mathbf{X})$ via

$$\forall l \in [L_F]: \quad Q^{(NF)}(\mathbf{X})_{:,l} := q^{(N)}(\widetilde{\mathbf{X}}_{:,l,:}) \quad (43)$$
$$= q^{(N)}([Q^{(F)}(\mathbf{X})_{:,l}, \sum_{f=1}^{F} \mathbf{X}_{:,f}, \sum_{c=1}^{C} \mathbf{X}_{:,F+c}]) \in \mathbb{R}^{L_N}.$$

By construction, $Q^{(NF)}$ is a polynomial from $\mathbb{R}^{N \times (F+C)} \setminus \mathcal{E}$ to $\mathbb{R}^{L_N \times L_F}$. We now prove that $Q^{(NF)}$ satisfies $\forall \mathbf{Z}, \mathbf{Y} \in \mathbb{R}^{N \times (F+C)} \setminus \mathcal{E}:$

$$Q^{(NF)}(\mathbf{Z}) = Q^{(NF)}(\mathbf{Y})$$
$$\Updownarrow \quad (44)$$
$$\exists (\sigma_{N-1}, \sigma_{F-1}, \sigma_C) \in S_{N-1} \times S_{F-1} \times S_C: \quad \mathbf{Y} = (\sigma_{N-1}, \sigma_{F-1}, \sigma_C) \cdot \mathbf{Z}.$$

**Direction $\Uparrow$ of Equation (44).** Let $\mathbf{Y}, \mathbf{Z} \in \mathbb{R}^{N \times (F+C)} \setminus \mathcal{E}$. Suppose there exists $(\sigma_{N-1}, \sigma_{F-1}, \sigma_C) \in S_{N-1} \times S_{F-1} \times S_C$ such that $\mathbf{Y} = (\sigma_{N-1}, \sigma_{F-1}, \sigma_C) \cdot \mathbf{Z}$. Recall that $Q^{(F)}$ is $S_N$-equivariant. This, together with Equation (35), gives

$$\forall l \in [L_F]: \quad Q^{(F)}(\mathbf{Y})_{:,l} = Q^{(F)}((\sigma_{N-1}, e, e) \cdot (e, \sigma_{F-1}, \sigma_C) \cdot \mathbf{Z}) \quad (45)$$
$$= \sigma_{N-1} \cdot Q^{(F)}((e, \sigma_{F-1}, \sigma_C) \cdot \mathbf{Z}) = \sigma_{N-1} \cdot Q^{(F)}(\mathbf{Z})_{:,l}.$$

Substituting Equation (45) into the definition of $\widetilde{\mathbf{Y}}$ in Equation (42), we get

$$\widetilde{\mathbf{Y}} = [\sigma_{N-1} \cdot Q^{(F)}(\mathbf{Z}), \sum_{f=1}^{F} \mathbf{Y}_{:,f} \mathbf{1}_{L_F}^\top, \sum_{c=1}^{C} \mathbf{Y}_{:,F+c} \mathbf{1}_{L_F}^\top].$$

Since $\mathbf{Y} = (\sigma_{N-1}, \sigma_{F-1}, \sigma_C) \cdot \mathbf{Z}$, and by the fact that summation is invariant to the permutations $\sigma_{F-1}$ and $\sigma_C$, we obtain

$$\widetilde{\mathbf{Y}} = \sigma_{N-1} \cdot [Q^{(F)}(\mathbf{Z}), \sum_{f=1}^{F} \mathbf{Z}_{:,f} \mathbf{1}_{L_F}^\top, \sum_{c=1}^{C} \mathbf{Z}_{:,F+c} \mathbf{1}_{L_F}^\top] = \sigma_{N-1} \cdot \widetilde{\mathbf{Z}}.$$

Hence, by Equation (41)

$$\forall l \in [L_F]: \quad q^{(N)}(\widetilde{Z}_{:,l,:}) = q^{(N)}(\widetilde{Y}_{:,l,:}),$$

and by the definition of $Q^{(NF)}$ in Equation (43), we obtain $Q^{(NF)}(Y) = Q^{(NF)}(Z)$.

**Direction $\Downarrow$ of Equation (44).** We now show the opposite direction. Let $Y, Z \in \mathbb{R}^{N \times (F+C)} \setminus \mathcal{E}$, and assume that $Q^{(NF)}(Z) = Q^{(NF)}(Y)$. By the definition of $Q^{(NF)}$ in Equation (43), we have that

$$\forall l \in [L_F]: \quad q^{(N)}(\widetilde{Z}_{:,l,:}) = q^{(N)}(\widetilde{Y}_{:,l,:}).$$

By Equation (41), this equality implies the existence of a permutation $\sigma_{N-1}^{(l)} \in S_{N-1}$, for each $l \in [L_F]$, such that

$$\widetilde{Y}_{:,l,:} = \sigma_{N-1}^{(l)} \cdot \widetilde{Z}_{:,l,:}. \tag{46}$$

Next, we would like to show that

$$\forall l_1, l_2 \in [L_F]: \quad \sigma_{N-1}^{(l_1)} = \sigma_{N-1}^{(l_2)}.$$

For that, consider two indices $l_1 \neq l_2 \in [L_F]$. By Equation (46), and by the definition of $\widetilde{Z}$ and $\widetilde{Y}$ (Equation (42)), we get, for $j = 1, 2$,

$$\sum_{f=1}^{F} Y_{:,f} \mathbf{1}_{L_F}^{\top} = \sigma_{N-1}^{(l_j)} \cdot \sum_{f=1}^{F} Z_{:,f} \mathbf{1}_{L_F}^{\top} \quad \text{and} \quad \sum_{c=1}^{C} Y_{:,F+c} \mathbf{1}_{L_F}^{\top} = \sigma_{N-1}^{(l_j)} \cdot \sum_{c=1}^{C} Z_{:,F+c} \mathbf{1}_{L_F}^{\top}. \tag{47}$$

Note that $\sum_{f=1}^{F} Z_{:,f} \mathbf{1}_{L_F}^{\top}$ has equal columns (or rows consisting of a scalar times $\mathbf{1}_{L_F}^{\top}$), and so does $\sum_{c=1}^{C} Z_{:,F+c} \mathbf{1}_{L_F}^{\top}$. The same property holds also for $\sigma_{N-1}^{(l_j)} \cdot \sum_{f=1}^{F} Z_{:,f} \mathbf{1}_{L_F}^{\top}$ and $\sigma_{N-1}^{(l_j)} \cdot \sum_{c=1}^{C} Z_{:,F+c} \mathbf{1}_{L_F}^{\top}$ for $j = 1, 2$. Moreover, since $S_{F-1} \times S_C$ permutes all rows simultaneously, Equation (47) is equivalent to

$$\sigma_{N-1}^{(l_1)} \cdot \sum_{f=1}^{F} Z_{:,f} = \sigma_{N-1}^{(l_2)} \cdot \sum_{f=1}^{F} Z_{:,f} \quad \text{and} \quad \sigma_{N-1}^{(l_1)} \cdot \sum_{c=1}^{C} Z_{:,F+c} = \sigma_{N-1}^{(l_2)} \cdot \sum_{c=1}^{C} Z_{:,F+c} \tag{48}$$

Recall that $Y, Z \in \mathbb{R}^{N \times (F+C)} \setminus \mathcal{E}$. Hence $\sum_{f=1}^{F} Y_{:,f}, \sum_{f=1}^{F} Z_{:,f}, \sum_{c=1}^{C} Y_{:,F+c}$ and $\sum_{c=1}^{C} Z_{:,F+c}$, each have $N$ distinct elements. Therefore the only way that Equation (48) holds is when all column–wise permutations coincide, i.e.,

$$\exists \sigma_{N-1} \in S_{N-1}, \forall l \in [L_F]: \quad \sigma_{N-1}^{(l)} = \sigma_{N-1}.$$

and by Equation (46)

$$\forall l \in [L_F]: \quad Q^{(F)}(Y)_{:,l} = \sigma_{N-1} \cdot Q^{(F)}(Z)_{:,l}.$$

Recall that $Q^{(F)}$ is $S_N$-equivariant, so

$$Q^{(F)}(Y) = \sigma_{N-1} \cdot Q^{(F)}(Z) = Q^{(F)}(\sigma_{N-1} \cdot Z).$$

Hence, by Equation (35)

$$\exists (\sigma_{F-1}, \sigma_C) \in S_{F-1} \times S_C: \quad Y = (\sigma_{N-1}, \sigma_{F-1}, \sigma_C) \cdot Z.$$

Thus, $Q^{(NF)}: \mathbb{R}^{N \times (F+C)} \to \mathbb{R}^{L_N \times L_F}$ satisfies Equation (44).

Similarly, we define a polynomial $Q^{(NC)}: \mathbb{R}^{N \times (F+C)} \setminus \mathcal{E} \to \mathbb{R}^{L_N \times L_C}$ as follows. Given an input $X \in \mathbb{R}^{N \times (F+C)} \setminus \mathcal{E}$, the output $Q^{(NF)}(X)$ is defined via the following composition of computations.

(i) Compute $Q^{(C)}(X) \in \mathbb{R}^{N \times L_C}$, as defined in Equation (40).

(ii) Consider the column vectors $\sum_{f=1}^{F} X_{:,f} \in \mathbb{R}^N$ and $\sum_{c=1}^{C} X_{:,F+c} \in \mathbb{R}^N$. Append these vectors to $Q_C(X)$ forming a new dimension. Namely, construct the intermediate tensor

$$\bar{X} = [Q^{(C)}(X), \sum_{f=1}^{F} X_{:,f} \mathbf{1}_{L_C}^{\top}, \sum_{c=1}^{C} X_{:,F+c} \mathbf{1}_{L_C}^{\top}] \in \mathbb{R}^{N \times L_C \times 3}.$$

(iii) Apply the function $q^{(N)}$ on $\bar{\boldsymbol{X}}$ to define $Q^{(NC)}(\boldsymbol{X})$ via

$$\forall l \in [L_C]: \quad Q^{(NC)}(\boldsymbol{X})_{:,l} := q^{(N)}(\bar{\boldsymbol{X}}_{:,l,:})$$

$$= q^{(N)}([Q^{(C)}(\boldsymbol{X})_{:,l}, \sum_{f=1}^{F} \boldsymbol{X}_{:,f}, \sum_{c=1}^{C} \boldsymbol{X}_{:,F+c}]) \in \mathbb{R}^{L_N}.$$

Then $Q^{(NC)}$ satisfies the analogue of Equation (44), i.e. $\forall \boldsymbol{Z}, \boldsymbol{Y} \in \mathbb{R}^{N \times (F+C)} \setminus \mathcal{E}$ :

$$Q^{(NC)}(\boldsymbol{Z}) = Q^{(NC)}(\boldsymbol{Y})$$

$$\Updownarrow$$

$$\exists (\sigma_{N-1}, \sigma_F, \sigma_{C-1}) \in S_{N-1} \times S_F \times S_{C-1}: \quad \boldsymbol{Y} = (\sigma_{N-1}, \sigma_F, \sigma_{C-1}) \cdot \boldsymbol{Z}.$$

**(3) Expressing $q^{(F)}$, $q^{(C)}$ and $q^{(N)}$ using DMPs and PMPs.**

Let $T_F = T_{M_F,2} = \binom{M_F+2}{2}$. Since $q^{(F)} : \mathbb{R}^{(F+C) \times 2} \to \mathbb{R}^{L_F}$ is an $(S_{F-1} \times S_C)$-polynomial descriptor of degree $M_F$, by Lemma D.19, there exist polynomials $U_{\boldsymbol{\beta}}^{(F)} : \mathbb{R}^{2T_F} \to \mathbb{R}^{L_F}$, parameterized by $\boldsymbol{\beta} \in \mathbb{Z}_0^2$, such that for $\boldsymbol{V} \in \mathbb{R}^{(F+C) \times 2}$

$$q^{(F)}(\boldsymbol{V}) = \sum_{\|\boldsymbol{\beta}\|_1 \leq M_F} \boldsymbol{V}_{1,:}^{\boldsymbol{\beta}} W_{\boldsymbol{\beta}}^{(F)}(\boldsymbol{V}), \tag{49}$$

where

$$W_{\boldsymbol{\beta}}^{(F)} = U_{\boldsymbol{\beta}}^{(F)} \circ t_{\mathrm{DMP}}^{(F+C,M_F,2)} \circ b^{(F+C,M_F,2)}. \tag{50}$$

Similarly, there exist polynomials $U_{\boldsymbol{\beta}}^{(C)} : \mathbb{R}^{2T_F} \to \mathbb{R}^{L_F}$ such that

$$q^{(C)}(\boldsymbol{V}) = \sum_{\|\boldsymbol{\beta}\|_1 \leq M_F} \boldsymbol{V}_{1,:}^{\boldsymbol{\beta}} W_{\boldsymbol{\beta}}^{(C)}(\boldsymbol{V}),$$

where

$$W_{\boldsymbol{\beta}}^{(C)} = U_{\boldsymbol{\beta}}^{(C)} \circ t_{\mathrm{DMP}}^{(F+C,M_F,2)} \circ b^{(F+C,M_F,2)}.$$

Lastly, for $T_N = \binom{M_N+3}{3}$, there exist polynomials $U_{\boldsymbol{\beta}}^{(N)} : \mathbb{R}^{T_N} \to \mathbb{R}^{L_N}$, parameterized by $\boldsymbol{\beta} \in \mathbb{Z}_0^3$, such that for $\boldsymbol{V} \in \mathbb{R}^{N \times 3}$

$$q^{(N)}(\boldsymbol{V}) = \sum_{\|\boldsymbol{\beta}\|_1 \leq M} \boldsymbol{V}_{1,:}^{\boldsymbol{\beta}} W_{\boldsymbol{\beta}}^{(N)}(\boldsymbol{V}), \tag{51}$$

where

$$W_{\boldsymbol{\beta}}^{(N)} = U_{\boldsymbol{\beta}}^{(N)} \circ t_{\mathrm{PMP}}^{(N,M_N,3)} \circ b^{(N,M_N,3)}. \tag{52}$$

**(4) Approximating $Q^{(F)}$ using TSNet$^{(F)}$.** Next, we show how to approximate $Q^{(F)}$ by a TSNet$^{(F)}$ via a composition of approximations of the components of $Q^{(F)}$.

**(4.1) The space of polynomials $\mathcal{B}_1$ and MLPs $\mathcal{N}_1$.** Note that $c^{(M_F,2)} : \mathbb{R}^2 \to \mathbb{R}^{T_F}$ of Definition D.17 belongs to the space $\mathcal{B}_1$ of all polynomials from $\mathbb{R}^2$ to $\mathbb{R}^{T_F}$. By the Universal Approximation Theorem (Theorem D.12), the space $\mathcal{N}_1$ of multilayer perceptrons (MLPs) from $\mathbb{R}^2$ to $\mathbb{R}^{T_F}$ is a universal approximator of $\mathcal{B}_1$.

**(4.2) The space of polynomials $\mathcal{B}_b$.** The polynomial $b^{(F+C,M_F,2)}$ applies $c^{(M_F,2)}$ row-wise (see Definition D.17) and belongs to the space $\mathcal{B}_b$ of all polynomials $z : \mathbb{R}^{(F+C) \times 2} \to \mathbb{R}^{(F+C) \times T_F}$ of the form $z(\boldsymbol{V}) = (y(\boldsymbol{V}_{1,:}), \ldots, y(\boldsymbol{V}_{F+C,:}))^\top$ for an arbitrary polynomial $y \in \mathcal{B}_1$.

**The space of MLPs $\mathcal{N}_b$.** We define a corresponding space $\mathcal{N}_b$ of MLPs from $\mathbb{R}^{(F+C) \times 2}$ to $\mathbb{R}^{(F+C) \times T_F}$, in which each function $\hat{z} \in \mathcal{N}_b$ is of the form $\hat{z}(\boldsymbol{V}) := (\hat{y}(\boldsymbol{V}_{1,:}), \ldots, \hat{y}(\boldsymbol{V}_{F+C,:}))^\top$ where $\hat{y} \in \mathcal{N}_1$ as defined in (4.1). It is easy to see that $\mathcal{N}_b$ is a universal approximator of $\mathcal{B}_b$. Indeed, let $z(\boldsymbol{V}) = (y(\boldsymbol{V}_{1,:}), \ldots, y(\boldsymbol{V}_{F+C,:}))^\top$ be a polynomial in $\mathcal{B}_b$ restricted to a compact domain $C \subset \mathbb{R}^{(F+C) \times 2}$, and let $\epsilon > 0$. The domain $C$ is contained in a larger compact domain of the form

$$\hat{C} = \underbrace{C' \times \ldots \times C'}_{F+C},$$

where $C' \subset \mathbb{R}^2$ is compact. Now, note that $y$ is extended uniquely to a polynomial $y : C' \to \mathbb{R}^T$. Hence, since $\mathcal{N}_1$ is a universal approximator of $\mathcal{B}_1$, $y$ can be uniformly approximated by some $\hat{y} \in \mathcal{N}_1$ over the compact domain $C'$, up to error $\epsilon$. As a result, $z$ is uniformly approximated by $\hat{z}(\boldsymbol{V}) := (\hat{y}(\boldsymbol{V}_{1,:}), \dots, \hat{y}(\boldsymbol{V}_{F+C,:}))^\top$ up to error $\epsilon$ over $\hat{C}$. Lastly, since $C \subset \hat{C}$, $z$ is uniformly approximated by $\hat{z}$ up to $\epsilon$ also in $C$.

**(4.3) The space of linear mappings $\mathcal{N}_t$.** The transformation $t_{\mathrm{DMP}}^{(F+C, M_F, 2)} : \mathbb{R}^{(F+C) \times T_F} \to \mathbb{R}^{2T_F}$ of Definition D.17 belongs to the space $\mathcal{N}_t$ of linear mappings spanned by the two linear mappings

$$[F_{\mathrm{sum}}] : \mathbb{R}^{(F+C) \times T_F} \to \mathbb{R}^{2T_F}, \quad [F_{\mathrm{sum}}](\boldsymbol{X}, \boldsymbol{Y}) = (\boldsymbol{X}\mathbf{1}_F, 0) \tag{53}$$

and

$$[C_{\mathrm{sum}}] : \mathbb{R}^{(F+C) \times T_F} \to \mathbb{R}^{2T_F}, \quad [C_{\mathrm{sum}}](\boldsymbol{X}, \boldsymbol{Y}) = (0, \boldsymbol{Y}\mathbf{1}_C). \tag{54}$$

Trivially, $\mathcal{N}_t$ is a universal approximator of itself.

**(4.4) The space of polynomials $\mathcal{B}_u$ and MLPs $\mathcal{N}_u$.** Consider the polynomial $U^{(F)} : \mathbb{R}^{2T_F} \to \mathbb{R}^{T_F \times L_F}$ defined by

$$U^{(F)} = \{U_{\boldsymbol{\beta}}^{(F)} : \mathbb{R}^{2T_F} \to \mathbb{R}^{L_F}\}_{\|\boldsymbol{\beta}\|_1 \le M_F}. \tag{55}$$

The mapping $U^{(F)}$ belongs to the space $\mathcal{B}_u$ of all polynomials from $\mathbb{R}^{2T_F}$ to $\mathbb{R}^{T_F \times L_F}$. By the Universal Approximation Theorem (Theorem D.12) the space $\mathcal{N}_u$ of multilayer perceptrons (MLPs) from $\mathbb{R}^{2T_F}$ to $\mathbb{R}^{T_F \times L_F}$ is a universal approximator of $\mathcal{B}_u$.

**(4.5) The space of polynomials $\mathcal{B}_w$ and MLPs $\mathcal{N}_w$.** Define the polynomial $W^{(F)} : \mathbb{R}^{(F+C) \times 2} \to \mathbb{R}^{T_F \times L_F}$ by

$$W^{(F)} = \{W_{\boldsymbol{\beta}}^{(F)}\}_{\|\boldsymbol{\beta}\|_1 \le M_F},$$

and note that by (50) we have

$$W^{(F)} = U^{(F)} \circ t_{\mathrm{DMP}}^{(F+C, M_F, 2)} \circ b^{(F+C, M_F, 2)}. \tag{56}$$

Define the space of MLPs $\mathbb{R}^{(F+C) \times 2} \to \mathbb{R}^{T_F \times L_F}$

$$\mathcal{N}_w := \{\theta_u \circ \theta_t \circ \theta_b \mid \theta_u \in \mathcal{N}_u, \theta_t \in \mathcal{N}_t, \theta_b \in \mathcal{N}_b\}.$$

By Theorem D.14 and the above derivation, the space of MLPs $\mathcal{N}_w$ is a universal approximator of the space of polynomial $\mathbb{R}^{(F+C) \times 2} \to \mathbb{R}^{T_F \times L_F}$

$$\mathcal{B}_w := \{f_u \circ f_t \circ f_b \mid f_u \in \mathcal{B}_u, f_t \in \mathcal{N}_t, f_b \in \mathcal{B}_b\}.$$

Moreover, by (56) we have $W^{(F)} \in \mathcal{B}_w$.

**(4.6) The space of polynomials $\mathcal{B}_F$.** Next, we construct a space $\mathcal{B}_F$ of polynomials from $\mathbb{R}^{N \times (F+C)} \setminus \mathcal{E}$ to $\mathbb{R}^{N \times L_F}$, and show that $Q^{(F)} \in \mathcal{B}_F$. A general function from the space $\mathcal{B}_F$ is defined constructively as follows. Given an input $\boldsymbol{X} \in \mathbb{R}^{N \times (F+C)} \setminus \mathcal{E}$, we first compute its row-wise sum $\sum_{i=1}^N \boldsymbol{X}_{i,:}$ and append it to a new dimension of $\boldsymbol{X}$, creating a tensor $\boldsymbol{V} \in \mathbb{R}^{N \times (F+C) \times 2}$ via

$$\forall n \in [N] : \quad \boldsymbol{V}_{n,:,:} = [\boldsymbol{X}_{n,:}, \sum_{i=1}^N \boldsymbol{X}_{i,:}].$$

Note that this step is a $\mathrm{TSNet}_{\mathrm{Eq}}$. Next, for some $Y \in \mathbb{N}$ (which can be any value from $\mathbb{N}$), we define for each multi-index $\boldsymbol{\beta}$ with $\|\boldsymbol{\beta}\|_1 < Y$, a polynomial $R_{\boldsymbol{\beta}} \in \mathcal{B}_w$ ($R_{\boldsymbol{\beta}}$ can be any function from $\mathcal{B}_w$). We then apply $R_{\boldsymbol{\beta}}$ row-wise on each $\boldsymbol{V}_{n,:,:}$, and multiply it by $\boldsymbol{V}_{n,j,:}^{\boldsymbol{\beta}}$, to create the following array in $\mathbb{R}^{N \times L_F}$

$$\left\{\sum_{\|\boldsymbol{\beta}\|_1 \le Y} \boldsymbol{V}_{n,1,:}^{\boldsymbol{\beta}} R_{\boldsymbol{\beta}}(\boldsymbol{V}_{n,:,:})\right\}_{n \in [N]}.$$

Consequently, by Equations (32) and (33) and 49, $Q^{(F)} \in \mathcal{B}_F$.

**The space of MLPs $\mathcal{N}_F$.** We now define a corresponding space of MLPs from $\mathbb{R}^{N \times (F+C)} \setminus \mathcal{E}$ to $\mathbb{R}^{N \times L_F}$, that we denote by $\mathcal{N}_F$, as follows. Given an input $\boldsymbol{X} \in \mathbb{R}^{N \times (F+C)}$, first compute its row-wise sum $\sum_{i=1}^N \boldsymbol{X}_{i,:}$ and append it to a new dimension of $\boldsymbol{X}$, creating the tensor $\boldsymbol{V} \in \mathbb{R}^{N \times (F+C) \times 2}$ defined by

$$\forall n \in [N] : \quad \boldsymbol{V}_{n,:,:} = [\boldsymbol{X}_{n,:}, \sum_{i=1}^N \boldsymbol{X}_{i,:}].$$

This step is a $\text{TSNet}_{\text{Eq}}$. Next, we apply on $\boldsymbol{V}_{n,j,:}$, for each $n \in [N]$ and $j \in [F+C]$, any sequence of general MLPs $\theta_l : \mathbb{R}^2 \to \mathbb{R}$, $l \in [L]$ for some $L \in \mathbb{N}$. Again, this computation is still a $\text{TSNet}_{\text{Eq}}$. Then, apply the same general MLP $\zeta \in \mathcal{N}_w$ on $\boldsymbol{V}_{n,:,:}$ for all $n \in [N]$. Lastly, we apply any general MLP $\kappa : \mathbb{R}^2 \to \mathbb{R}$ on each $(\theta_l(\boldsymbol{V}_{n,1,:}), \zeta(\boldsymbol{V}_{n,:,:}))$ and sum over $l$ to obtain

$$\left\{ \sum_{l=1}^{L} \kappa(\theta_l(\boldsymbol{V}_{n,1,:}), \zeta(\boldsymbol{V}_{n,:,:})) \right\}_{n \in [N]}.$$

Since we can approximate the function that takes the product of two input numbers (for us $\boldsymbol{V}_{n,1,:}^{\boldsymbol{\beta}}$ and $R_{\boldsymbol{\beta}}(\boldsymbol{V}_{n,:,:})$) by an MLP $\kappa : \mathbb{R}^2 \to \mathbb{R}$, since we can approximate monomials $\boldsymbol{V}_{n,1,:}^{\boldsymbol{\beta}}$ by MLPs $\mathbb{R}^2 \to \mathbb{R}$, and since we can approximate polynomials from $\mathcal{B}_w$ by MLPs from $\mathcal{N}_w$, the above space $\mathcal{N}_F$ is a universal approximator of $\mathcal{B}_F$. Moreover, by construction, the space $\mathcal{N}_F$ consists only of $\text{TSNets}^{(F)}$.

**(5) Approximating $Q^{(NF)}$ using TSNet\*.** Next, we show how to approximate $Q^{(NF)}$ by a $\text{TSNet}^{(NF)}$ via a composition of approximations of the components of $Q^{(NF)}$.

**(5.1) The space of polynomials $\mathcal{B}_b^{(N)}$.** Note that $c^{(M_N,3)} : \mathbb{R}^3 \to \mathbb{R}^{T_N}$ of Definition D.17 belongs to the space $\mathcal{B}_1^{(N)}$ of all polynomials from $\mathbb{R}^3$ to $\mathbb{R}^{T_N}$. By the Universal Approximation Theorem (Theorem D.12), the space $\mathcal{N}_1^{(N)}$ of multilayer perceptrons (MLPs) from $\mathbb{R}^3$ to $\mathbb{R}^{T_N}$ is a universal approximator of $\mathcal{B}_1^{(N)}$.

**(5.2) The space of polynomials $\mathcal{B}_b^{(N)}$.** The polynomial $b^{(N,M_N,3)}$ applies $c^{(M_N,3)}$ row-wise (see Definition D.17) and belongs to the space $\mathcal{B}_b^{(N)}$ of all polynomials $z : \mathbb{R}^{N \times 3} \to \mathbb{R}^{N \times T_N}$ of the form $z(\boldsymbol{V}) = (y(\boldsymbol{V}_{1,:}), \dots, y(\boldsymbol{V}_{N,:}))^\top$ for an arbitrary polynomial $y \in \mathcal{B}_1^{(N)}$.

    **The space of MLPs $\mathcal{N}_b^{(N)}$.** We define a corresponding space $\mathcal{N}_b^{(N)}$ of MLPs from $\mathbb{R}^{N \times 3}$ to $\mathbb{R}^{N \times T_N}$, in which each function $\hat{z} \in \mathcal{N}_b^{(N)}$ is of the form $\hat{z}(\boldsymbol{V}) := (\hat{y}(\boldsymbol{V}_{1,:}), \dots, \hat{y}(\boldsymbol{V}_{N,:}))^\top$ where $\hat{y} \in \mathcal{N}_1^{(N)}$ as defined in (5.1). It is easy to see that $\mathcal{N}_b^{(N)}$ is a universal approximator of $\mathcal{B}_b^{(N)}$. Indeed, let $z(\boldsymbol{V}) = (y(\boldsymbol{V}_{1,:}), \dots, y(\boldsymbol{V}_{n,:}))^\top$ be a polynomial in $\mathcal{B}_b^{(N)}$ restricted to a compact domain $C \subset \mathbb{R}^{N \times 3}$, and let $\epsilon > 0$. The domain $C$ is contained in a larger compact domain of the form

$$\hat{C} = \underbrace{C' \times \dots \times C'}_{\text{N}},$$

where $C' \subset \mathbb{R}^3$ is compact. Now, note that $y$ is extended uniquely to a polynomial $y : C' \to \mathbb{R}^T$. Hence, since $\mathcal{N}_1^{(N)}$ is a universal approximator of $\mathcal{B}_1^{(N)}$, $y$ can be uniformly approximated by some $\hat{y} \in \mathcal{N}_1^{(N)}$ over the compact domain $C'$, up to error $\epsilon$. As a result, $z$ is uniformly approximated by $\hat{z}(\boldsymbol{V}) := (\hat{y}(\boldsymbol{V}_{1,:}), \dots, \hat{y}(\boldsymbol{V}_{N,:}))^\top$ up to error $\epsilon$ over $\hat{C}$. Lastly, since $C \subset \hat{C}$, $z$ is uniformly approximated by $\hat{z}$ up to $\epsilon$ also in $C$.

**(5.3) The space of linear mappings $\mathcal{N}_t^{(N)}$.** The transformation $t_{\text{PMP}}^{(N,M_N,3)} : \mathbb{R}^{N \times T_N} \to \mathbb{R}^{T_N}$ of Definition D.17 belongs to the space $\mathcal{N}_t^{(N)}$ of linear mappings spanned by the linear mapping

$$[N_{\text{sum}}] : \mathbb{R}^{N \times T_N} \to \mathbb{R}^{T_N}, \quad [N_{\text{sum}}](\boldsymbol{X}) = \boldsymbol{X} \mathbf{1}_N. \tag{57}$$

Trivially, $\mathcal{N}_t^{(N)}$ is a universal approximator of itself.

**(5.4) The space of polynomials $\mathcal{B}_u^{(N)}$ and MLPs $\mathcal{N}_u^{(N)}$.** Consider the polynomial $U^{(N)} : \mathbb{R}^{T_N} \to \mathbb{R}^{T_N \times L_N}$ defined by

$$U^{(N)} = \{U_{\boldsymbol{\beta}}^{(N)} : \mathbb{R}^{T_N} \to \mathbb{R}^{L_N}\}_{\|\boldsymbol{\beta}\|_1 \leq M_N}. \tag{58}$$

The mapping $U^{(N)}$ belongs to the space $\mathcal{B}_u^{(N)}$ of all polynomials from $\mathbb{R}^{T_N}$ to $\mathbb{R}^{T_N \times L_N}$. By the Universal Approximation Theorem (Theorem D.12) the space $\mathcal{N}_u^{(N)}$ of multilayer perceptrons (MLPs) from $\mathbb{R}^{T_N}$ to $\mathbb{R}^{T_N \times L_N}$ is a universal approximator of $\mathcal{B}_u^{(N)}$.

**(5.5) The space of polynomials $\mathcal{B}_w^{(N)}$ and MLPs $\mathcal{N}_w^{(N)}$.** Define the polynomial $W^{(N)} : \mathbb{R}^{N \times 3} \to \mathbb{R}^{T_N \times L_N}$ by

$$W^{(N)} = \{W_{\boldsymbol{\beta}}^{(N)}\}_{\|\boldsymbol{\beta}\|_1 \leq M_F},$$

and note that by (52) we have

$$W^{(N)} = U^{(N)} \circ t_{\text{PMP}}^{(N,M_N,3)} \circ b^{(N,M_N,3)}. \tag{59}$$

Define the space of MLPs $\mathbb{R}^{N \times 3} \to \mathbb{R}^{T_N \times L_N}$

$$\mathcal{N}_w^{(N)} := \big\{ \theta_u \circ \theta_t \circ \theta_b \mid \theta_u \in \mathcal{N}_u^{(N)}, \theta_t \in \mathcal{N}_t^{(N)}, \theta_b \in \mathcal{N}_b^{(N)} \big\}.$$

By Theorem D.14 and the above derivation, the space of MLPs $\mathcal{N}_w^{(N)}$ is a universal approximator for the space of polynomial $\mathbb{R}^{N \times 3} \to \mathbb{R}^{T_N \times L_N}$

$$\mathcal{B}_w^{(N)} := \big\{ f_u \circ f_t \circ f_b \mid f_u \in \mathcal{B}_u^{(N)}, f_t \in \mathcal{B}_t^{(N)}, f_b \in \mathcal{B}_b^{(N)} \big\}.$$

Moreover, by (59) we have $W^{(N)} \in \mathcal{B}_w^{(N)}$.

**(5.6) The space of polynomials $\mathcal{B}_{NF}$.** Next, we construct a space $\mathcal{B}_F$ of polynomials from $\mathbb{R}^{N \times (F+C)} \setminus \mathcal{E}$ to $\mathbb{R}^{L_N \times L_F}$, and show that $Q^{(NF)} \in \mathcal{B}_{NF}$. A general function from the space $\mathcal{B}_{NF}$ is defined constructively as follows. Given an input $\boldsymbol{X} \in \mathbb{R}^{N \times (F+C)} \setminus \mathcal{E}$, we first apply a polynomial $E \in \mathcal{B}_F$ on $\boldsymbol{X}$ and compute the sums $\sum_{f=1}^{F} \boldsymbol{X}_{:,f}$ and $\sum_{c=1}^{C} \boldsymbol{X}_{:,F+c}$. The sums $\sum_{f=1}^{F} \boldsymbol{X}_{:,f}$ and $\sum_{c=1}^{C} \boldsymbol{X}_{:,F+c}$ are then appended to a new dimension of $E(\boldsymbol{X})$, creating a tensor $\boldsymbol{Z} \in \mathbb{R}^{N \times L_F \times 3}$ via

$$\forall l \in [L_F]: \quad \boldsymbol{Z}_{:,l,:} = [E(\boldsymbol{X})_{:,l}, \sum_{f=1}^{F} \boldsymbol{X}_{:,f}, \sum_{c=1}^{C} \boldsymbol{X}_{:,F+c}].$$

Note that this step is a $\text{TSNet}_{\text{Equi}}$. Next, for some $Y \in \mathbb{N}$ (which can be any value from $\mathbb{N}$), we define for each multi-index $\boldsymbol{\beta}$ with $\|\boldsymbol{\beta}\|_1 < Y$, a polynomial $G_{\boldsymbol{\beta}} \in \mathcal{B}_w^{(N)}$ ($G_{\boldsymbol{\beta}}$ can be any function from $\mathcal{B}_w^{(N)}$). We then apply $G_{\boldsymbol{\beta}}$ column-wise on each $\boldsymbol{Z}_{:,l,:}$, and multiply it by $\boldsymbol{Z}_{n,l,:}^{\boldsymbol{\beta}}$, to create the following array in $\mathbb{R}^{L_N \times L_F}$

$$\Big\{ \sum_{\|\boldsymbol{\beta}\|_1 \leq Y} \boldsymbol{Z}_{1,l,:}^{\boldsymbol{\beta}} G_{\boldsymbol{\beta}}(\boldsymbol{Z}_{:,l,:}) \Big\}_{l \in [L_F]}.$$

Consequently, by Equations (42) and (43) and 51, $Q^{(NF)} \in \mathcal{B}_{NF}$.

**The space of MLPs $\mathcal{N}_{NF}$.** We now define a corresponding space of MLPs from $\mathbb{R}^{N \times (F+C)} \setminus \mathcal{E}$ to $\mathbb{R}^{L_N \times L_F}$, that we denote by $\mathcal{N}_{NF}$, as follows. Given an input $\boldsymbol{X} \in \mathbb{R}^{N \times (F+C)} \setminus \mathcal{E}$, we first apply a polynomials $E \in \mathcal{B}_F$ on $\boldsymbol{X}$ and compute the sums $\sum_{f=1}^{F} \boldsymbol{X}_{:,f}$ and $\sum_{c=1}^{C} \boldsymbol{X}_{:,F+c}$. The sums $\sum_{f=1}^{F} \boldsymbol{X}_{:,f}$ and $\sum_{c=1}^{C} \boldsymbol{X}_{:,F+c}$ are then appended to a new dimension of $E(\boldsymbol{X})$, creating a tensor $\boldsymbol{Z} \in \mathbb{R}^{N \times L_F \times 3}$ via

$$\forall l \in [L_F]: \quad \boldsymbol{Z}_{:,l,:} = [E(\boldsymbol{X})_{:,l}, \sum_{f=1}^{F} \boldsymbol{X}_{:,f}, \sum_{c=1}^{C} \boldsymbol{X}_{:,F+c}].$$

This step is a $\text{TSNet}_{\text{Equi}}$. Next, we apply on $\boldsymbol{Z}_{n,l,:}$, for each $n \in [N]$ and $l \in [L_F]$, any sequence of general MLPs $\hat{\theta}_i : \mathbb{R}^3 \to \mathbb{R}$, $i \in [I]$ for some $I \in \mathbb{N}$. Again, this computation is still a $\text{TSNet}_{\text{Equi}}$. Then, apply the same general MLP $\hat{\zeta} \in \mathcal{N}_w^{(N)}$ on $\boldsymbol{Z}_{:,l,:}$ for all $l \in [L_F]$. Lastly, we apply any general MLP $\hat{\kappa} : \mathbb{R}^2 \to \mathbb{R}$ on each $(\hat{\theta}_i(\boldsymbol{Z}_{n,l,:}), \hat{\zeta}(\boldsymbol{Z}_{:,l,:}))$ and sum over $i$ to obtain

$$\Big\{ \sum_{i=1}^{I} \hat{\kappa}(\hat{\theta}_i(\boldsymbol{V}_{1,l,:}), \hat{\zeta}(\boldsymbol{Z}_{:,l,:})) \Big\}_{l \in [L_F]}.$$

Since we can approximate the function that takes the product of two input numbers (for us $\boldsymbol{Z}_{1,l,:}^{\boldsymbol{\beta}}$ and $E_{\boldsymbol{\beta}}(\boldsymbol{Z}_{:,l,:})$) by an MLP $\hat{\kappa} : \mathbb{R}^2 \to \mathbb{R}$, since we can approximate monomials $\boldsymbol{V}_{1,l,:}^{\boldsymbol{\beta}}$ by MLPs $\mathbb{R}^3 \to \mathbb{R}$, and since we can approximate polynomials from $\mathcal{B}_w^{(N)}$ by MLPs from $\mathcal{N}_w^{(N)}$, the above space $\mathcal{N}_{NF}$ is a universal approximator of $\mathcal{B}_{NF}$. Moreover, by construction, the space $\mathcal{N}_{NF}$ consists only of $\text{TSNets}^{(NF)}$.

$\qquad \square$

### D.11 Topological Spaces

In mathematics, one often formulates the most general setting under which a certain property can be analyzed. The most general setting for defining and analyzing continuous functions is *(point-set) topology*. In topology, all notions related to continuity are defined by axiomatically formulating the notion of open sets. This generalizes the concept of continuity as defined for functions between metric spaces. A topological space is a set $\mathcal{B}$ with a set of subsets $\mathcal{B} \subset 2^{\mathcal{B}}$ called the *open sets*, or the *topology*, which satisfy some list of axioms: the empty set is open ($\emptyset \in \mathcal{B}$), any finite or infinite union of open sets in open, and the intersection of any finite number of open sets in open. The complement of an open set is called *closed*.

A mapping $f : \mathcal{B} \to \mathcal{R}$ between two topological spaces $\mathcal{B}$ and $\mathcal{R}$ is called *continuous* if the preimage $f^{-1}(R) := \{b \in \mathcal{B} \mid f(b) \in \mathcal{R}\} \subset \mathcal{B}$ of every open set $R \subset \mathcal{R}$ is open. Equivalently, $f$ is continuous if and only if $f^{-1}(R)$ is closed whenever $R \subset \mathcal{R}$ is closed. The mapping $f$ is called *open* if $f(B) \subset \mathcal{R}$ is open whenever $B \subset \mathcal{B}$ is open. A continuous open bijection is called a *homeomorphism*. Two topological spaces are called homeomorphic if there is a homeomorphism between them. Two homeomorphic spaces are interpreted as "essentially the same" in the context of topology, as they share all topological properties, like continuity of functions and compactness.

In the *standard topology* of $\mathbb{R}^D$, the open sets are exactly those sets $B \subset \mathbb{R}^D$ such that for every $x \in B$ there is an open ball $C \subset \mathbb{R}^D$ (with respect to the standard Euclidean metric) such that $x \in C \subset B$. Under this topology, the classical notion of continuity from calculus is equivalent to the topological notion of continuity. More generally, every metric space has a canonical topology associated with it, defined as above, with the open balls defined with respect to the metric.

A subset $\mathcal{C} \subset \mathcal{B}$ of a topological space $\mathcal{B}$ is called *compact* if every open covering of $\mathcal{C}$ has a finite sub-covering. The topological space $\mathcal{B}$ is called compact if it is compact as a subset $\mathcal{B} \subset \mathcal{B}$. One can show that every closed set in a compact space is compact. In general, a compact set need not be closed. However, one can show that when the space $\mathcal{B}$ is a metric space, then every compact subset is closed. One can also show that continuous functions map compact sets to compact sets.

### D.12 Quotient Spaces

In the proof of Theorem 4.3 we consider the quotient space of $\mathbb{R}^{N \times (F+C)} \setminus \mathcal{E}$ with respect to orbits of the triple-symmetric group. Roughly speaking, a quotient space of a topological space $\mathcal{R}$ is another topological space $\mathcal{R}/\sim$ obtained from $\mathcal{R}$ by declaring that certain sets of points $R \subset \mathcal{R}$ should be treated as points and not sets. We will develop quotient spaces only in our setting of interest, by declaring that orbits in $\mathbb{R}^D$ (for us $D = N(F + C)$) under the action of a subgroup of the symmetric group $S_D$ (for us a triple-symmetric group) should be treated as points (see Definition A.4 for the notion of orbit).

Let $D \in \mathbb{N}$. Consider a subgroup $G$ of the symmetric group $S_D$, acting on $\mathbb{R}^D$ in the standard way. Let $\mathcal{R} \subset \mathbb{R}^D$ be an invariant set, i.e., $g \cdot \mathcal{R} = \mathcal{R}$ for every $g \in G$. Consider the subspace topology on $\mathcal{R}$, namely, a set is open in $\mathcal{R}$ if and only if it is of the form $\mathcal{R} \cap B$ where $B$ is open in $\mathbb{R}^D$. Consider the *equivalence relation* $\mathbb{R}^D \ni x \sim y \in \mathbb{R}^D$ if and only if there exists $g \in G$ such that $x = g \cdot y$. The *equivalence class* of $y \in \mathcal{R}$ is defined to be

$$[\boldsymbol{y}] := \{g \cdot \boldsymbol{y} \mid g \in G\}.$$

Note that $[\boldsymbol{y}]$ is nothing else but the orbit of $\boldsymbol{y}$ under $G$. For a set $S \subset \mathcal{R}$, we denote $[S] := \{[\boldsymbol{y}] \mid \boldsymbol{y} \in S\}$. We defined the *quotient space* $\mathcal{R}/\sim$ as follows. As a set, $\mathcal{R}/\sim$ is the set of equivalence classes, i.e.

$$\mathcal{R}/\sim := \{[\boldsymbol{y}] \mid \boldsymbol{y} \in \mathcal{R}\}.$$

Define the *canonical projection* $\pi : \mathcal{R} \to (\mathcal{R}/\sim)$ to be $\pi(\boldsymbol{y}) = [\boldsymbol{y}]$ for $\boldsymbol{y} \in \mathcal{R}$. The quotient space is equipped with the finest topology that makes the canonical projection continuous. This topology is called the *quotient topology*. More explicitly, the open sets in $\mathcal{R}/\sim$ are all sets $\tilde{B} \subset \mathcal{R}/\sim$ such that $\pi^{-1}(\tilde{B}) := \{\boldsymbol{y} \in \mathbb{R}^D \mid \pi(\boldsymbol{y}) \in \tilde{B}\}$ is open in $\mathbb{R}^D$. We can also derive a more explicit characterization of the open sets of $\mathcal{R}/\sim$.

**Lemma D.25.** *Consider the above setting. A set $\tilde{B}$ is open in $\mathcal{R}/\sim$ if and only if there exists an open set $B$ in $\mathcal{R}$ such that $B = g \cdot B$ for every $g \in G$ and $\tilde{B} = [B]$.*

*Proof.* For the first direction, let $\tilde{B}$ be open in $\mathcal{R}/\sim$. Note that $\pi^{-1}(\tilde{B}) = [B]$ for

$$B := \bigcup_{[\boldsymbol{y}]\in\tilde{B}} [\boldsymbol{y}].$$

Moreover, since $\pi$ is continuous and $\tilde{B}$ is open in $\mathcal{R}/\sim$, we must have that $B$ is open in $\mathcal{R}$. Lastly, since $B$ is a union of orbits, we must have $B = g \cdot B$ for every $g \in G$,

For the other direction, let $\tilde{B} \subset \mathcal{R}/\sim$ satisfy $\tilde{B} = [B]$ where $B$ is open in $\mathcal{R}$ and satisfy $B = g \cdot B$ for every $g \in G$. Note that the invariance of $B$ to $G$ can be written as

$$B = \bigcup_{\boldsymbol{x}\in B} [\boldsymbol{x}] = \bigcup_{[\boldsymbol{y}]\in[B]} [\boldsymbol{y}] = \{\boldsymbol{y} \in \mathcal{R}|\, \pi(\boldsymbol{y}) \in [B]\}.$$

Since $\tilde{B} = [B]$, this is equivalent to $B = \pi^{-1}(\tilde{B})$. Since, by definition, the open sets in $\mathcal{R}/\sim$ are exactly those sets $S \subset \mathcal{R}/\sim$ such that $\pi^{-1}(\tilde{B})$ are open in $\mathcal{R}$, and $B = \pi^{-1}(\tilde{B})$ is open in $\mathcal{R}$, $\tilde{B}$ must be open in $\mathcal{R}/\sim$. $\qquad\square$

We next prove a simple lemma about continuous symmetry preserving functions.

**Lemma D.26.** *Let $D \in \mathbb{N}$, and let $G \le S_D$ act on $\mathbb{R}^D$ in the standard way. Let $\mathcal{R} \subset \mathbb{R}^D$ be compact and satisfy $g \cdot \mathcal{R} = \mathcal{R}$ for every $g \in G$.*

1. *Let $q : \mathcal{R} \to \mathbb{R}^L$ be a continuous $G$-invariant function. Then, there exists a unique continuous function $\tilde{q} : \mathcal{R}/\sim\, \to \mathbb{R}^L$ such that $q = \tilde{q} \circ \pi$.*

2. *Let $q : \mathcal{R} \to \mathbb{R}^L$ be a continuous $G$-descriptor (Definition D.15). Then, there exists a unique continuous and open injective function $\tilde{q} : \mathcal{R}/\sim\, \to \mathbb{R}^L$ such that $q = \tilde{q} \circ \pi$. Namely, $\tilde{q}$ is a homeomorphism between $\mathcal{R}/\sim$ and $q(\mathcal{R})$.*

*Proof.* By the definition of G-invariant function, we have $q(\boldsymbol{x}) = q(\boldsymbol{y})$ whenever $\boldsymbol{x}$ and $\boldsymbol{y}$ belong to the same orbit. As a result, there exists a unique function $\tilde{q} : \mathcal{R}/\sim\, \to \mathbb{R}$ such that $q = \tilde{q} \circ \pi$. Next, we show that $\tilde{q}$ is continuous. Let $B \subset \mathbb{R}$ be an open set. By invariance of $q$ to the action of $G$, $q^{-1}(B)$ is invariant to $G$, i.e., $g \cdot q^{-1}(B) = q^{-1}(B)$ for every $g \in G$. Moreover, $q^{-1}(B)$ is open in $\mathcal{R}$ by continuity of $q$. Hence, by Lemma D.25, $[q^{-1}(B)]$ is open in $\mathcal{R}/\sim$. Note that $[q^{-1}(B)] = \tilde{q}^{-1}(B)$. This shows that $\tilde{q}^{-1}(B)$ is open whenever $B$ is open.

When $q$ is a continuous $G$-descriptor, $\tilde{q}$ is a continuous bijection. By continuity, $q$ maps compact sets to compact, and since it is a mapping between a compact metric space to a metric space, it maps closed to closed sets. Indeed, every closed set in a compact space is compact, and every compact set in a metric space is closed. Let $\tilde{C}$ be closed in $\mathcal{R}/\sim$. Then, by continuity of $\pi$, $\pi^{-1}(\tilde{C})$ is closed. Hence, $q(\pi^{-1}(\tilde{C}))$ is closed. Now, by the fact that $q$ is constant on orbits and by the definition of $\tilde{q}$, $q(\pi^{-1}(\tilde{C})) = \tilde{q}(\tilde{C})$, so $\tilde{q}$ also maps closed to closed sets. Since $\tilde{q}$ is invertible, $\tilde{q} = (\tilde{q}^{-1})^{-1}$ maps closed to closed sets, which is equivalent to the pre-images of all closed sets under $\tilde{q}^{-1}$ being closed. Namely, $(\tilde{q}^{-1})^{-1}(\tilde{C}) \subset \mathbb{R}^L$ is closed whenever $\tilde{C} \subset \mathcal{R}/\sim$ is closed. This is equivalent to continuity of $\tilde{q}^{-1}$, so $\tilde{q}$ is an open mapping, and together with injectivity and continuity of $\tilde{q}$, this makes $\tilde{q} : \mathcal{R}/\sim\, \to q(\mathcal{R})$ a homeomorphism. $\qquad\square$

### D.13  The Proof of Theorem 4.3

The following universal approximation theorem (Lemma 4.2) follows the same proof strategy as Theorem 3 of [26] with added elements from [38] to correct the issue detailed in Section 4.2. The lemma modifies Theorem 3 of [26] to accommodate our triple-symmetry. For a high-level overview of the proof of Lemma 4.2 and its connection to [38, 26], we refer the reader to the flow chart in Figure 4, which outlines the main steps, the key lemmas and their dependencies.

**Lemma 4.2.** *Let $\mathcal{K} \subset \mathbb{R}^{N\times(F+C)}$ be a compact domain such that $\mathcal{K} = \cup_{g\in S_N \times S_F \times S_C} g\mathcal{K}$ and $\mathcal{K} \cap \mathcal{E} = \emptyset$, where $\mathcal{E} \subset \mathbb{R}^{N\times(F+C)}$ is the exclusion set corresponding to $\mathbb{R}^{N\times(F+C)}$ (Definition D.20). Then, $TSNet_{Eq}$ is a universal approximator (i.e. in $L_\infty$) of continuous $\mathcal{K} \to \mathbb{R}^{N\times(F+C)}$ functions that are $(S_N \times S_F \times S_C)$-equivariant.*

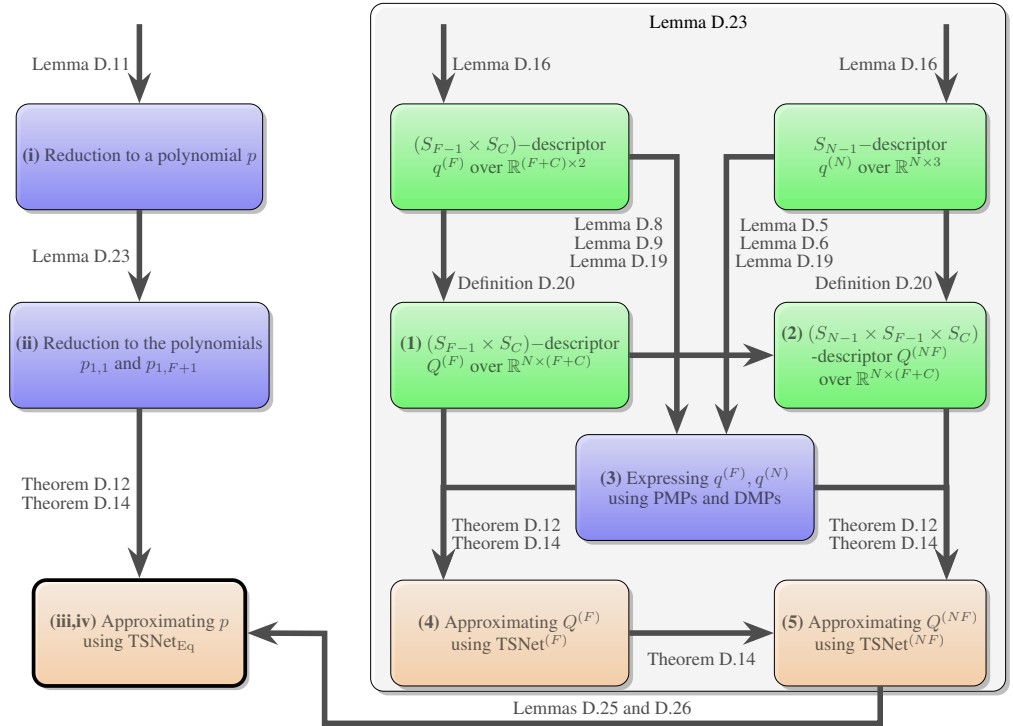

Figure 4: Flow chart illustrating the main steps in the proof of Lemmas 4.2 and D.23, which are the key lemmas used in establishing Theorem 4.3. Each box represents a logical step, and arrows indicate the lemmas, arguments, and dependencies connecting them. Blue and green boxes denote steps that extend techniques similar to, or inspired by, [38] and [26]. The proof steps of Lemma D.23 and Lemma 4.2 are labeled (1–5) and (i–iv), respectively, with the final step **(iii,iv)** highlighted with a thicker outline.

*Proof.* **(i) Reduction to polynomials.** Let $q : \mathcal{K} \to \mathbb{R}^{N \times (F+C)}$ be a continuous $(S_N \times S_F \times S_C)$-equivariant function, and let $\epsilon > 0$. By Lemma D.11, there exists an $(S_N \times S_F \times S_C)$-equivariant polynomial $p : \mathbb{R}^{N \times (F+C)} \to \mathbb{R}^{N \times (F+C)}$ of some degree $M$, such that

$$\sup_{\boldsymbol{X} \in \mathcal{K}} \|p(\boldsymbol{X}) - q(\boldsymbol{X})\|_\infty < \frac{\epsilon}{2}. \tag{60}$$

For the remainder of the proof, we show how to approximate the $(S_N \times S_F \times S_C)$-equivariant polynomial $p : \mathbb{R}^{N \times (F+C)} \to \mathbb{R}^{N \times (F+C)}$ using TSNet$_{\text{Eq}}$.

**(ii) Reduction to $p_{1,1}$ and $p_{1,F+1}$.** Recall that we denote by $p_{i,j}(\boldsymbol{X}) \in \mathbb{R}$ the $(i,j)$-th entry of $p(\boldsymbol{X})$. Note that the polynomial $p$ is $(S_N \times S_F \times S_C)$-equivariant, meaning that

$$p(\boldsymbol{X}) = (\sigma_N, \sigma_F, \sigma_C)^{-1} \cdot p((\sigma_N, \sigma_F, \sigma_C) \cdot \boldsymbol{X}), \tag{61}$$

for every $(\sigma_N, \sigma_F, \sigma_C) \in S_N \times S_F \times S_C$. By choosing $\sigma_N$ as the identity $e$, $\sigma_N = (1n)$ and $\sigma_F = (1f)$ in (61), we obtain

$$p_{n,f}(\boldsymbol{X}) = p_{1,1}((\sigma_N, \sigma_F, e) \cdot \boldsymbol{X}), \quad \forall n \in [N], f \in [F]. \tag{62}$$

Moreover, by choosing $\sigma_N = (1n)$, $\sigma_F$ as the identity $e$, and $\sigma_C = (1c)$, we obtain

$$p_{n,F+c}(\boldsymbol{X}) = p_{1,1}((\sigma_N, e, \sigma_C) \cdot \boldsymbol{X}), \quad \forall n \in [N], c \in [C]. \tag{63}$$

Thus, every entry in the output of $p$ can be recovered from $p_{1,1}$ or $p_{1,F+1}$, reducing the problem to the approximation of $p_{1,1}$ and $p_{1,F+1}$. In fact, we will approximate $p_{1,1}$ and $p_{1,F+1}$ by two MLPs $\theta^{(NF)} \in$ TSNet$^{(NF)}$ and $\theta^{(NC)} \in$ TSNet$^{(NC)}$ respectively, and Lemma D.23 will allow us to reconstruct a TSNet$_{\text{Eq}}$ from $\theta^{(NF)}$ and $\theta^{(NC)}$ which approximates $p$.

**The symmetries of $p_{1,1}$ and $p_{1,F+1}$.** The output element in the first row and column $p_{1,1}$ can be viewed as the composition of $p$ with the projection map $\pi_{1,1} : \mathbb{R}^{N\times(F+C)} \to \mathbb{R}$, defined by

$$\pi_{1,1}(\boldsymbol{X}) = \boldsymbol{X}_{1,1},$$

which extracts the $(1,1)$-st element of $\boldsymbol{X} \in \mathbb{R}^{N\times(F+C)}$. Namely, $p_{1,1} = \pi_{1,1} \circ p$. The projection map $\pi_{1,1}$ is invariant to the representation of $S_{N-1} \times S_{F-1} \times S_C$, as such permutations leave the $(1,1)$-st element unaffected. Now, since $p$ is an $(S_N \times S_F \times S_C)$-equivariant map, it is also an $(S_{N-1} \times S_{F-1} \times S_C)$-equivariant map. Thus, $p_{1,1}$ is a composition of an $(S_{N-1} \times S_{F-1} \times S_C)$-equivariant map $p$ and an $(S_{N-1} \times S_{F-1} \times S_C)$-invariant map $\pi_{1,1}$, making $p_{1,1}$ an $(S_{N-1} \times S_{F-1} \times S_C)$-invariant polynomial. By the same reasoning, $p_{1,F+1}$ is an $(S_{N-1} \times S_F \times S_{C-1})$-invariant polynomial.

**(iii) Approximating $p_{1,1}$ by a TSNet$^{(NF)}$.** For $\boldsymbol{Y}, \boldsymbol{Z} \in \mathcal{K} \subset \mathbb{R}^{N\times(F+C)}$, define the equivalence relation $\boldsymbol{Y} \sim \boldsymbol{Z}$ if there exists $(\sigma_{N-1}, \sigma_{F-1}, \sigma_C) \in S_{N-1} \times S_{F-1} \times S_C$ such that $\boldsymbol{Y} = (\sigma_{N-1}, \sigma_{F-1}, \sigma_C) \cdot \boldsymbol{Z}$. Consider the quotient space $\mathcal{K}/\sim$ and the canonical projection $\pi : \mathcal{K} \to (\mathcal{K}/\sim)$, $\pi(\boldsymbol{Y}) = [\boldsymbol{Y}]$ (see Subsection D.12). Recall that $\mathcal{K} \subset \mathbb{R}^{N\times(F+C)}$ is compact, invariant to $S_N \times S_F \times S_C$, and $\mathcal{K} \cap \left(\mathbb{R}^{N\times(F+C)} \setminus \mathcal{E}\right) = \emptyset$. Consider as well the $S_{N-1} \times S_{F-1} \times S_C$-polynomial descriptor $Q^{(NF)} : \mathcal{K} \to \mathbb{R}^L$ from the space of polynomials $\mathcal{B}^{(NF)}$ mapping $\mathcal{K}$ to $\mathbb{R}^L$, guaranteed by Lemma D.24. Here, by the lemma, the space TSNet$^{(NF)}$ of MLPs from $\mathcal{K}$ to $\mathbb{R}^L$ is a universal approximator of $\mathcal{B}^{(NF)}$. Since $Q^{(NF)}$ is a continuous $(S_{N-1} \times S_{F-1} \times S_C)$-descriptor and $p_{1,1} : \mathcal{K} \to \mathbb{R}$ is continuous and $(S_{N-1} \times S_{F-1} \times S_C)$-invariant, by Lemma D.26 there exist a homeomorphism $\widetilde{Q}^{(NF)} : (\mathcal{K}/\sim) \to Q^{(NF)}(\mathcal{K})$ and a continuous function $\widetilde{p}_{1,1} : (\mathcal{K}/\sim) \to p_{1,1}(\mathcal{K})$ such that

$$p_{1,1} = \widetilde{p}_{1,1} \circ \pi \quad \text{and} \quad Q^{(NF)} = \widetilde{Q}^{(NF)} \circ \pi.$$

Thus, we can now write $p_{1,1}$ as

$$p_{1,1} = (\widetilde{p}_{1,1} \circ (\widetilde{Q}^{(NF)})^{-1}) \circ \widetilde{Q}^{(NF)} \circ \pi = (\widetilde{p}_{1,1} \circ (\widetilde{Q}^{(NF)})^{-1}) \circ Q^{(NF)} = r \circ Q^{(NF)},$$

where $r := \widetilde{p}_{1,1} \circ (\widetilde{Q}^{(NF)})^{-1} : Q^{(NF)}(\mathcal{K}) \to \mathbb{R}$ belongs to the space $\mathcal{B}_r$ of all continuous mappings from $Q^{(NF)}(\mathcal{K})$ to $\mathbb{R}$. Since $\mathcal{K}$ is compact and $Q^{(NF)}$ is continuous, the set $Q^{(NF)}(\mathcal{K}) \subset \mathbb{R}^L$ is compact. By the Universal Approximation Theorem (Theorem D.12), the space $\mathcal{N}_r$ of multilayer perceptrons (MLPs) from $Q^{(NF)}(\mathcal{K})$ to $\mathbb{R}$ is a universal approximator of $\mathcal{B}_r$. Define

$$\mathcal{N} := \left\{ \theta_r \circ \theta_N \mid \theta_r \in \mathcal{N}_r, \theta_N \in \text{TSNet}^{(NF)} \right\}.$$

By Theorem D.14, the function space $\mathcal{N}$ is a universal approximator for the class

$$\mathcal{B} := \left\{ f_r \circ f_N \mid f_r \in \mathcal{B}_r, f_N \in \mathcal{B}^{(NF)} \right\},$$

noting that $p_{1,1} = r \circ Q^{(NF)} \in \mathcal{B}$ by the above construction.

This shows that we can approximate $p_{1,1}$ by a TSNet$^{(NF)}$ $\theta^{(NF)}$ up to error $\epsilon/2$. Similarly, we can approximate $p_{1,F+1}$ by a TSNet$^{(NC)}$ $\theta^{(NC)}$ up to error $\epsilon/2$.

**(iv) Approximating $p$ using TSNet$_{\text{Eq}}$.** Define an $(S_N \times S_F \times S_C)$-equivariant network $\theta : \mathbb{R}^{N\times(F+C)} \to \mathbb{R}^{N\times(F+C)}$ via

$$\theta_{n,f}(\boldsymbol{X}) = \theta^{(NF)}((\sigma_N, \sigma_F, e) \cdot \boldsymbol{X}), \quad \forall n \in [N], f \in [F], \tag{64}$$

and

$$\theta_{n,F+c}(\boldsymbol{X}) = \theta^{(NC)}((\sigma_N, e, \sigma_C) \cdot \boldsymbol{X}), \quad \forall n \in [N], c \in [C]. \tag{65}$$

Lemma D.23 and (62,63), $\theta$ is a TSNet$_{\text{Eq}}$ which approximates $p$ up to error $\epsilon/2$ in $L_\infty$. This, together with (60), finally gives

$$\|\theta - q\|_\infty \le \|\theta - p\|_\infty + \|p - q\|_\infty \le \epsilon.$$

$\square$

**Theorem 4.3.** *Let $K \subset \mathbb{R}^{N\times(F+C)}$ be a compact domain such that $\mathcal{K} = \cup_{g \in S_N \times S_F \times S_C} gK$ and $\mathcal{K} \cap \mathcal{E} = \emptyset$, where $\mathcal{E} \subset \mathbb{R}^{N\times(F+C)}$ is the exclusion set corresponding to $\mathbb{R}^{N\times(F+C)}$ (Definition D.20). Then, TSNets$_{\text{LI}}$ is a universal approximator of continuous $\mathcal{K} \to \mathbb{R}^{N\times C}$ functions that are $(S_N \times S_C)$-equivariant and $S_F$-invariant.*

*Proof.* Let $q : \mathcal{K} \to \mathbb{R}^{N \times C}$ be a continuous $S_N \times S_C$-equivariant and $S_F$-invariant continuous function and let $\epsilon > 0$. We define the lifted function $q^* : \mathcal{K} \to \mathbb{R}^{N \times (F+C)}$ by

$$q^*(\boldsymbol{X}) = \left(\boldsymbol{0}^{N \times F}, q(X)\right) \in \mathbb{R}^{N \times (F+C)},$$

where we pad $q(\boldsymbol{X})$ with zeros in the first $F$ columns. It is easy to verify that $q^*$ is $(S_F \times S_C \times S_N)$-equivariant, as the $S_F$ part of the group action acts only on the zero-padded section, and thus does not affect $q(\boldsymbol{X})$.

By Lemma 4.2, there exists a network TSNet$_{\mathrm{Eq}}$, denoted by $f_\varepsilon^* : \mathbb{R}^{N \times (F+C)} \to \mathbb{R}^{N \times (F+C)}$, such that

$$\|q^*(\boldsymbol{X}) - f_\varepsilon^*(\boldsymbol{X})\|_\infty < \varepsilon \quad \text{for all} \quad \boldsymbol{X} \in \mathcal{K}.$$

Consider the label projection map $\rho$ defined by

$$\rho(\boldsymbol{X}, \boldsymbol{Y}) = \boldsymbol{Y}, \quad \forall (\boldsymbol{X}, \boldsymbol{Y}) \in \mathbb{R}^{N \times F} \times \mathbb{R}^{N \times C}.$$

Notice that $q := \rho \circ q^*$ and that $f_\varepsilon := \rho \circ f_\varepsilon^*$ is a TSNet$_{\mathrm{LI}}$ architecture. Thus, we get

$$\|q(\boldsymbol{Z}) - f_\varepsilon(\boldsymbol{Z})\|_\infty = \|rho\left(q^*(\boldsymbol{Z}) - f_\varepsilon^*(\boldsymbol{Z})\right)\|_\infty \le \|q^*(\boldsymbol{Z}) - f_\varepsilon^*(\boldsymbol{Z})\|_\infty < \varepsilon,$$

for all $\boldsymbol{Z} \in \mathcal{K}$. Hence, there exists a TSNet$_{\mathrm{LI}}$ that approximates $q$ on $\mathcal{K}$ up to error $\epsilon$. $\qquad \square$

# E Theorem 3 in [26]

In this section, we present the $(S_N \times H)$-equivariant DSS layers and the corresponding DSSNet architecture, along with their universality result, as introduced in [26], for an arbitrary subgroup $H \subseteq S_F$. We then explain why the issue discussed in Section 4.2 invalidates the general universality claim for arbitrary subgroups $H \subset S_F$ and how, by drawing on techniques from [38], the claim can still be upheld in the special case $H = S_F$ – a key insight that enables us to extend the revised Theorem 3 by [26] to the triple-symmetry setting.

We begin by restating Theorem 4 from [26], which characterizes the $(S_N \times H)$-equivariant linear layers over $\mathbb{R}^{N \times F}$, for any subgroup $H \subseteq S_F$.

**Theorem E.1** (Theorem 4 in [26])**.** *Let $H \subseteq S_F$. Any linear $(S_N \times H)$-equivariant layer $L :$ $\mathbb{R}^{K_1 \times N \times F} \to \mathbb{R}^{K_2 \times N \times F}$ is of the form*

$$L(\boldsymbol{X})_{:,n,:} = L_1(\boldsymbol{X}_{:,n,:}) + L_2(\sum_{i=1}^{N} \boldsymbol{X}_{:,i,:}), \quad \forall n \in [N]$$

*where $L_1, L_2 \in \mathbb{R}^{K_1 \times F} \to \mathbb{R}^{K_2 \times F}$ are linear $H$-equivariant functions.*

**Deep sets for symmetric elements networks (DSSNets).** Given $H \subseteq S_F$, [26] define a DSS network $F : \mathbb{R}^{1 \times N \times F} \to \mathbb{R}^{1 \times N \times F}$ as:

$$F = T^{(L)} \circ \sigma \circ T^{(L-1)} \circ \sigma \circ \cdots \circ T^{(1)}, \tag{66}$$

where each $T^{(\ell)} : \mathbb{R}^{K^{(\ell)} \times N \times F} \to \mathbb{R}^{K^{(\ell+1)} \times N \times F}$ is an $(S_N \times H)$-equivariant linear layer as in Equation (66), $\sigma$ is a non-linearity and $K^{(0)}, K^{(L)} = 1$.

The universality claim in [26] uses a different exclusion set than our Theorem 4.3, but serves the same purpose – to enable the composition of descriptors while preserving our triple symmetry.

**Definition E.2** (The DSSNet exclusion set)**.** *The set*

$$\mathcal{E}' = \bigcup_{l=1}^{N} \bigcup_{1 \le f_1 < f_2 \le F} \left\{ \boldsymbol{X} \in \mathbb{R}^{N \times (F+C)} \, \middle| \, \sum_{n \ne l}^{N} \boldsymbol{X}_{n,f_1} = \sum_{n \ne l}^{N} \boldsymbol{X}_{n,f_2} \right\}$$

*is called the* DSSNet exclusion set corresponding to $\mathbb{R}^{N \times F}$.

**Theorem E.3** (Theorem 3 in [26])**.** *Let $H \subseteq S_F$. Let $\mathcal{K} \subset \mathbb{R}^{N \times F}$ be a compact domain such that $\mathcal{K} = \cup_{g \in S_N \times S_F} g\mathcal{K}$ and $\mathcal{K} \cap \mathcal{E}' = \emptyset$, where $\mathcal{E}' \subset \mathbb{R}^{N \times F}$ is the exclusion set corresponding to $\mathbb{R}^{N \times F}$ (Definition E.2). Then, DSSNet are universal approximators in $L_\infty$ of continuous $\mathcal{K} \to \mathbb{R}^{N \times F}$ functions that are $(S_N \times H)$-equivariant.*

**Discussion on universality in the general subgroup case ($H \subseteq S_F$).** The issue identified in Theorem E.3, discussed earlier in Section 4.2, stems from an attempt to span the space of $(S_{N-1} \times H)$-equivariant polynomials $p_1 : \mathbb{R}^{N \times F} \to \mathbb{R}^F$, i.e.

$$p_1(h \cdot \boldsymbol{X}) = h \cdot p_1(\boldsymbol{X}) \quad \forall h \in H. \tag{67}$$

[26] expand $p_1(\boldsymbol{X})$ using the polynomial basis $\{b_l\}_{l=0}^L : \mathbb{R}^F \to \mathbb{R}^F$ in $\boldsymbol{X}_{1,:}$ as

$$p_1(\boldsymbol{X}) = \sum_{l=0}^L \alpha_l(\boldsymbol{X}_{2,:}, \ldots, \boldsymbol{X}_{N,:}) b(\boldsymbol{X}_{1,:}),$$

where $\{c_l\} : \mathbb{R}^{(N-1) \times F} \to \mathbb{R}$ are the base coefficients and by Equation (67), we have

$$\sum_{l=0}^L \alpha_l(h \cdot \boldsymbol{X}_{2,:}, \ldots, h \cdot \boldsymbol{X}_{N,:}) b(h \cdot \boldsymbol{X}_{1,:}) = h \left( \sum_{l=0}^L \alpha_l(\boldsymbol{X}_{2,:}, \ldots, \boldsymbol{X}_{N,:}) b(\boldsymbol{X}_{1,:}) \right). \tag{68}$$

The misstep arises in their assertion that 'If we fix $\boldsymbol{X}_2, ..., \boldsymbol{X}_N$, then $p_1(\boldsymbol{X}_1, \ldots, \boldsymbol{X}_N)$ is an $H$-equivariant polynomial in $\boldsymbol{X}_1$", which does not hold generally as explained in Section 4.2. The goal of this step was to express the equivariant function in terms of invariant functions to later leverage the richer approximation theory for invariant functions.

We avoid the aforementioned pitfall for the special case $H = S_F$ by exploiting the stronger symmetry structure provided by the permutation groups rather than their subgroups. Following similar reasoning to Lemma 1 of [38], we show in step (2) of the proof of Lemma 4.2 that the first component $p_{1,1} : \mathbb{R}^{N \times F} \to \mathbb{R}$ of the $(S_{N-1} \times S_F)$-equivariant polynomial $p_1 : \mathbb{R}^{N \times F} \to \mathbb{R}^F$ is $(S_{N-1} \times S_{F-1})$-invariant and fully determines the remaining components (see Equation (62)). This step corrects the flawed transition in [26]–namely, the decomposition of an equivariant polynomial into invariant components–allowing us to recover a corrected (though less general) version of their Theorem 3 for $H = S_F$, and to extend the result to our triple-symmetry setting in Theorem 4.3.

## F  Time and Memory Complexity of TS-GNNs

**Time complexity.** Given a graph with $N$ nodes, $F$ features, and $E$ edges, TS-GNNs operate in three main stages:

- **Local aggregation:** A non-learnable, standard aggregation over each node's neighborhood. This step has a time complexity of $\mathcal{O}(EF)$.

- **Least-squares (LS) preprocessing:** A one-time LS problem is solved using the aggregated features. This step has a complexity of $\mathcal{O}(NF^2 + F^3)$. Importantly, this computation is performed once per graph and does not scale with the number of layers or training epochs.

- **Message-passing (MP):** During the forward pass, standard MP operations are performed for each layer, with a time complexity of $\mathcal{O}(EF)$ per layer.

For comparison, GraphAny, the only other zero-shot baseline, uses a similar preprocessing pipeline. It performs multiple local aggregations and LS solutions over different neighborhood radii, followed by a transformer over the resulting representations. Despite these architectural differences, its overall computational cost remains on the order of $\mathcal{O}(E)$, making it comparable to TS-GNN and consistent with the standard complexity of message-passing neural networks (MPNNs).

**Memory complexity.** The intermediate node representations produced by TS-GNN have size $N \times F \times K$, where $K$ is the number of channels in the model. During message-passing, tensors of size $E \times F \times K$ must also be kept in memory. Compared to standard message-passing neural networks, this represents a $K$-fold increase in memory usage.

Although the asymptotic memory complexity remains the same in $\mathcal{O}$-notation with $K = \mathcal{O}(1)$, in practice the additional factor $K$ increases memory usage, which can reduce efficiency and potentially create bottlenecks for very large graphs or high-dimensional feature spaces.

Table 2: Zero-shot per-dataset accuracy of TS-Mean and average accuracy of GraphAny.

| | cora | chameleon | roman-empire | wiki-cs | average |
|---|---|---|---|---|---|
| actor | 28.25 ± 1.28 | 31.62 ± 1.04 | 29.91 ± 0.91 | 26.16 ± 0.67 | 28.98 ± 0.98 |
| amazon-ratings | 42.03 ± 1.44 | 44.31 ± 1.19 | 38.42 ± 1.26 | 41.99 ± 0.53 | 41.69 ± 1.10 |
| arxiv | 55.78 ± 2.02 | 54.01 ± 5.35 | 48.98 ± 5.01 | 56.46 ± 2.38 | 53.81 ± 3.69 |
| blogcatalog | 76.70 ± 3.48 | 73.89 ± 3.85 | 67.30 ± 11.46 | 73.59 ± 2.57 | 72.87 ± 5.34 |
| brazil | 38.46 ± 5.44 | 39.23 ± 9.96 | 32.31 ± 6.99 | 29.23 ± 8.85 | 34.81 ± 7.81 |
| citeseer | 68.66 ± 0.18 | 68.14 ± 0.71 | 66.62 ± 2.07 | 70.84 ± 0.23 | 68.56 ± 0.80 |
| co-cs | 90.93 ± 0.47 | 90.04 ± 0.67 | 91.12 ± 0.52 | 90.56 ± 0.30 | 90.66 ± 0.49 |
| co-physics | 92.52 ± 0.73 | 91.36 ± 1.74 | 92.92 ± 0.48 | 91.67 ± 0.87 | 92.12 ± 0.95 |
| computers | 81.15 ± 0.97 | 76.76 ± 2.08 | 76.73 ± 6.50 | 78.42 ± 2.13 | 78.27 ± 2.92 |
| cornell | 70.27 ± 4.68 | 62.16 ± 3.31 | 74.05 ± 4.10 | 48.65 ± 10.98 | 63.78 ± 5.77 |
| deezer | 52.46 ± 2.57 | 52.29 ± 2.15 | 51.95 ± 2.40 | 50.87 ± 2.18 | 51.89 ± 2.33 |
| europe | 35.37 ± 6.70 | 39.12 ± 5.41 | 31.25 ± 4.57 | 30.88 ± 3.44 | 34.16 ± 5.03 |
| full-DBLP | 66.31 ± 3.43 | 66.55 ± 1.97 | 67.06 ± 4.38 | 67.09 ± 5.09 | 66.75 ± 3.72 |
| full-cora | 53.35 ± 1.07 | 53.97 ± 1.21 | 49.05 ± 4.06 | 57.52 ± 0.60 | 53.47 ± 1.74 |
| last-fm-asia | 77.69 ± 0.93 | 78.11 ± 0.93 | 76.71 ± 1.96 | 78.57 ± 0.93 | 77.77 ± 1.19 |
| minesweeper | 80.71 ± 0.54 | 80.03 ± 0.05 | 81.00 ± 0.84 | 80.03 ± 0.05 | 80.44 ± 0.37 |
| photo | 90.09 ± 1.20 | 87.05 ± 1.61 | 88.51 ± 4.38 | 89.61 ± 1.96 | 88.82 ± 2.29 |
| pubmed | 75.18 ± 0.54 | 75.28 ± 1.13 | 75.48 ± 0.45 | 76.44 ± 1.00 | 75.60 ± 0.78 |
| questions | 97.02 ± 0.01 | 97.02 ± 0.01 | 97.15 ± 0.08 | 97.02 ± 0.00 | 97.05 ± 0.03 |
| squirrel | 41.10 ± 1.44 | 49.38 ± 2.82 | 40.37 ± 1.24 | 35.43 ± 3.07 | 41.57 ± 2.14 |
| texas | 75.68 ± 3.31 | 72.97 ± 5.06 | 78.38 ± 9.17 | 62.16 ± 5.06 | 72.30 ± 5.65 |
| tolokers | 78.16 ± 0.04 | 78.16 ± 0.00 | 76.89 ± 2.83 | 78.16 ± 0.00 | 77.84 ± 0.72 |
| usa | 42.38 ± 1.99 | 39.03 ± 8.37 | 40.32 ± 7.36 | 41.37 ± 2.11 | 40.78 ± 4.96 |
| wiki-attr | 70.40 ± 1.63 | 71.23 ± 2.27 | 70.36 ± 1.69 | 71.91 ± 1.21 | 70.97 ± 1.70 |
| wisconsin | 61.63 ± 10.11 | 60.44 ± 11.39 | 62.67 ± 9.49 | 62.82 ± 9.38 | 61.89 ± 10.09 |
| TS-Mean average (25 graphs) | 65.69 ± 2.25 | 65.29 ± 2.97 | 64.22 ± 3.77 | 63.50 ± 2.62 | 64.67 ± 2.90 |
| GraphAny average (25 graphs) | 64.23 ± 1.47 | 65.40 ± 1.63 | 64.60 ± 1.69 | 64.30 ± 1.64 | 64.63 ± 1.61 |

# G   Additional Experiments

In this section, we address the following questions introduced in Section 6:

**Q3** How sensitive are TS-GNNs to the choice of pretraining dataset in low-data regimes?

**Q4** What is the impact of least-squares mixing and the bias term on TS-GNNs' generalization performance?

**Q5** Is triple-symmetry necessary for graph foundation models?

Specifically, we answer **Q3**, **Q4**, and **Q5** in Sections G.1 to G.3, respectively. Additionally, in Section G.4, we extend the experiments from Section 6.1 to include more GNN variants and in Section G.5, we provide the full per-dataset results for Figure 3.

## G.1   Sensitivity of TS-GNNs to Pretraining Data

**Setup.** We answer **Q3** by repeating the same experimental setup in Section 6.1 – we explore how sensitive our method is when a single dataset is used in training. In addition to cora, we train TS-Mean on roman-empire, chameleon, and wiki-cs. All models are evaluated zero-shot on the remaining 25 test graphs.

**Results.** Table 2 demonstrates that both TS-Mean and GraphAny maintain strong performance across different pretraining sources in the low-data regime, indicating that TS-GNNs are largely insensitive to the choice of pretraining data. While all pretraining datasets lead to comparable zero-shot accuracy, TS-Mean slightly outperforms GraphAny on average.

These observations are consistent with the results shown in Figure 3, which illustrate that across varying training set sizes, both approaches remain competitive in the low data regime when labeled examples are scarce. Notably, the true strength of TS-GNNs becomes apparent in the foundation model scenario, that is, the high data regime.

## G.2   The Importance of the Least-Squares Mixing and the Bias Term

**Setup.** To address Q4, we isolate the contributions of (a) the least-squares (LS) feature-label mixing and (b) the inclusion of a bias term. First, we evaluate a non-parametric setup across our test set of

Table 3: Accuracy of non-parametric least-squares variants with and without bias and mean aggregation.

|  | random | LS | LS+bias | LS+bias+mean |
|---|---|---|---|---|
| Average | 20.98 | $58.39 \pm 1.24$ | $58.75 \pm 1.58$ | $59.00 \pm 2.11$ |

Table 4: Zero-shot accuracy of TS-Mean and TS-GAT with and without the least-squares mixing and the bias term.

|  | plain | +LS | +LS+bias |
|---|---|---|---|
| TS-Mean | $55.43 \pm 1.12$ | $63.21 \pm 2.08$ | $66.21 \pm 1.93$ |
| TS-GAT | $55.19 \pm 1.54$ | $62.66 \pm 2.20$ | $65.85 \pm 2.38$ |

27 datasets. We compare four configurations: LS alone, LS with a bias term, LS with a bias term followed by a local mean aggregation step (analogous to MEAN-GNN), and a random baseline. Next, we assess the impact of these components on TS-Mean and TS-GAT, trained on Cora and evaluated zero-shot on the 28 datasets. We consider three configurations: plain (no LS, no bias), LS only, and LS plus bias. This allows us to understand how each component contributes to generalization in parametric TS-GNNs.

**Results.** As shown in Table 3, the plain TS-GNN variants slightly lag behind the LS baseline. This behavior is expected, since global pooling in plain TS-GNNs creates an information bottleneck that limits feature-label mixing. Nevertheless, even without LS or a bias term, plain TS-GNNs perform substantially above random guessing, indicating that the model captures transferable graph structure on its own.

Table 4 highlights that including LS alone provides a notable performance boost, demonstrating that LS and TS-GNNs capture complementary patterns. Adding a bias term further improves generalization, with the combination of LS and bias yielding the best performance across both TS-Mean and TS-GAT. The combination of the two yields greater generalization than either alone.

### G.3 Necessity of Triple-Symmetry in GFMs

**Setup.** To investigate **Q5**, we examine the importance of feature and label symmetries in graph foundation models. We consider three architectures with varying symmetries: (i) MeanGNN, which is node-permutation equivariant; (ii) DSS with local mean aggregation, denoted as DSS-Mean, which is node-equivariant and feature-permutation invariant; and (iii) **TS-Mean**, which is fully symmetric to node, feature, and label permutations (triple-symmetric). Following the setup in Section G.1, each model is trained on a single dataset, choosing from cora, roman-empire, chameleon, or wiki-cs, and evaluated zero-shot on the remaining 25 test graphs. To handle varying input and output dimensions across datasets, we pad or truncate features and labels when necessary.

**Results.** As shown in Table 5, performance increases with the inclusion of additional symmetries. MeanGNN achieves the lowest accuracy, DSS-Mean improves by incorporating feature symmetry, and TS-Mean attains the highest zero-shot accuracy by aligning with all three symmetries that are required of a GFM. This provides concrete empirical evidence that both feature and label symmetries (the triple-symmetry criteria) is crucial for strong generalization in GFMs.

### G.4 TS-GNNs variants

We extend Table 1 to include zero-shot experiments for TS-SGC, TS-GCN, and TS-GCNII, which are TS-GNN variants employing SGC [46], GCN [20], and GCNII [5] as their respective aggregation functions. Each model is trained on the cora dataset and evaluated on the remaining 28 datasets. All reported results reflect the mean accuracy and standard deviation over five random seeds.

**Results.** Table 6 reveals a interesting trend: as the GNN architecture becomes more complex–from SGC to GCN to GCNII, the zero-shot performance degrades in the low-data regime. This supports a hypothesis that complex GNN architectures, while beneficial in fully supervised settings, may overfit to training distributions and generalize poorly in transfer scenarios (when trained on a single graph).

Table 5: Zero-shot accuracy of models with varying symmetries.

|  | cora | roman-empire | chameleon | wiki-cs | average |
|---|---|---|---|---|---|
| MeanGNN | $23.30 \pm 5.12$ | $21.78 \pm 6.76$ | $23.11 \pm 4.52$ | $22.28 \pm 6.01$ | $22.62 \pm 5.60$ |
| DSS-Mean | $28.30 \pm 4.97$ | $23.69 \pm 6.33$ | $27.06 \pm 3.70$ | $24.22 \pm 4.81$ | $25.82 \pm 4.95$ |
| TS-Mean | $65.69 \pm 2.25$ | $65.29 \pm 2.97$ | $64.22 \pm 3.77$ | $63.50 \pm 2.62$ | $64.67 \pm 2.90$ |

Table 6: Zero-shot accuracy of TS-SGC, TS-GCN and TS-GCNII, all trained on cora.

|  | TS-SGC | TS-GCN | TS-GCNII |
|---|---|---|---|
| actor | $30.24 \pm 0.56$ | $29.59 \pm 1.07$ | $26.95 \pm 1.81$ |
| amazon-ratings | $41.24 \pm 1.62$ | $40.78 \pm 1.71$ | $38.10 \pm 1.94$ |
| arxiv | $54.79 \pm 1.80$ | $53.25 \pm 3.59$ | $54.53 \pm 2.13$ |
| blogcatalog | $78.94 \pm 9.06$ | $79.15 \pm 7.04$ | $56.52 \pm 5.20$ |
| brazil | $30.00 \pm 9.58$ | $26.15 \pm 9.18$ | $44.62 \pm 17.54$ |
| chameleon | $48.86 \pm 5.53$ | $48.29 \pm 5.12$ | $52.59 \pm 4.56$ |
| citeseer | $69.74 \pm 0.87$ | $69.70 \pm 0.69$ | $65.90 \pm 0.27$ |
| co-cs | $91.75 \pm 0.41$ | $91.65 \pm 0.45$ | $89.30 \pm 0.54$ |
| co-physics | $92.76 \pm 0.70$ | $92.54 \pm 0.72$ | $90.98 \pm 1.71$ |
| computers | $80.63 \pm 1.43$ | $80.01 \pm 1.59$ | $73.99 \pm 1.81$ |
| cornell | $69.19 \pm 4.52$ | $67.57 \pm 5.06$ | $60.54 \pm 3.08$ |
| deezer | $52.61 \pm 2.91$ | $52.94 \pm 2.87$ | $50.91 \pm 2.27$ |
| europe | $32.38 \pm 5.60$ | $31.75 \pm 7.06$ | $35.25 \pm 9.34$ |
| full-DBLP | $68.01 \pm 2.49$ | $67.67 \pm 1.49$ | $65.76 \pm 2.49$ |
| full-cora | $53.01 \pm 1.64$ | $52.56 \pm 0.86$ | $54.52 \pm 0.75$ |
| last-fm-asia | $76.23 \pm 1.06$ | $75.79 \pm 1.17$ | $78.22 \pm 0.76$ |
| minesweeper | $80.47 \pm 0.47$ | $80.38 \pm 0.26$ | $80.00 \pm 0.00$ |
| photo | $90.49 \pm 0.94$ | $90.23 \pm 0.98$ | $84.62 \pm 1.35$ |
| pubmed | $77.32 \pm 0.69$ | $77.60 \pm 0.31$ | $76.98 \pm 0.68$ |
| questions | $97.08 \pm 0.07$ | $97.03 \pm 0.03$ | $97.04 \pm 0.02$ |
| roman-empire | $66.99 \pm 1.40$ | $66.13 \pm 1.57$ | $52.58 \pm 1.72$ |
| squirrel | $33.22 \pm 1.93$ | $32.45 \pm 1.79$ | $34.08 \pm 3.27$ |
| texas | $72.97 \pm 10.64$ | $75.68 \pm 6.62$ | $82.70 \pm 3.63$ |
| tolokers | $72.38 \pm 8.62$ | $78.24 \pm 0.15$ | $63.37 \pm 23.79$ |
| usa | $38.81 \pm 3.08$ | $38.59 \pm 4.18$ | $31.32 \pm 6.26$ |
| wiki-attr | $74.60 \pm 1.65$ | $74.85 \pm 1.25$ | $69.70 \pm 1.62$ |
| wiki-cs | $72.62 \pm 0.80$ | $68.00 \pm 3.32$ | $65.05 \pm 10.78$ |
| wisconsin | $64.40 \pm 2.22$ | $62.85 \pm 1.97$ | $60.18 \pm 2.59$ |
| Average (28 graphs) | $64.70 \pm 2.94$ | $64.34 \pm 2.57$ | $62.01 \pm 4.00$ |

### G.5 Per-dataset Results for TS-Mean in Figure 3

In Section 6.2, we studied how the size of the training set affects zero-shot generalization by training TS-Mean and GraphAny on progressively larger subsets of a held-out training pool. Table 7 reports the per-dataset zero-shot accuracy of TS-Mean for training subset sizes of 1, 3, 5, 7, and 9. Each larger subset includes all datasets from the smaller subsets and is sampled from a held-out pool of 9 datasets. Zero-shot evaluation is performed on the remaining 20 datasets, following the protocol described in Section 6.2.

Table 7: Zero-shot accuracy of TS-Mean across 20 datasets when trained on 1, 3, 5, 7 and 9 held-out graphs.

| # Training Graphs | 1 | 3 | 5 | 7 | 9 |
|---|---|---|---|---|---|
| amazon-ratings | $42.16 \pm 1.41$ | $41.93 \pm 2.20$ | $39.92 \pm 1.30$ | $39.79 \pm 0.79$ | $42.20 \pm 0.72$ |
| arxiv | $56.20 \pm 2.50$ | $55.47 \pm 4.24$ | $57.28 \pm 1.91$ | $56.88 \pm 1.53$ | $56.13 \pm 1.67$ |
| blogcatalog | $76.22 \pm 2.90$ | $76.39 \pm 5.38$ | $77.82 \pm 5.19$ | $77.01 \pm 2.87$ | $77.90 \pm 3.80$ |
| brazil | $39.23 \pm 5.70$ | $38.46 \pm 8.16$ | $36.15 \pm 5.83$ | $33.08 \pm 7.98$ | $40.77 \pm 14.80$ |
| chameleon | $60.83 \pm 5.53$ | $58.51 \pm 4.03$ | $53.86 \pm 5.60$ | $55.88 \pm 2.82$ | $56.97 \pm 3.68$ |
| citeseer | $68.66 \pm 0.18$ | $67.58 \pm 0.50$ | $67.80 \pm 0.56$ | $67.88 \pm 0.70$ | $68.14 \pm 0.30$ |
| co-cs | $90.89 \pm 0.44$ | $90.94 \pm 0.65$ | $91.52 \pm 0.39$ | $91.45 \pm 0.43$ | $91.36 \pm 0.36$ |
| co-physics | $92.56 \pm 0.71$ | $92.75 \pm 0.64$ | $92.83 \pm 0.34$ | $92.97 \pm 0.33$ | $92.80 \pm 0.54$ |
| cornell | $68.65 \pm 2.42$ | $67.57 \pm 1.91$ | $77.84 \pm 6.16$ | $78.38 \pm 4.27$ | $74.59 \pm 4.91$ |
| deezer | $52.31 \pm 2.53$ | $51.70 \pm 2.99$ | $51.47 \pm 2.34$ | $52.06 \pm 2.22$ | $51.88 \pm 2.85$ |
| full-DBLP | $66.49 \pm 3.62$ | $67.51 \pm 2.94$ | $66.94 \pm 4.40$ | $68.97 \pm 2.82$ | $66.64 \pm 3.55$ |
| full-cora | $53.57 \pm 0.76$ | $51.04 \pm 2.15$ | $51.45 \pm 1.52$ | $53.39 \pm 1.43$ | $53.20 \pm 1.70$ |
| last-fm-asia | $78.01 \pm 1.01$ | $77.69 \pm 0.73$ | $77.40 \pm 0.85$ | $78.72 \pm 0.87$ | $78.07 \pm 0.76$ |
| minesweeper | $80.65 \pm 0.38$ | $80.41 \pm 0.41$ | $80.86 \pm 0.94$ | $80.98 \pm 0.34$ | $80.05 \pm 0.05$ |
| pubmed | $75.06 \pm 0.52$ | $75.44 \pm 1.19$ | $76.60 \pm 0.82$ | $77.92 \pm 0.45$ | $77.82 \pm 0.56$ |
| questions | $97.02 \pm 0.01$ | $97.02 \pm 0.02$ | $96.91 \pm 0.39$ | $97.18 \pm 0.02$ | $97.03 \pm 0.02$ |
| squirrel | $41.61 \pm 1.25$ | $40.31 \pm 2.24$ | $37.29 \pm 2.41$ | $37.89 \pm 0.67$ | $37.35 \pm 1.62$ |
| wiki-attr | $69.84 \pm 1.23$ | $70.76 \pm 1.58$ | $73.03 \pm 1.63$ | $74.73 \pm 0.65$ | $73.71 \pm 1.61$ |
| wiki-cs | $73.89 \pm 2.22$ | $73.13 \pm 4.16$ | $73.37 \pm 1.40$ | $72.82 \pm 3.46$ | $74.31 \pm 0.90$ |
| wisconsin | $60.78 \pm 11.76$ | $72.55 \pm 6.93$ | $76.08 \pm 2.56$ | $80.39 \pm 2.77$ | $80.47 \pm 6.04$ |
| Average (20 graphs) | $67.23 \pm 2.35$ | $67.36 \pm 2.65$ | $67.82 \pm 2.33$ | $68.42 \pm 1.87$ | $68.57 \pm 2.52$ |

# H    Lack of Node-regression Benchmarks

Our proposed architecture is designed to handle both node classification and regression tasks, as its theoretical derivation imposes no assumptions on label semantics. However, our empirical evaluation is restricted to node classification – not by design, but due to a fundamental limitation in the current ecosystem of graph machine learning benchmarks.

In our experiments, we evaluated performance across 28 node classification datasets sourced from the major graph learning libraries such as PyG [10], DGL[44], or OGB[16]. Yet, not a single node-level regression benchmark was available. This absence is not an isolated gap but reflects a broader issue in the field: the lack of standardized, diverse, and meaningful benchmarks. As recently emphasized by Bechler-Speicher et al. [2], the progress of graph foundation models is increasingly bottlenecked not by modeling capacity or computational resources, but by the lack of representative evaluation tasks.

We strongly encourage future works to develop standardized benchmarks for node-level regression. In the absence of such benchmarks, our results in Section 6 on node classification serve as a strong indicator of the method's potential for more general node-level prediction tasks.

# I    Dataset Statistics

The statistics of the 29 node classification datasets used in Table 1 can be found in Table 8. The node classification datasets used for training in Figure 3 are detailed in Table 9.

# J    Hyperparameters

We adopt the hyperparameters from [52] for the experiments in Table 1 over the MeanGNN, GAT and GraphAny architectures. The complete hyperparameter configurations, including those used for TS-Mean and TS-GAT, is provided in Table 10.

The same hyperparameters detailed in Table 10 are also used for the experiment in Figure 3 when training on a single graph. Hyperparameters used for training on more than one graph in Figure 3 are reported in Table 11.

Table 8: Statistics of the 28 node node classification datasets.

| Dataset | #Nodes | #Edges | #Feature | #Classes | Train/Val/Test Ratios (%) |
|---|---|---|---|---|---|
| actor | 7600 | 30019 | 932 | 5 | 48.0/32.0/20.0 |
| amazon-ratings | 24492 | 186100 | 300 | 5 | 50.0/25.0/25.0 |
| Arxiv | 169343 | 1166243 | 128 | 40 | 53.7/17.6/28.7 |
| blogcatalog | 5196 | 343486 | 8189 | 6 | 2.3/48.8/48.8 |
| brazil | 131 | 1074 | 131 | 4 | 61.1/19.1/19.8 |
| chameleon | 2277 | 36101 | 2325 | 5 | 48.0/32.0/20.0 |
| citeseer | 3327 | 9104 | 3703 | 6 | 3.6/15.0/30.1 |
| co-cs | 18333 | 163788 | 6805 | 15 | 1.6/49.2/49.2 |
| co-physics | 34493 | 495924 | 8415 | 5 | 0.3/49.9/49.9 |
| computers | 13752 | 491722 | 767 | 10 | 1.5/49.3/49.3 |
| cora | 2708 | 10556 | 1433 | 7 | 5.2/18.5/36.9 |
| cornell | 183 | 554 | 1703 | 5 | 47.5/32.2/20.2 |
| deezer | 28281 | 185504 | 128 | 2 | 0.1/49.9/49.9 |
| europe | 399 | 5995 | 399 | 4 | 20.1/39.8/40.1 |
| full-DBLP | 17716 | 105734 | 1639 | 4 | 0.5/49.8/49.8 |
| full-cora | 19793 | 126842 | 8710 | 70 | 7.1/46.5/46.5 |
| last-fm-asia | 7624 | 55612 | 128 | 18 | 4.7/47.6/47.6 |
| minesweeper | 10000 | 78804 | 7 | 2 | 50.0/25.0/25.0 |
| photo | 7650 | 238162 | 745 | 8 | 2.1/49.0/49.0 |
| pubmed | 19717 | 88648 | 500 | 3 | 0.3/2.5/5.1 |
| questions | 48921 | 307080 | 301 | 2 | 50.0/25.0/25.0 |
| roman-empire | 22662 | 65854 | 300 | 18 | 50.0/25.0/25.0 |
| squirrel | 5201 | 217073 | 2089 | 5 | 48.0/32.0/20.0 |
| texas | 183 | 558 | 1703 | 5 | 47.5/31.7/20.2 |
| tolokers | 11758 | 1038000 | 10 | 2 | 50.0/25.0/25.0 |
| usa | 1190 | 13599 | 1190 | 4 | 6.7/46.6/46.6 |
| wiki | 2405 | 17981 | 4973 | 17 | 14.1/42.9/43.0 |
| wiki-cs | 11701 | 431206 | 300 | 10 | 5.0/15.1/49.9 |
| wisconsin | 251 | 900 | 1703 | 5 | 47.8/31.9/20.3 |

Table 9: Datasets used for training in Figure 3.

| Train-set size | Graphs |
|---|---|
| 1 | cora |
| 3 | cora, texas, tolokers |
| 5 | cora, texas, tolokers, photo, roman-empire |
| 7 | cora, texas, tolokers, photo, roman-empire, usa, actor |
| 9 | cora, texas, tolokers, photo, roman-empire, usa, actor, computers, europe |

Table 10: Hyperparameters used in Table 1.

| | MeanGNN | GAT | GraphAny | TS-Mean | TS-GAT |
|---|---|---|---|---|---|
| lr | $2 \cdot 10^{-4}$ $5 \cdot 10^{-4}$ | $2 \cdot 10^{-4}$ $5 \cdot 10^{-4}$ | $2 \cdot 10^{-4}$ | 0.01 | 0.01 0.03 |
| hidden dimension | 64, 128 | 64, 128 | 32, 64, 128 | 16 | 16 |
| # layers | 2 | 2 | 2 | 2 | 2 |
| # batch | - | - | 128 | - | - |
| visible train labels (%) | - | - | 0.5 | 0.25, 0.3, 0.35, 0.4 | 0.25, 0.3, 0.35, 0.4 |
| # epochs | 400 | 400 | 1000 | 2000 | 2000 |
| Entropy | - | - | 1, 2 | - | - |
| # MLP layers | 0 | 0 | 1, 2 | 0 | 0 |

Table 11: Hyperparameters used in Figure 3.

| | GraphAny | TS-Mean |
|---|---|---|
| lr | $2 \cdot 10^{-4}$ | $3 \cdot 10^{-3}$ 
 $5 \cdot 10^{-3}$ |
| hidden dimension | 32, 64, 128 | 16 |
| # layers | 2 | 2 |
| # batch | 128 | 1 |
| visible train labels (%) | 0.5 | 0.6, 0.7, 0.8 |
| # epochs | 500, 1000 | 50, 100, 300, 500 |
| Entropy | 1, 2 | - |
| # MLP layers | 1, 2 | 0 |

