# OpenReview forum: "Equivariance Everywhere All At Once: A Recipe for Graph Foundation Models"
_NeurIPS.cc/2025/Conference — NeurIPS 2025 poster_

### Official Review · Reviewer_Yes9 · 2025-07-01

**Clarity:** 3
**Significance:** 3
**Originality:** 3
**Rating:** 5
**Confidence:** 4

**Summary:**

This paper focuses on designing a graph foundation model (GFM) for node tasks. The authors approach this problem from the viewpoint of equivariance, where they argue that three main types of symmetries are necessary for proper generalization: (a) Node permutation-equivariance, (b) Label permutation-equivariance, (c) Feature permutation-invariance. For example, the labels can of course differ based on the graph and task at hand. Also the node features can vary considerably by both semantic meaning and dimension. As such, respecting these symmetries are helpful to learning a GNN that are general and not specific to the graph itself (thus improving generalization). The authors propose a new model, TSNet, that respects these symmetries. The authors detail some experiments that show promising performance of their method.

**Questions:**

1. Can you report the performance in the same manner to GraphAny or a similar setting (as noted in Weakness 1). Then for each, test them on the holdout datasets and average the results. This will form a much more comprehensive set of results.

2. Can you run some ablation studies on the framework. Specifically, as noted in Weakness 2, it would be interesting to ablate some of the symmetries and the feature-label mixing.

3. This is fairly open-ended, but I'd appreciate some insight (preferably empirical) as to why one should choose TSNet over GraphAny (assuming the performance is still essentially the same as in the paper). What scenarios does TSNet outperform GraphAny? One suggestion may be to check few-shot scenarios, where GraphAny may struggle (this is just an idea).

4. Additional runtime/complexity results would be appreciated but are not necessary.

**Ethical Concerns:**

["NO or VERY MINOR ethics concerns only"]

**Final Justification:**

Overall, I like this paper a lot and the authors did a great job during the rebuttal. To me it's a clear accept.

**Limitations:**

yes

**Quality:**

2

**Strengths And Weaknesses:**

Strengths:
---------

1. I really like the motivation behind this work. Graphs can vary wildly in their properties, node features, and the downstream labels. Because of that, it makes sense to design a method that respects fundamental symmetries that help create a moe *general* GNN. As the authors note, this is a good way of learning a "general graph vocabulary", since respecting these symmetries help learn information that are more high-level and not too specific to each graph. I therefore think the approach introduced by the authors is really promising and intuitive.

2. The paper is written and explained quite well. It makes reading the paper very easy and quite enjoyable.

3. The theoretical proof in Theorem 4.2, that shows TSNet is symmetry preserving, is appreciated. I also find it's interesting that they find a mistake in the universality proof from *Maron et al. [2020]*, and show that the proof is true under certain conditions ($H = S_D$).


Weaknesses:
----------------

1. **Insufficient Main Experiments**: I don't think the authors provide enough experiments for their *main results*. The authors test when training on Cora and performing zero-shot classification on the other 27 datasets. However, an issue I have is that they only test when pretraining on one dataset (being Cora). However, why should we choose Cora? Why not another dataset? My point being, while the results may be good when training on Cora, maybe they won't be when training on other datasets. It's the same logic as running an experiment over $k$ random seeds, maybe Cora is just a "lucky seed". An alternative, used by GraphAny, is to instead train 4 different versions of their model each on a different dataset (i.e., Cora, Wisconsin, Arxiv, and Products). I recommend the authors either use this approach or a similar type of approach for a fairer and more robust comparison.

2. **Lack of Ablation Studies**: The authors introduce many components in their framework. However, no ablation study is given. This is important, as it's unclear of the benefit of various components. This is very important for some cases as it's unclear what is really driving the performance. For example, (a) How important are the use of DSS? What if you replace them with a basic DeepSet? (b) Are each type of symmetry necessary, what if you remove one? (c) How important is the feature-label mixing?. This last one is very important to me, as the authors adopt the approach from GraphAny by solving the least squares problem. If this is a big reason for the high performance, then it could suggest that many of the claims about the importance of symmetry for this problem may not be true (to an extent). Without any of these studies, I'm unsure how to practically evaluate the proposed framework.


3. **Lack of Clear Benefit over GraphAny**: I'm not arguing that SOTA performance isn't necessary for a framework/paper to be good. Furthermore, I think the proposed method is interesting on its own merits. However, the performance compared to GraphAny is nearly identical and nonetheless perfectly within variance. This raises an obvious question which is, *why should we use TSNet*? The authors don't really answer this. The authors say, on line 111, that as opposed to their proposed framework, the design of GraphAny is "very limiting for the model design space". I understand their point, but it's really hard to argue without any tangible experimental results that show when TSNet can outperform GraphAny. I strongly encourage the authors to try to find some empirical evidence as to when/why their method is better and preferred.


4. **Lack of Discussion on Model Complexity**: This is a fairly minor weakness and has little effect on my score. But I thought I'd mention it since including it would, imo, strengthen the paper. Basically, some discussion on this or some efficiency experiments would be appreciated. It's not necessary, but TSNet does introduce a little more overhead when preserving symmetry, so it'll be useful to see how this effects runtime for practical usage.


Other:
-----

1. This is a small detail, but I'd recommend changing the title to reflect that the model is only designed for node-level tasks. As of now, I think the title slightly oversells the model to be more encompassing than it actually is. A good change would be "Equivariance Everywhere All At Once: A Recipe for Graph Foundation Models for **Node-Level Task**". This is just my opinion.

---

> ### Author Rebuttal · Authors · 2025-07-31
>
> We thank the reviewer for their thorough review, for recognizing our theoretical contributions and rigour and for describing the paper as an “enjoyable read.”
>
> > W1: Insufficient Main Experiments: ... they only test when pretraining on one dataset (being Cora)... train 4 different versions of their model each on a different dataset...
>
> > Q1: Can you report the performance in the same manner to GraphAny ...
>
> Thanks for this comment. As per your suggestion, we explore how sensitive our method is when a single dataset is used in training. In addition to Cora, we train TS-Mean on roman-empire, chameleon, and wiki-cs. All models are evaluated zero-shot on the same 27 test graphs. Due to space constraints, we report only the averages; full per-dataset results will be included in the camera-ready version.
>
> ||cora|roman-empire|chameleon|wiki-cs|average|
> |-|-|-|-|-|-|
> |TS-Mean|65.96±1.90|64.63±1.33|65.66±2.60|64.52±2.34|65.19±2.04|
> |GraphAny|64.63±1.42|65.15±1.61|65.80±1.55|65.11±1.53|65.17±1.53|
>
> These results further validate that TS-Mean and GraphAny are consistently competitive in the low-data regime.
>
> This finding aligns with our last set of additional experiments, in which we vary the training set size, reinforcing that both methods are competitive in the low-data regime. However, the true advantage of TS-GNNs emerges in the foundation model setting, where multiple training graphs are introduced, instead of one. ​​We refer the reviewer to our response to W3/Q3, where we present further experiments demonstrating that TS-GNNs consistently outperform GraphAny in this more realistic high-data regime.
>
> > W2: Lack of Ablation Studies: ... (a) How important are the use of DSS? What if you replace them with a basic DeepSet? (b) Are each type of symmetry necessary, what if you remove one?
>
> These are excellent points. Let us clarify: (a) The DSS architecture plays a fundamental role in generalizing to arbitrary features. Replacing DSS with a standard DeepSet would remove the feature equivariance, yielding a model that depends on the number of features and their order. This results in having a different model for each graph, preventing it from generalizing across different feature spaces -- a key limitation of current GNNs that our work seeks to address. (b) Removing any one of the symmetry constraints would disable generalization across arbitrary graphs. This is the core novelty of our work: we derive the linear layer that corresponds to the complete set of symmetries required for a GFM. Removing any symmetry would reduce the model to a task- or graph-specific learner rather than a foundation model.
>
> That said, in response to your request for additional ablations, we have extended Table 1 in the main paper to include three additional GNN-based baselines: TS-SGC, TS-GCN and TS-GCNII. The updated table below reports the average accuracy across all 27 test graphs. Due to space constraints, we report only the averages; full per-dataset results will be included in the camera-ready version.
>
> ||GraphAny|TS-Mean|TS-GAT|TS-SGC|TS-GCN|TS-GCNII|
> |-|-|-|-|-|-|-|
> |Average|65.83±1.66|**66.21±1.93**|65.85±2.38|64.71±3.06 |64.39±2.61|62.45±3.43|
>
> The results reveal an interesting trend: as the GNN architecture becomes more complex—from SGC to GCN to GCNII, performance degrades in the zero-shot with low training data setting. This supports a hypothesis that complex GNN architectures, while beneficial in fully supervised settings, may overfit to training distributions and generalize poorly in transfer scenarios when trained on a single graph.
>
> > (c) How important is the feature-label mixing?...
>
> > Q2: Can you run some ablation ... ablate some of the the feature-label mixing.
>
> As suggested, we conduct an ablation study to isolate the effect of two key components in our TS-GNN architecture: (a) the feature-label mixing step via least squares (LS), and (b) the inclusion of a bias term. We first evaluate a non-parametric setup across our test set of 27 datasets. We compare the performance of LS alone, LS with a bias term, and LS with both a bias term and a subsequent local mean aggregation step (similar to MEAN-GNN), exhibiting similar performance.
>
> ||random|LS|LS+bias|LS+bias+mean|
> |-|-|-|-|-|
> |Average|20.98|58.39±1.24|58.75±1.58|59.00±2.11|
>
> We then assess the impact of these components on TS-Mean and TS-GAT, trained on Cora and evaluated on the 27 datasets. We assess three configurations: without LS and bias (plain), with LS only, and with both LS and bias.
>
> ||plain|+LS|+LS+bias|
> |-|-|-|-|
> |TS-Mean|55.43±1.12|63.21±2.08|**66.21±1.93**|
> |TS-GAT|55.19±1.54|62.66±2.20|**65.85±2.38**|
>
> These results show that the plain versions of TS-Mean and TS-GAT lag behind the LS. This outcome is expected as noted in lines 265–270: "...while such operations are theoretically sufficient for our universal approximation result (Section 4.2), they often suffer from practical limitations, because all feature or label information is compressed into a single global vector representation, creating a representational bottleneck.". Despite this limitation, the plain TS-GNN variants still perform substantially better than random guessing, indicating that it captures transferable structure even without the LS step or bias term.
>
> Next, the inclusion of the LS step alone offers a performance boost, suggesting that LS and TS-GNNs capture complementary patterns. The combination of the two yields greater generalization than either alone. This analysis parallels GraphAny's, where entropy normalization and distance features improve over plain GraphAny.
>
> We would like to emphasize that the central contribution of our work is theoretical and not in its performance: we provide a principled, symmetry-based recipe for constructing GFMs from GNNs.
>
> > W3: Lack of Clear Benefit over GraphAny: ... why should we use TSNet? ... it's really hard to argue without any tangible experimental results ...
>
> > Q3: ... What scenarios does TSNet outperform GraphAny? ...
>
> This is an excellent point. Due to time constraints, we were unable to include experiments demonstrating that the benefits of TS-GNNs over GraphAny are evident in the high-data regime, where the model is trained on multiple graphs. That is because GraphAny can only mix a set of fixed patterns (the least squares solutions), whereas TS-GNNs can learn complex patterns that improve as more training graphs are introduced. Moreover, training on a single graph is not only limiting but also unrealistic for a foundation model setting.
>
> We now train TS-Mean and GraphAny on increasingly larger subsets of a held-out training pool. We reserve 9 representative graphs for training and construct training subsets of sizes 1, 3, 5, 7, and 9, where each larger subset includes all datasets from the smaller trainsets. The remaining 19 datasets (plus the newly added wisconson dataset) are used for zero-shot evaluation. For each subset size, we report mean zero-shot accuracy, averaged over five random seeds.
>
> |#Train graphs|1|3|5|7|9|
> |-|-|-|-|-|-|
> |GraphAny|66.62|**67.34**|67.29|67.28|67.22|
> |TS-Mean|67.23|67.36|67.82|68.42|**68.57**|
>
> Results show that the zero-shot accuracy of **TS-Mean improves as more training graphs are introduced**. This behavior aligns with the expected characteristics of a GFM – the ability to benefit from increased data diversity. In contrast, **GraphAny’s performance remains unchanged** as more training graphs are added. These results suggest that GraphAny is not well-suited for the foundation model setting, and may instead be better tailored to scenarios with limited training data.
>
> Interestingly, both models perform similarly when trained on 1–3 graphs, indicating that TS-Mean is already competitive in the low-data regime. However, once the training set grows, TS-Mean consistently outperforms GraphAny, exhibiting the desired scailing law behavior, highlighting its superior scalability and generalization capability.
>
> > W4: Lack of Discussion on Model Complexity: ... some discussion on this .. would be appreciated...
>
> Thank you for raising this important aspect. Given a graph, we first perform (a non-learnable, standard) local aggregation over each node’s neighborhood and then solve a least-squares (LS) problem as a one-time preprocessing step. The resulting features, along with the original inputs, are passed into the TS-GNN, which performs standard message-passing (MP) during the forward pass.
>
> For a graph with $N$ nodes, $F$ features and $E$ edges, the local aggregation and message-passing steps each have time complexity $\mathcal{O}(EF)$, while the LS solution has complexity $\mathcal{O}(NF^2+F^3)$. Importantly, the LS computation is performed once per graph and does not scale with the number of layers or training epochs.
>
> The only other zero-shot baseline, GraphAny, also uses a similar preprocessing pipeline involving multiple local aggregations and LS solutions for different neighborhood radii. It then applies a transformer over the resulting representations. Despite these differences, its overall computational cost is also on the order of $\mathcal{O}(E)$, comparable to our method and consistent with the standard complexity of message-passing neural networks.
>
> We will include in the final version a comprehensive theoretical analysis of the runtime, comparing our method with GraphAny and classical GNNs.
>
> > W5: This is a small detail, but I'd recommend changing the title to ... "Equivariance Everywhere All At Once: A Recipe for Graph Foundation Models for Node-Level Task"...
>
> We appreciate your thorough review and will therefore change the title as per your suggestion for the camera-ready version.
>
> **We will include all the suggestions you've provided and experimental results in the camera-ready version with the appropriate discussion.**
>
> We hope our responses address your concerns and would greatly appreciate your reconsideration of the evaluation and increasing your score.

---

> > ### Comment · Reviewer_Yes9 · 2025-08-02
> >
> > > we explore how sensitive our method is when a single dataset is used in training
> >
> > Appreciate the results. Seems that TS-Mean and GraphAny are essentially the same in terms of performance under this setting.
> >
> > > Ablation studies
> >
> > I apologize for any confusion. I understand that these components are the core novelty of your work. However, in my opinion, it's still important to empirically verify that your assumptions are valid. For example, a main argument is that feature equivariance is vital. To prove this, we should remove this equivariance constraint and show that w/o it performance decreases. This is because saying that a type of symmetry is not enough, we need to show empirically that it really is needed.
> >
> > I do appreciate the results when considering more GNN variants though.
> >
> > > feature-label mixing
> >
> > Thanks for these experiments. I think they do indeed show that both the LS and TS components matter.
> >
> > >  TSNet outperform GraphAny
> >
> > These are really interesting results and show a clear scenario where TS is better than GraphAny. If accepted, I would definitely recommend putting these results in the main content of the paper.
> >
> > > Model Complexity
> >
> > Thanks for including this.
> >
> > **Conclusion**: I've increased my score to a 4. The only reason it isn't higher is due to the lack of ablation study on the different symmetries. While I understand the core argument made by the authors, it would be helpful to empirically show that each type of symmetry is important.

---

> > > ### Author Response · Authors · 2025-08-02
> > >
> > > We are happy to have resolved your concerns. Thank you for the positive feedback, for increasing your score and for engaging in the discussion period.
> > >
> > > > …it's still important to empirically verify that your assumptions are valid… we should remove this equivariance constraint and show that w/o it performance decreases…
> > >
> > > Thank you for your clarification. We conduct targeted experiments to verify the need for feature- and label-equivariance.
> > >
> > > **The challenge in removing the symmetries.** As we previously noted, once one of our proposed feature or label symmetries is removed, **the resulting model can no longer be applied to different graphs due to incompatible feature dimension sizes.** This mirrors the behavior of standard GNNs—for example, a GNN trained on cora (input size 1433) cannot be directly applied to pubmed (input size 500) due to incompatible feature dimensions. Consequently, it is non-trivial to build a model that can operate on graphs with different feature sets without enforcing the symmetries we propose.
> > >
> > > Nonetheless, **we are committed to addressing your concerns and designed experiments to test the necessity of our symmetry assumptions in GFMs.** We compare three architectures with varying symmetries: (i) MeanGNN: node-permutation equivariant, similar to DeepSets. (ii) DSS with local mean-aggregation (DSS-Mean for brevity): node- and feature-permutation equivariant. (iii) TS-Mean: fully symmetric to node, feature, and label permutations. This setup allows us to isolate the contribution of each symmetry to performance.
> > >
> > > We train DSS and MeanGNN on a single dataset (cora, roman-empire, chameleon or wiki-cs) and evaluate them in a zero-shot setting on the 27 test graphs. To match the varying input and output dimensions across the different graphs, we either pad or truncate the features and labels.
> > >
> > > ||cora|roman-empire|chameleon|wiki-cs|average|
> > > |-|-|-|-|-|-|
> > > |MeanGNN|23.30±5.12|21.78±6.76|23.11±4.52|22.28±6.01|22.62±5.60|
> > > |DSS-Mean|28.30±4.97|23.69±6.33|27.06±3.70|24.22±4.81|25.82±4.95|
> > > |TS-Mean|65.96±1.90|64.63±1.33|65.66±2.60|64.52±2.34|65.19±2.04|
> > >
> > > **These results allow us to quantify the individual contributions of each symmetry.** The improvement from MeanGNN to DSS is attributed to feature symmetry, while the further gain from MeanGNN to TS-Mean stems from TS-Mean aligning with all the symmetries that are required of a GFM. This provides concrete empirical evidence that both types of symmetry are crucial to achieving strong generalization.
> > >
> > > After your clarification, we'd also like to revisit and better address this part of your original concern:
> > >
> > > >... (a) How important are the use of DSS? What if you replace them with a basic DeepSet?...
> > >
> > > Due to the challenges in removing the symmetries, we cannot apply DSS or DeepSets directly in the cross-graph setting without addressing feature and label mismatches. Thus, we experiment with DSS-Mean and MeanGNN, augmented with trainable encoder–decoder modules that are fine-tuned on each test graph. **This setup allows us to resolve dimensional mismatches.**
> > >
> > > Specifically, MeanGNN lacks both feature and label equivariance, so we pair it with both an encoder and a decoder. DSS-Mean is feature-equivariant and can handle varying input features; thus, we only attach an MLP decoder to adapt to different label spaces. Both models are trained on a single graph. At inference time, for each new test graph, we freeze the DSS/GNN backbone and only train a new encoder or decoder to adapt the graph-specific features to the hidden representation learned during training.
> > >
> > > ||cora|roman-empire|chameleon|wiki-cs|average|
> > > |-|-|-|-|-|-|
> > > |MeanGNN|55.61±2.93|56.30±3.00|53.35±3.62|54.94±3.36|55.05±3.23|
> > > |DSS-Mean|61.86±2.53|61.07±2.19|60.77±2.63|60.84±2.24|61.14±2.40|
> > > |TS-Mean|65.96±1.90|64.63±1.33|65.66±2.60|64.52±2.34|65.19±2.04|
> > >
> > > As expected, these models perform better than their counterparts from the previous experiment. The improvement is due to the ability to learn graph-specific adaptations via encoder–decoder modules, allowing for better generalization.
> > >
> > > Although **this comparison is not entirely fair since TS-Mean is evaluated in a zero-shot setting, and MeanGNN and DSS-Mean are fine-tuned on the test graphs, we still observe that TS-Mean outperforms both DSS and MeanGNN**. Moreover, the relative performance hierarchy also remains consistent, with MeanGNN < DSS-Mean < TS-Mean.
> > >
> > > Finaly, the training procedure itself in fine-tuning commutes with feature and label permutations. Hence, the resulting end-to-end model, while using MeanGNN and DSS-Mean, is symmetry preserving.
> > >
> > > > The only reason it isn't higher is due to the lack of ablation study on the different symmetries
> > >
> > > We sincerely hope that our additional experiments have addressed your remaining concerns. If there are any further questions or points of clarification, we would be more than happy to address them.
> > >
> > > **We deeply appreciate your thoughtful feedback and hope our revisions merit an increase of score.**

---

> > > > ### Author Response · Authors · 2025-08-04
> > > >
> > > > Thank you for encouraging the ablation study on the necessity of our feature and label symmetries. We believe to have addressed all of your concerns. As the discussion period is closing soon, we would highly appreciate your response as this would give us a chance to address any remaining issues. We look forward to hearing from you. Thank you.

---

> > > > ### Comment · Reviewer_Yes9 · 2025-08-05
> > > >
> > > > I really do appreciate you humoring me and providing all the additional experiments. I understand the problem with removing symmetries, as you mentioned, but do still think it's important to do so in some way. I think the first table is fantastic and will greatly enhance the paper.
> > > >
> > > > I've raised my score to a 5.

---

> > > > > ### Author Response · Authors · 2025-08-06
> > > > >
> > > > > We are glad you found the experiments helpful. Thank you again for your clear intent to improve and support our work, for engaging in the discussion period, and for raising your score.

---

### Official Review · Reviewer_HcMt · 2025-07-01

**Clarity:** 2
**Significance:** 3
**Originality:** 3
**Rating:** 5
**Confidence:** 4

**Summary:**

This paper focuses on the design of Graph Foundation Models (GFMs), aiming to address the issue that traditional Graph Neural Networks (GNNs) struggle to generalize across different graph structures, feature sets, and tasks. Its core contribution is proposing a theoretical framework based on "triple-symmetry": node permutation-equivariance (predictions change consistently with the permutation of node indices), label permutation-equivariance (predictions change consistently with the permutation of labels/targets), and feature permutation-invariance (predictions are unaffected by the ordering of features).
Theoretically, the authors characterize the space of linear transformations that satisfy these triple symmetries and prove that the resulting Triple-Symmetry Network (TSNet) is a universal approximator for set functions that respect these symmetries. In practice, by applying TSNet to the local neighborhood aggregation of graphs, they propose Triple-Symmetry Graph Neural Networks (TS-GNN), which can convert any GNN into a GFM and achieve zero-shot generalization. Experimentally, the performance of TS-GNN (based on MeanGNN and GAT) is validated on 27 real-world node classification datasets, and its zero-shot performance outperforms traditional GNNs and the existing method GraphAny.

**Questions:**

1. The current framework is restricted to node-level tasks and not extended to edge-level or graph-level tasks, which is explicitly acknowledged as a limitation, restricting its generality.
2. Although the theory supports regression tasks, experiments only validate classification, lacking empirical support for regression scenarios, weakening the framework's comprehensiveness.

**Ethical Concerns:**

["NO or VERY MINOR ethics concerns only"]

**Limitations:**

The feature-label mixing relies on least-squares projections, but its performance in scenarios with weak feature-label correlation (e.g., noisy features) is not discussed, potentially leading to generalization bottlenecks.

**Quality:**

3

**Strengths And Weaknesses:**

1. Proposes "triple symmetry" (node permutation-equivariance, label permutation-equivariance, feature permutation-invariance) as the core design criterion for graph foundation models (GFMs), providing a principled framework to address the cross-graph generalization issue of traditional GNNs and filling the gap in the theoretical foundation of graph foundation models.
2. Proves that Triple-Symmetry Networks (TSNets) are universal approximators for set functions respecting the triple symmetry, providing mathematical guarantees for the model's effectiveness.
3. Validates the model on 27 real-world datasets, showing that its zero-shot performance outperforms baselines, providing sufficient empirical support.
4. Open-sources code and detailed experimental settings (e.g., datasets, hyperparameters), ensuring the reproducibility of results.

---

> ### Author Rebuttal · Authors · 2025-07-31
>
> We sincerely thank the reviewer for recognizing our framework as principled and for acknowledging our contribution in addressing a key gap in the theoretical foundations of Graph Foundation Models (GFMs). We greatly appreciate your thoughtful and encouraging feedback.
>
> > Q1: The current framework is restricted to node-level tasks and not extended to edge-level or graph-level tasks, which is explicitly acknowledged as a limitation, restricting its generality.
>
> We fully agree that this is a limitation, and we explicitly acknowledge it in lines 346–347: “Future work could explore extending this recipe to edge-level or graph-level tasks as our recipe is currently limited to the node-level case.”
>
> Our primary aim was to provide a principled approach to constructing GFMs, beginning with the node-level setting, which is foundational yet non-trivial and has been neglected in prior work. We view this paper as an important first step toward more general-purpose graph foundation models. Extending this framework to edge- and graph-level tasks is indeed part of our research vision, and we are actively exploring ways to build similarly principled models for these settings.
>
> We hope the reviewer agrees that establishing a solid theoretical footing for node-based GFMs is a necessary milestone toward achieving broader generalization in graph learning.
>
> > Q2: Although the theory supports regression tasks, experiments only validate classification, lacking empirical support for regression scenarios, weakening the framework's comprehensiveness.
>
> Our empirical evaluation is restricted to node classification – not by design, but due to a fundamental limitation in the current ecosystem of graph machine learning benchmarks. Specifically, while our experiments span 27 datasets sourced from major libraries (PyG, DGL, SNAP, OGB), we could not identify any node-level regression benchmark. This reflects a broader issue in the field: the lack of diverse and meaningful evaluation tasks for graph foundation models. As emphasized in [1], the progress of graph learning is increasingly bottlenecked not by model expressivity or compute, but by the lack of representative benchmarks.
>
> We would be glad to conduct any further experiments that you believe could strengthen the paper and warrant a higher score.
>
> > L1: The feature-label mixing relies on least-squares projections, but its performance in scenarios with weak feature-label correlation (e.g., noisy features) is not discussed, potentially leading to generalization bottlenecks.
>
> While we agree that weak feature-label correlation can present challenges, such scenarios are generally less problematic in node classification tasks compared to domains like spatio-temporal prediction (e.g., weather or traffic forecasting), where noisy or weakly informative features are more common.
>
> It is also worth noting that when faced with noisy features, the mean aggregation step performed before the least-squares (LS) projection and the GNN’s aggregation helps mitigate noise and stabilise the signal in our framework. That said, we find your observation an exciting direction for future research. In particular, extending our framework to spatio-temporal graphs could pave the way toward a general-purpose spatio-temporal Graph Foundation Model (ST-GFM). We will include a discussion of this denoising effect in the revised manuscript for clarity.
>
> We hope our responses address your concerns. **We will include all the suggestions you've provided and experimental results in the camera-ready version with the appropriate discussion.**

---

### Official Review · Reviewer_pt6w · 2025-07-03

**Clarity:** 3
**Significance:** 3
**Originality:** 3
**Rating:** 5
**Confidence:** 4

**Summary:**

This paper investigates two important properties for graph learning: label permutation-equivariance and feature permutation-invariance, in addition to node permutation-equivariance. The authors propose a novel model, the Triple-Symmetry Network (TSNet), building on the prior work "On Learning Sets of Symmetric Elements."

**Questions:**

1. The average performance reported in Table 1 is comparable to the baseline method, GraphAny.

2. The experiments currently lack graph datasets for social media, such as Instagram and Reddit, to analyze user properties. Please include these datasets during the rebuttal phase.

3. The pseudo-code for the proposed method would greatly enhance clarity and understanding.

4. Please further discuss how this model could be applied to GFM when language serves as node or edge features.

**Ethical Concerns:**

["NO or VERY MINOR ethics concerns only"]

**Final Justification:**

maintain my original score of 5

**Limitations:**

The authors write down the limitations in the introduction and discussion, noting that it is designed for node-level tasks.

**Quality:**

3

**Strengths And Weaknesses:**

The paper is generally easy to follow, and the theoretical claims are well-supported by proofs provided in the supplementary materials. However, graph datasets for social media should be added to demonstrate the strength of TSNet.

---

> ### Author Rebuttal · Authors · 2025-07-31
>
> We sincerely thank the Reviewer for finding our paper “easy to follow” and the theoretical claims “well-supported by proofs.” We value your constructive and supportive comments. We provide below the answers to the main concerns raised:
>
> > W1: However, graph datasets for social media should be added to demonstrate the strength of TSNet.
>
> > Q2: The experiments currently lack graph datasets for social media...
>
> We understand the reviewer's interest in evaluating our method on social media datasets, as this is a common and important application area for node-level prediction tasks.
>
> Unfortunately, large-scale social media graphs such as Reddit or Instagram pose a significant computational challenge. Specifically, for a graph with $N$ nodes, $E$ edges, and $F$ features, our model produces intermediate node representations of size $N\times F\times K$, where $K$ is the number of channels—typically greater than 8, and the message-passing operation retains tensors of size $E\times F\times K$. For graphs of the scale seen in social media datasets, these tensor sizes exceed the memory capacity of non-industrial GPUs.
>
> While we are unable to provide results on these datasets in this version due to resource limitations, this highlights an important direction for future research. Some potential strategies to improve scalability include tensor sparsification, subgraph sampling, and low-rank approximations.
>
> That said, we remain committed to further validating our framework. As demonstrated throughout the rebuttal, we have added new baselines, conducted detailed ablations, and explored training on a broader range of datasets. We would be happy to carry out any additional experiments within our resource limits that the reviewer believes would strengthen the work or warrant a higher score.
>
> We list the rest of the experiments performed in this rebuttal below, for the reviewer's convenience and interest:
>
> **(1) Evaluating TS-GNNs and GraphAny in the high-data regime** by varying the number of training graphs (see Q1 below)
>
> **(2) Introducing three new TS-GNNs variants**
>
> We extended Table 1 in the main paper to include three more GNN variants: TS-SGC, TS-GCN, and TS-GCNII. Below is the average accuracy over 27 datasets:
>
> ||GraphAny|TS-Mean|TS-GAT|TS-SGC|TS-GCN|TS-GCNII|
> |-|-|-|-|-|-|-|
> |Average|65.83±1.66|**66.21±1.93**|65.85±2.38|64.71±3.06 |64.39±2.61|62.45±3.43|
>
> The results reveal an interesting trend: as the GNN architecture becomes more complex -- from SGC to GCN to GCNII, performance degrades. This supports a hypothesis that complex GNN architectures, while beneficial in fully supervised settings, may overfit to training distributions and generalize poorly in transfer scenarios when trained on a single graph.
>
> **(3) Ablation over the least-Squares mixing and bias term**
>
> We isolate the effect of two key components in our TS-GNN architecture: (a) the feature-label mixing step via least squares (LS), and (b) the inclusion of a bias term. We first evaluate a non-parametric setup across our test set of 27 datasets. We compare the performance of LS alone, LS with a bias term, and LS with both a bias term and a subsequent local mean aggregation step (similar to MEAN-GNN), exhibiting similar performance.
>
> ||random|LS|LS+bias|LS+bias+mean|
> |-|-|-|-|-|
> |Average|20.98|58.39±1.24|58.75±1.58|59.00±2.11|
>
> We then assess the impact of these components on TS-Mean and TS-GAT, trained on Cora and evaluated on the 27 datasets. We assess three configurations: without LS and bias (plain), with LS only, and with both LS and bias.
>
> ||plain|+LS|+LS+bias|
> |-|-|-|-|
> |TS-Mean|55.43±1.12|63.21±2.08|**66.21±1.93**|
> |TS-GAT|55.19±1.54|62.66±2.20|**65.85±2.38**|
>
> These results show that the plain versions of TS-Mean and TS-GAT slightly lag behind the LS. This outcome is expected as the global pooling in plain TS-GNNs creates an information bottleneck that prevents feature-label mixing, as noted in lines 265–270. Despite this limitation, the plain TS-GNN variants still perform substantially better than random guessing, indicating that it captures transferable structure even without the LS step or bias term.
>
> Next, the inclusion of the LS step alone offers a performance boost, suggesting that LS and TS-GNNs capture complementary patterns. The combination of the two yields greater generalization than either alone. This analysis parallels GraphAny's, where entropy normalization and distance features improve over plain GraphAny.
>
> **(4) Ablation over the pretraining dataset**
>
> We explore how sensitive our method is when a single dataset is used in training. In addition to Cora, we train TS-Mean on roman-empire, chameleon, and wiki-cs. All models are evaluated zero-shot on the same 27 test graphs.
>
> ||cora|roman-empire|chameleon|wiki-cs|average|
> |-|-|-|-|-|-|
> |TS-Mean|65.96±1.90|64.63±1.33|65.66±2.60|64.52±2.34|65.19±2.04|
> |GraphAny|64.63±1.42|65.15±1.61|65.80±1.55|65.11±1.53|65.17±1.53|
>
> These results further validate that TS-Mean and GraphAny are consistently competitive in the low-data regime.
>
> This finding aligns with our experiments that vary the training set size, reinforcing that both methods are competitive in the low-data regime. However, the true advantage of TS-GNNs emerges in the foundation model setting—i.e., the high-data regime.
>
> > Q1: The average performance reported in Table 1 is comparable to the baseline method, GraphAny.
>
> This is an excellent point. Due to time constraints, we were unable to include experiments demonstrating that the benefits of TS-GNNs over GraphAny are evident in the high-data regime, where the model is trained on multiple graphs. That is because GraphAny can only mix a set of fixed patterns (the least squares solutions), whereas TS-GNNs can learn complex patterns that improve as more training graphs are introduced. Moreover, training on a single graph is not only limiting but also unrealistic for a foundation model setting.
>
> We now train TS-Mean and GraphAny on increasingly larger subsets of a held-out training pool. We reserve 9 representative graphs for training and construct training subsets of sizes 1, 3, 5, 7, and 9, where each larger subset includes all datasets from the smaller trainsets. The remaining 19 datasets (plus the newly added wisconson dataset) are used for zero-shot evaluation. For each subset size, we report mean zero-shot accuracy, averaged over five random seeds.
>
> |#Train graphs|1|3|5|7|9|
> |-|-|-|-|-|-|
> |GraphAny|66.62|**67.34**|67.29|67.28|67.22|
> |TS-Mean|67.23|67.36|67.82|68.42|**68.57**|
>
> Results show that the zero-shot accuracy of **TS-Mean improves as more training graphs are introduced**. This behavior aligns with the expected characteristics of a GFM – the ability to benefit from increased data diversity. In contrast, **GraphAny’s performance remains unchanged** as more training graphs are added. These results suggest that GraphAny is not well-suited for the foundation model setting, and may instead be better tailored to scenarios with limited training data.
>
> Interestingly, both models perform similarly when trained on 1–3 graphs, indicating that TS-Mean is already competitive in the low-data regime. However, once the training set grows, TS-Mean consistently outperforms GraphAny, exhibiting the desired scailing law behavior, highlighting its superior scalability and generalization capability.
>
> > Q3: The pseudo-code for the proposed method would greatly enhance clarity and understanding.
>
> Thanks for this comment. We will include a detailed pseudo-code for TS-GNNs in the camera-ready version.
>
> > Q4: Please further discuss how this model could be applied to GFM when language serves as node or edge features.
>
> Our model is designed to accommodate a wide variety of node features, including those derived from text. In fact, several of the datasets used in our experiments already incorporate textual embeddings. For example, in the citation networks Cora, Pubmed, and Citeseer, node features are constructed by applying a TF-IDF embedding to the titles and abstracts of papers. Similarly, in the OGB dataset arxiv, the textual content is embedded using a skip-gram language model. In the photo and computers datasets, nodes represent products, and the features are bag-of-words representations of product reviews, demonstrating our model’s ability to handle diverse textual node-features.
>
> As for textual edge features, while our current formulation focuses on node features, the architecture can be extended to accommodate edge features by introducing permutation symmetry over the edge-feature space, analogous to how we treat node features. Although edge features are less common in node prediction tasks and more relevant for graph-level or edge-level tasks (e.g., in molecular property prediction), we agree that this is a valuable extension. Formalizing linear layers that are permutation-invariant over edge features is non-trivial and presents an interesting direction for future research. We appreciate the suggestion and plan to explore this as part of extending our framework toward more general-purpose GFMs.
>
> We hope our responses address your concerns. **We will include all the suggestions you've provided and experimental results in the camera-ready version with the appropriate discussion.**

---

> ### Comment · Reviewer_pt6w · 2025-08-04
>
> Thank you for the rebuttal, which addresses my concerns to some extent. The authors outline TSNet’s advantages over Graphany and explain current scalability limitations and edge-feature extensions. Given that I already provided a positive score, I will maintain it.

---

### Official Review · Reviewer_KTq8 · 2025-07-06

**Clarity:** 2
**Significance:** 2
**Originality:** 2
**Rating:** 3
**Confidence:** 2

**Summary:**

This paper proposes a theoretically-grounded recipe for designing Graph Foundation Models (GFMs) for node-level classification and regression tasks. The central idea is to identify and enforce three key symmetries: node permutation-equivariance, label permutation-equivariance, and feature permutation-invariance. The authors characterize the space of linear transformations that respect these "triple-symmetries" and prove that the resulting "Triple-Symmetry Network" (TSNet) is a universal approximator on multisets. This theoretical foundation is then used to develop Triple-Symmetry Graph Neural Networks (TS-GNNs), a modular framework that can transform any standard GNN into a GFM capable of zero-shot generalization across graphs with varying structures, feature configurations, and label semantics. Extensive experiments on 27 real-world node classification datasets demonstrate that TS-GNNs (TS-Mean and TS-GAT) consistently outperform their non-transferable baselines and also improve upon the prior state-of-the-art GraphAny in a zero-shot setting.

**Questions:**

See weaknesses.

And:
- Could the authors discuss potential future work on extending this recipe to more complex graph structures, such as heterogeneous graphs with multiple node and edge types, or dynamic graphs where the topology changes over time?

- What about the scalability of the proposed method? When talking about pre-training the graph neural network, what scale and computational cost and downstream task coverage is being discussed? Conceptually compare this with large pre-trained language models or vision models. Can the experiments go beyond the small simple node classification tasks?

**Ethical Concerns:**

["NO or VERY MINOR ethics concerns only"]

**Final Justification:**

Many of the experimental concerns were resolved, but there are some remaining concerns of generalizability and scale of the approach (empirically), which the authors acknowledge. While I keep my original rating, I would not object if the paper gets accepted. I will leave the other reviewers and AC on the final recommendation, as I do not have a good grasp of the theoretic vs. empirical importance in this work.

**Limitations:**

The paper does not include explicit limitation sections. Although in the checklist limitations were briefly mentioned, I would encourage the authors to include an explicit limitations section.

**Quality:**

2

**Strengths And Weaknesses:**

Strengths

- The paper's primary strength lies in its rigorous theoretical analysis. The identification of the "triple-symmetry" criterion and the subsequent characterization of equivariant linear layers, culminating in the universality proof for TSNets, provides a robust mathematical underpinning for the proposed GFMs. This distinguishes the work from many empirically-driven approaches.

- The TS-GNN framework offers a practical and modular way to convert existing GNN architectures into GFMs. This approach allows for leveraging the strengths of diverse GNN designs while enabling zero-shot generalization.

- The extensive experiments on 27 diverse node classification datasets provide evidence for the effectiveness of TS-GNNs in a zero-shot setting.


Weaknesses

- While the experiments are extensive in terms of the number of datasets, they are exclusively focused on node classification tasks. The paper states that the theory supports node regression as well, but no empirical validation is provided for this. This limits the empirical scope of the claims.

- The computational complexity of the algorithm is not thoroughly explained. For example, it involves solving the least square transformations into each layer which might be expensive for large graphs. The scalability aspect and computational cost should be discussed more, both theoretically and empirically in experiments. This shed light on the practicability of the proposed algorithm.

- The experimental study is limited and only compared with very few baselines. Moreover, the gains seem relatively small, which is hard to tell the exact benefits. More ablation studies and sensitivity analysis could benefit the paper.

---

> ### Author Rebuttal · Authors · 2025-07-31
>
> We thank the reviewer for their thoughtful feedback and for recognizing the novelty, ingenuity, and theoretical contributions of our approach.
>
> > W1: While the experiments are extensive... they are exclusively focused on node classification tasks...
>
> We agree with this concern. Our evaluation is restricted to node classification – not by design, but due to the absence of public node-level regression benchmarks. We surveyed 27 datasets from PyG, DGL, SNAP, and OGB and found no such benchmarks, reflecting a broader issue in the field. As shown in [1], progress of graph learning is bottlenecked by the lack of representative benchmarks. Still, the purpose of our paper is to lay the groundwork for future GFMs that may perform node-level regression, with the hope that better benchmarks will be available in the future.
>
> > W2: The computational complexity of the algorithm is not thoroughly explained...
>
> Thank you for raising this important aspect. Given a graph, we first perform (a non-learnable, standard) local aggregation over each node’s neighborhood and then solve a least-squares (LS) problem as a one-time preprocessing step. The resulting features, along with the original inputs, are passed into the TS-GNN, which performs standard message-passing (MP) during the forward pass.
>
> For a graph with $N$ nodes, $E$ edges, and $F$-dimensional node features, the local aggregation and MP steps each have time complexity $\mathcal{O}(EF)$, while the LS solution has complexity $\mathcal{O}(NF^2+F^3)$. Importantly, the LS computation is performed once per graph and does not scale with the number of layers or training epochs.
>
> GraphAny, the only other zero-shot baseline, also applies local aggregation and LS with varying neighborhood radii, followed by a transformer. Its complexity is also $\mathcal{O}(EF)$, comparable to our method.
>
> We will include in the final version a comprehensive theoretical analysis of the runtime, comparing our method with GraphAny and classical GNNs.
>
> > W3: The experimental study is limited and only compared with very few baselines
>
> Unfortunately, GraphAny is the only existing method that matches this setting. Due to the lack of baselines, we also compare with standard end-to-end GNNs, although they are not designed for zero-shot generalization and thus do not constitute a fair comparison.
>
> That said, in response to your suggestion, we extended Table 1 to include three more GNN variants: TS-SGC, TS-GCN, and TS-GCNII. Below is the average accuracy over 27 datasets:
>
> ||GraphAny|TS-Mean|TS-GAT|TS-SGC|TS-GCN|TS-GCNII|
> |-|-|-|-|-|-|-|
> |Average|65.83±1.66|**66.21±1.93**|65.85±2.38|64.71±3.06 |64.39±2.61|62.45±3.43|
>
> The results reveal that as GNNs become more complex -- from SGC to GCN to GCNII, performance degrades. This supports a hypothesis that complex GNNs, while beneficial when training and testing on the same graph, may overfit to training distributions and generalize poorly in transfer scenarios.
>
> > Moreover, the gains seem relatively small, which is hard to tell the exact benefits
>
> This is an excellent point. Due to time constraints, we were unable to include experiments demonstrating that the benefits of TS-GNNs over GraphAny are evident in the high-data regime, where the model is trained on multiple graphs. That is because GraphAny can only mix a set of fixed patterns (the least squares solutions), whereas TS-GNNs can learn complex patterns that improve as more training graphs are introduced. Moreover, training on a single graph is not only limiting but also unrealistic for a foundation model setting.
>
> We added a new experiment where TS-Mean and GraphAny are trained on increasingly larger subsets (1, 3, 5, 7, 9) of 9 held-out training graphs, with each subset nested within the next. Zero-shot evaluation is performed on the remaining 19 datasets plus the newly added wisconsin dataset.
>
> |#Train graphs|1|3|5|7|9|
> |-|-|-|-|-|-|
> |GraphAny|66.62|**67.34**|67.29|67.28|67.22|
> |TS-Mean|67.23|67.36|67.82|68.42|**68.57**|
>
> Results show that **TS-Mean’s accuracy increases with more training graphs**, exhibiting the desired scaling law behavior. This behavior aligns with the expected characteristics of a GFM – the ability to benefit from increased data diversity. In contrast, **GraphAny’s performance remains unchanged**. This suggests that GraphAny is not a well-suited candidate for a foundation model, and may instead be better tailored only to scenarios with limited training data.
>
> > More ablation studies and sensitivity analysis could benefit the paper
>
> We agree and have provided sensitivity analyses of two types:
>
> **(1) Least-Squares mixing and bias term**
>
> We isolate the effect of (a) the least squares (LS) feature-label mixing, and (b) the inclusion of a bias term. We first evaluate a non-parametric setup across our test set of 27 datasets. We compare the accuracy of LS alone, LS with a bias term, and LS with both a bias term and a subsequent local mean aggregation step (similar to MEAN-GNN), exhibiting similar performance.
>
> ||random|LS|LS+bias|LS+bias+mean|
> |-|-|-|-|-|
> |Average|20.98|58.39±1.24|58.75±1.58|59.00±2.11|
>
> We then assess the impact of these components on TS-Mean and TS-GAT, trained on Cora and evaluated on the 27 datasets. We assess three configurations: without LS and bias (plain), with LS only, and with both LS+bias.
>
> ||plain|+LS|+LS+bias|
> |-|-|-|-|
> |TS-Mean|55.43±1.12|63.21±2.08|**66.21±1.93**|
> |TS-GAT|55.19±1.54|62.66±2.20|**65.85±2.38**|
>
> These results show that the plain versions of TS-Mean and TS-GAT slightly lag behind the LS. This is expected, as the global pooling in plain TS-GNNs creates an information bottleneck that prevents feature-label mixing, as noted in lines 265–270. Despite this limitation, the plain TS-GNN variants still perform substantially better than random guessing, indicating that it captures transferable structure even without the LS step or bias term.
>
> Next, the inclusion of the LS step alone offers a performance boost, suggesting that LS and TS-GNNs capture complementary patterns. The combination of the two yields greater generalization than either alone. This analysis parallels GraphAny's, where entropy normalization and distance features improve over plain GraphAny.
>
> **(2) Sensitivity to pretraining dataset**
>
> We explore how sensitive our method is when a single dataset is used in training. In addition to Cora, we train TS-Mean on roman-empire, chameleon, and wiki-cs. All models are evaluated zero-shot on the same 27 test graphs.
>
> ||cora|roman-empire|chameleon|wiki-cs|average|
> |-|-|-|-|-|-|
> |TS-Mean|65.96±1.90|64.63±1.33|65.66±2.60|64.52±2.34|65.19±2.04|
> |GraphAny|64.63±1.42|65.15±1.61|65.80±1.55|65.11±1.53|65.17±1.53|
>
> These results further validate that TS-Mean and GraphAny are consistently competitive in the low-data regime, aligning with our earlier results over varying training set size. However, as explained above, the true advantage of TS-GNNs emerges in the foundation model setting – the high-data regime.
>
> > Q1: Could the authors discuss potential future work on extending this recipe to more complex graph structures, such as heterogeneous graphs...
>
> One route for extending TS-GNNs to heterogeneous is to incorporate the permutation symmetry $S_R$ over the set of relation types $R$.  We derive theoretical guarantees over sets and transform the resulting global pooling to a local graph pooling, similar to the expressivity analysis of GIN. For heterogeneous graphs, we can adopt a strategy analogous to that of relational GCNs, treating each relation type as a distinct graph, where we enforce our existing symmetry $S_N\times S_F\times S_C$ for each graph. To handle relations in a principled way, we stack the relation-specific representations such that it is $S_R$-equivariant. This results in an intermediate tensor of size $N\times F\times R\times K$, which is independent of the graph-specific relation types. A similar approach can also handle multiple node types.
>
> Another direction is to combine our principled approach with meta-graph-based foundation models such as ULTRA [2]. These models are not derived from Schur's Lemma and a universal approximation analysis, but may complement our method in practice.
>
> > Q2: What about the scalability of the proposed method?...compare this with large pre-trained language models or vision models...
>
> First, we clarify that our method is not limited to small-scale graphs. In our experiments, we include large-scale datasets such as arxiv, which contains >150K nodes and >1M edges, demonstrating that our method scales to large graphs.
>
> As noted in the paper, our approach is task-agnostic and applies to any node-level task, including node regression, node classification, and node ranking. We focus on classification due to dataset availability (see W1).
>
> Regarding large pre-trained models in NLP/vision: our work shares the same philosophy—learning transferable representations across tasks and domains. Just as BERT/ViT is able to generalize across language/image data, TS-GNNs aim to learn generalized node-level representations across graphs. While the scale of pretraining in graph learning is in an early stage compared to NLP/vision, our results already show scaling behavior – performance improves with more training graphs (see W3). Due to the current limitations in large-scale node-based graph datasets [1], we are not yet able to assess our model’s behavior on large graph collections.
>
> > The paper does not include explicit limitation sections
>
> We will include a dedicated limitations section.
>
> [1] Position: Graph Learning Will Lose Relevance Due To Poor Benchmarks, ICML25
>
> [2] Towards Foundation Models for Knowledge Graph Reasoning, ICLR24
>
> **We will include all the experimental results and suggestions provided by the Reviewer in the final version with the appropriate discussion.**
>
> We hope our responses address your concerns and would greatly appreciate your reconsideration of the evaluation and increasing your score.

---

> > ### Author Response · Authors · 2025-08-03
> >
> > Thank you for encouraging a stronger empirical demonstration of the benefits of TS-GNNs. We believe to have addressed all of your concerns. As the discussion period is closing soon, we would highly appreciate feedback at this stage since this would give us a chance to address any remaining issues. We look forward to hearing from you. Thank you.

---

> > ### Comment · Reviewer_KTq8 · 2025-08-08
> >
> > Thank the authors for providing more details and further experimental results! Many of the experimental concerns were resolved. There are some remaining concerns of generalizability and scale of the approach (empirically), which the authors acknowledge. I would encourage the authors to include all the results and analysis.

---

### Note · Authors · 2025-08-14

We addressed each concern in detail in our individual responses to the Reviewers, resulting in highly positive feedback and scores. **We thank the reviewers for their valuable suggestions and the AC for monitoring the discussion.** Below are our final remarks.

**The novelty of our work:**

* We introduce a practical method to transform **any GNN architecture into a GFM.**

* We characterize linear maps, equivariant to permutations of nodes, features, and labels, proving our architecture is a **universal function approximator over sets.**

* We evaluate on 27 real-world datasets, establishing **the first GFM to hint at a scaling law** -- improved zero-shot accuracy with more training data.

**Reviewers unanimously acknowledged the novelty of our work:**

"*This distinguishes the work from many empirically-driven approaches.*" -- Reviewer KTq8

"*The authors propose a novel model ... the theoretical claims are well-supported*" -- Reviewer pt6w

"*...filling the gap in the theoretical foundation of graph foundation models.*" -- Reviewer HcMt

"*I really like the motivation behind this work... really promising and intuitive.*" -- Reviewer Yes9

**Reviewers' main request – additional experiments: we address this with 6 extensive new experiments each on a set of 27 graphs.**

(1) **Added variants TS-SGC, TS-GCN, & TS-GCNII** to Table 1. Interestingly, complex GNNs, while helpful when training on the same graph, may overfit and generalize poorly in transfer settings.

(2) **Scaling law behavior of TS-GNNs.** Training TS-Mean and GraphAny on progressively larger subsets (1, 3, 5, 7, 9) of 9 held-out graphs shows TS-GNNs as the first GFM to hint at a scaling law -- accuracy improves with more training graphs, while GraphAny remains unchanged.

(3) **Importance of the least-Squares (LS) mixing and bias term.** We test TS-GNNs without LS/bias, with LS only, and with both, showing incremental gains from each.

(4) **Sensitivity to pretraining dataset.** We train TS-Mean and GraphAny using 4 different single training graphs and show competitive zero-shot performance in this setting.

(5) **Necessity of symmetry assumptions.** We test DeepSets, DSS, and TS-GNNs in the zero-shot setting, showing that without the triple-symmetry, accuracy is near-random.

(6) **TS-Nets vs DSS vs DeepSets as linear layers.** We show that even when DeepSets and DSS are fine-tuned on test graphs, TS-Mean outperforms both.

**We will include all experimental results in the final version.**

---

### Decision · Program_Chairs · 2025-09-17

**Decision:**

Accept (poster)

**Comment:**

This paper proposes a method  for designing graph foundation models (GFMs) for node classification and regression tasks. The paper argues that a GFM must respect three symmetries: it must be equivariant to node and label permutations and invariant to permutations of input features. The TSNet model is then introduced, a universal approximator on multisets that respect the aforementioned symmetries. The proposed method is applied to the multisets of features induced by the local graph aggregation and can be used to transform any GNN model into a GFM. The proposed method is evaluated on a large number of real-world node classification datasets.

The reviewers agree that the development of GFMs is an important topic, and this paper makes a significant contribution in this direction. One of the strengths of the proposed method is that it is very general since it can transform any GNN model into a GFM. TSNet's expressive power is supported by theoretical guarantees, as it is proven to be a universal approximator on multisets that respect the considered symmetries. The reviewers appreciate the breadth of the experiments, which cover 27 node classification datasets. They also acknowledged that the paper is generally well-written and easy to follow. The main weakness of the paper is that it can currently deal only with node-level tasks. The reviewers also raised concerns about the computational complexity of the proposed method, the marginal improvements over GraphAny and the empirical validation of the method. Specifically, the method was not evaluated on any node regression dataset, no social media datasets were considered, the method was compared against a single baseline, while the different methods were trained only on the Cora dataset which raises questions about the significance of the results. Moreover, no ablation study was conducted to assess the contribution of the different components to the method's performance.

During the rebuttal, the authors conducted further experiments and addressed most of the raised concerns. For instance, they conducted experiments in which the different methods were trained on datasets other than Cora, as well as on subsets of the full dataset collection to demonstrate that the proposed method achieves greater performance gains over GraphAny when trained on multiple graphs. They also introduced three new variants of the proposed method. They claimed that there is a lack of node regression datasets, while they also conducted an ablation study to isolate the effect of the proposed method's key components. Finally, the authors acknowledged that the method cannot be applied to graphs of the scale seen in social media datasets.

To summarize, the problem addressed in this paper is highly relevant. The paper offers a novel solution for the development of GFMs. As already discussed, one of the main limitations of the proposed method is that it can only deal with node-level tasks. However, given that many problems can be framed as node-level tasks, it is likely that the method can have significant practical relevance. While the proposed method is evaluated on a large number of datasets, I believe that the evaluation of the method could be further improved. GFMs are expected to be trained on large amounts of data and subsequently employed for inference on new, unseen data. This paper follows the opposite direction. The model is trained on a single dataset and is evaluated on a very large number of datasets. In their response, the authors reported new experiments where the models were trained on increasingly larger subsets of datasets. However, further experiments are necessary to fully evaluate the model's potential, and I would suggest the authors conduct these experiments and report them in the next version of the paper. Based on the above, I am leaning towards acceptance. I strongly suggest the authors incorporate reviewer feedback into the next version of the paper.